# Supervised Kernel Thinning

**Albert Gong**    **Kyuseong Choi**    **Raaz Dwivedi**

Cornell Tech, Cornell University

`agong,kc728,dwivedi@cornell.edu`

## Abstract

The kernel thinning algorithm of [10] provides a better-than-i.i.d. compression of a generic set of points. By generating high-fidelity coresets of size significantly smaller than the input points, KT is known to speed up unsupervised tasks like Monte Carlo integration, uncertainty quantification, and non-parametric hypothesis testing, with minimal loss in statistical accuracy. In this work, we generalize the KT algorithm to speed up supervised learning problems involving kernel methods. Specifically, we combine two classical algorithms—Nadaraya-Watson (NW) regression or kernel smoothing, and kernel ridge regression (KRR)—with KT to provide a *quadratic* speed-up in both training and inference times. We show how distribution compression with KT in each setting reduces to constructing an appropriate kernel, and introduce the Kernel-Thinned NW and Kernel-Thinned KRR estimators. We prove that KT-based regression estimators enjoy significantly superior computational efficiency over the full-data estimators and improved statistical efficiency over i.i.d. subsampling of the training data. En route, we also provide a novel multiplicative error guarantee for compressing with KT. We validate our design choices with both simulations and real data experiments.

## 1   Introduction

In supervised learning, the goal of coreset methods is to find a representative set of points on which to perform model training and inference. On the other hand, coreset methods in *unsupervised* learning have the goal of finding a representative set of points, which can then be utilized for a broad class of downstream tasks—from integration [10, 9] to non-parametric hypothesis testing [8]. This work aims to bridge these two research threads.

Leveraging recent advancements from compression in the unsupervised setting, we tackle the problem of non-parametric regression (formally defined in Sec. 2). Given a dataset of $n$ i.i.d. samples, $(x_i, y_i)_{i=1}^n$, we want to learn a function $f$ such that $f(x_i) \approx y_i$. The set of allowable functions is determined by the kernel function, which is a powerful building block for capturing complex, non-linear relationships. Due to its powerful performance in practice and closed-form analysis, non-parametric regression methods based on kernels (a.k.a "kernel methods") have become a popular choice for a wide range of supervised learning tasks [13, 20, 24].

There are two popular approaches to non-parametric kernel regression. First, perhaps a more classical approach, is kernel smoothing, also referred to as Nadaraya-Watson (NW) regression. The NW estimator at a point $x$ is effectively a smoothing of labels $y_i$ such that $x_i$ is close to $x$. These weights are computed using the kernel function (see Sec. 2 for formal definitions). Importantly, the NW estimator takes $\Theta(n)$ pre-processing time (to simply store the data) and $\Theta(n)$ inference time for each test point $x$ ($n$ kernel evaluations and $n$ simple operations).

Another popular approach is kernel ridge regression (KRR), which solves a non-parametric least squares subject to the regression function lying in the reproducing kernel Hilbert space (RKHS) of a

specified reproducing kernel function. Remarkably, KRR admits a closed-form solution via inverting the associated kernel matrix, and takes $\mathcal{O}(n^3)$ training time and $\Theta(n)$ inference time for each test point $x$.

Our goal is to overcome the computational bottlenecks of kernel methods, while retaining their favorable statistical properties. Previous attempts at using coreset methods include the work of Boutsidis et al. [4], Zheng and Phillips [32], Phillips [18], which depend on a projection type compression, having similar spirit to the celebrated Johnson–Lindenstrauss lemma, a metric preserving projection result. So accuracy and running depend unfavorably on the desired statistical error rate. Kpotufe [14] propose an algorithm to reduce the query time of the NW estimator to $\mathcal{O}(\log n)$, but the algorithm requires super-linear preprocessing time.

Other lines of work exploit the structure of kernels more directly, especially in the KRR literature. A slew of techniques from numerical analysis have been developed, including work on Nyström subsampling by El Alaoui and Mahoney [11], Avron et al. [1], Díaz et al. [7]. Camoriano et al. [5] and Rudi et al. [22] combine early stopping with Nyström subsampling. Though more distant from our approach, we also note the approach of Rahimi and Recht [21] using random features, Zhang et al. [31] using Divide-and-Conquer, and Tu et al. [28] using block coordinate descent.

**Our contributions.**   In this work, we show how coreset methods can be used to speed up both training and inference in non-parametric regression for a large class of function classes/kernels. At the heart of these algorithms is a general procedure called kernel thinning [10, 9], which provides a worst-case bound on integration error (suited for problems in the original context of unsupervised learning and MCMC simulations). In Sec. 3, we introduce a meta-algorithm that recovers our two thinned non-parametric regression methods each based on NW and KRR. We introduce the *kernel-thinned Nadaraya-Watson estimator* (KT-NW) and the *kernel-thinned kernel ridge regression* estimator (KT-KRR).

We show that KT-NW requires $\mathcal{O}(n \log^3 n)$ time during training and $\mathcal{O}(\sqrt{n})$ time at inference, while achieving a mean square error (MSE) rate of $n^{-\frac{\beta}{\beta+d}}$ (Thm. 1)—a strict improvement over uniform subsampling of the original input points. We show that KT-KRR requires $\mathcal{O}(n^{3/2})$ time during training and and $\mathcal{O}(\sqrt{n})$ time during inference, while achieving an near-minimax optimal rate of $\frac{m \log n}{n}$ when the kernel has finite dimension (Thm. 2). We show how our KT-KRR guarantees can also be extended to the infinite-dimension setting (Thm. 3). In Sec. 5, we apply our proposed methods to both simulated and real-world data. In line with our theory, KT-NW and KT-KRR outperform standard thinning baselines in terms of accuracy while retaining favorable runtimes.

## 2   Problem setup

We now formally describe the non-parametric regression problem. Let $x_1, \ldots, x_n$ be i.i.d. samples from the data distribution $\mathbb{P}$ (with density $p$) over the domain $\mathcal{X} \subset \mathbb{R}^d$ and $w_1, \ldots, w_n$ be i.i.d. samples from $\mathcal{N}(0, 1)$. In the sequel, we use $\| \cdot \|$ to denote the Euclidean norm unless otherwise stated. Then define the response variables $y_1, \ldots, y_n$ by the follow data generating process:

$$y_i \triangleq f^\star(x_i) + v_i \quad \text{for} \quad i = 1, 2, \ldots, n, \tag{1}$$

where $f^\star : \mathcal{X} \to \mathcal{Y} \subset \mathbb{R}$ is the *regression function* and $v_i \triangleq \sigma w_i$ for some noise level $\sigma > 0$. Our task is to build an estimate for $f^\star$ given the $n$ observed points, denoted by

$$\mathcal{S}_{\text{in}} \triangleq ((x_1, y_1), \ldots, (x_n, y_n)).$$

**Nadaraya-Watson (NW) estimator.**   A classical approach to estimate the function $f^\star$ is kernel smoothing, where one estimates the function value at a point $z$ using a weighted average of the observed outcomes. The weight for outcome $y_i$ depends on how close $x_i$ is to the point $z$; let $\kappa : \mathbb{R}^d \to \mathbb{R}$ denote this weighting function such that the weight for $x_i$ is proportional to $\kappa(\|x_i - z\|/h)$ for some bandwidth parameter $h > 0$. Let $\mathbf{k} : \mathbb{R}^d \times \mathbb{R} \to \mathbb{R}$ denote a shift-invariant kernel defined as

$$\mathbf{k}(x_1, x_2) = \kappa(\|x_1 - x_2\|/h).$$

Then this smoothing estimator, also known as Nadaraya-Watson (NW) estimator, can be expressed as

$$\widehat{f}(\cdot) \triangleq \frac{\sum_{(x,y) \in \mathcal{S}_{\text{in}}} \mathbf{k}(\cdot, x) y}{\sum_{x \in \mathcal{S}_{\text{in}}} \mathbf{k}(\cdot, x)} \tag{2}$$

whenever the denominator in the above display is non-zero. In the case the denominator in (2) is zero, we can make a default choice, which for simplicity here we choose as zero. We refer to the estimator (2) as FULL-NW estimator hereafter. One can easily note that FULL-NW requires $\mathcal{O}(n)$ storage for the input points and $\mathcal{O}(n)$ kernel queries for inference at each point.

**Kernel ridge regression (KRR) estimator.** Another popular approach to estimate $f^\star$ is that of non-parametric (regularized) least squares. The solution in this approach, often called as the kernel ridge regression (KRR), is obtained by solving a least squares objective where the fitted function is posited to lie in the RKHS $\mathcal{H}$ of a reproducing kernel $\mathbf{k}$, and a regularization term is added to the objective to avoid overfitting.[1] Overall, the KRR estimate is the solution to the following regularized least-squares objective, where $\lambda > 0$ denotes a regularization hyperparameter:

$$\min_{f \in \mathcal{H}} L_{\mathcal{S}_{\text{in}}} + \lambda \|f\|_{\mathbf{k}}^2, \quad \text{where} \quad L_{\mathcal{S}_{\text{in}}} \triangleq \frac{1}{n} \sum_{(x,y) \in \mathcal{S}_{\text{in}}} (f(x) - y)^2. \tag{3}$$

Like NW, an advantage of KRR is the existence of a closed-form solution

$$\widehat{f}_{\text{full},\lambda}(\cdot) \triangleq \sum_{i=1}^n \alpha_i \mathbf{k}(\cdot, x_i) \quad \text{where} \tag{4}$$

$$\boldsymbol{\alpha} \triangleq (\mathbf{K} + n\lambda \mathbf{I}_n)^{-1} \begin{bmatrix} y_1 \\ \vdots \\ y_n \end{bmatrix} \in \mathbb{R}^n \quad \text{and} \quad \mathbf{K} \triangleq [\mathbf{k}(x_i, x_j)]_{i,j=1}^n \in \mathbb{R}^{n \times n}. \tag{5}$$

Notably, the estimate $\widehat{f}_{\text{full},\lambda}$, which we refer to as the FULL-KRR estimator, can also be seen as yet another instance of weighted average of the observed outcomes. Notably, NW estimator imposes that the weights across the points sum to $1$ (and are also non-negative whenever $\mathbf{k}$ is), KRR allows for generic weights that need not be positive (even when $\mathbf{k}$ is) and need not sum to $1$. We note that naïvely solving $\widehat{f}_{\text{full},\lambda}$ requires $\mathcal{O}(n^2)$ kernel evaluations to compute the kernel matrix, $\mathcal{O}(n^3)$ to compute a matrix inverse, and $\mathcal{O}(n)$ kernel queries for inference at each point. One of our primary goals in this work is to tackle this high computational cost of FULL-KRR.

## 3 Speeding up non-parametric regression

We begin with a general approach to speed up regression by thinning the input datasets. While computationally superior, a generic approach suffers from a loss of statistical accuracy motivating the need for a strategic thinning approach. To that end, we briefly review kernel thinning and finally introduced our supervised kernel thinning approach.

### 3.1 Thinned regression estimators: Computational and statistical tradeoffs

Our generic approach comprises two main steps. First, we compress the input data by choosing a coreset $\mathcal{S}_{\text{out}} \subset \mathcal{S}_{\text{in}}$ of size $n_{\text{out}} \triangleq |\mathcal{S}_{\text{out}}|$. Second, we apply our off-the-shelf non-parametric regression methods from Sec. 2 to the compressed data. By setting $n_{\text{out}} \ll n$, we can obtain notable speed-ups over the FULL versions of NW and KRR.

Before we introduce the thinned versions of NW and KRR, let us define the following notation. Given an input sequence $\mathcal{S}_{\text{in}}$ and output sequence $\mathcal{S}_{\text{out}}$, define the empirical probability measures

$$\mathbb{P}_{\text{in}} \triangleq \frac{1}{n} \sum_{(x,y) \in \mathcal{S}_{\text{in}}} \delta_{(x,y)} \quad \text{and} \quad \mathbb{Q}_{\text{out}} \triangleq \frac{1}{n_{\text{out}}} \sum_{(x,y) \in \mathcal{S}_{\text{out}}} \delta_{(x,y)}. \tag{6}$$

**Thinned NW estimator.** The thinned NW estimator is the analog of Full-NW except that $\mathcal{S}_{\text{in}}$ is replaced by $\mathcal{S}_{\text{out}}$ in (2) so that the *thinned-NW estimator* is given by

$$\widehat{f}_{\mathcal{S}_{\text{out}}}(\cdot) \triangleq \frac{\sum_{(x,y) \in \mathcal{S}_{\text{out}}} \mathbf{k}(\cdot, x) y}{\sum_{x \in \mathcal{S}_{\text{out}}} \mathbf{k}(\cdot, x)} = \frac{\mathbb{Q}_{\text{out}}(y\mathbf{k})}{\mathbb{Q}_{\text{out}}\mathbf{k}} \tag{7}$$

---

[1] We note that while KRR approach (14) does require $\mathbf{k}$ to be reproducing, the NW approach (2) in full generality is valid even when $\mathbf{k}$ is a not a valid reproducing kernel.

whenever the denominator in the display is not zero; and $0$ otherwise. When compared to the FULL-NW estimator, we can easily deduce the computational advantage of this estimator: more efficient $\mathcal{O}(n_{\text{out}})$ storage as well as the faster $\mathcal{O}(n_{\text{out}})$ computation for inference at each point.

**Thinned KRR estimator.** Similarly, we can define the *thinned KRR estimator* as

$$\widehat{f}_{\mathcal{S}_{\text{out}},\lambda'}(\cdot) = \sum_{i=1}^{n_{\text{out}}} \alpha_i' \mathbf{k}(\cdot, x_i'), \quad \text{where} \tag{8}$$

$$\boldsymbol{\alpha}' \triangleq (\mathbf{K}' + n_{\text{out}}\lambda' \mathbf{I}_{n_{\text{out}}})^{-1} \begin{bmatrix} y_1' \\ \vdots \\ y_{n_{\text{out}}}' \end{bmatrix} \in \mathbb{R}^{n_{\text{out}}} \quad \text{and} \quad \mathbf{K}' \triangleq [\mathbf{k}(x_i', x_j')]_{i,j=1}^{n_{\text{out}}} \in \mathbb{R}^{n_{\text{out}} \times n_{\text{out}}}$$

given some regularization parameter $\lambda' > 0$. When compared to FULL-KRR, $\widehat{f}_{\mathcal{S}_{\text{out}},\lambda'}$ has training time $\mathcal{O}(n_{\text{out}}^3)$ and prediction time $\mathcal{O}(n_{\text{out}})$.

A baseline approach is standard thinning, whereby we let $\mathcal{S}_{\text{out}}$ be an i.i.d. sample of $n_{\text{out}} = \sqrt{n}$ points from $\mathcal{S}_{\text{in}}$. For NW, let us call the resulting $\widehat{f}_{\mathcal{S}_{\text{out}}}$ (7) the standard-thinned Nadaraya-Watson (ST-NW) estimator. When $n_{\text{out}} = \sqrt{n}$, ST-NW achieves an excess risk rate of $\mathcal{O}(n^{-\frac{\beta}{2\beta+d}})$ compared to the FULL-NW rate of $\mathcal{O}(n^{-\frac{2\beta}{2\beta+d}})$. For KRR, let us call the resulting $\widehat{f}_{\mathcal{S}_{\text{out}},\lambda'}$ (8) the standard-thinned KRR (ST-KRR) estimator. When $n_{\text{out}} = \sqrt{n}$, ST-KRR achieves an excess risk rate of $\mathcal{O}(\frac{m}{n_{\text{out}}})$ compared to the FULL-KRR rate of $\mathcal{O}(\frac{m}{n})$. Our goal is to provide good computational benefits without trading off statistical error. Moreover, we may be able to do better by leveraging the underlying geometry of the input points and summarize of the input distribution more succinctly than i.i.d. sampling.

## 3.2 Background on kernel thinning

A subroutine central to our approach is kernel thinning (KT) from Dwivedi and Mackey [10, Alg. 1]. We use a variant called KT-COMPRESS++ from Shetty et al. [23, Ex. 6] (see full details in App. A), which provides similar approximation quality as the original KT algorithm of Dwivedi and Mackey [10, Alg. 1], while reducing the runtime from $\mathcal{O}(n^2)$ to $\mathcal{O}(n \log^3 n)$.[2] Given an input kernel $\mathbf{k}_{\text{ALG}}$ and input points $\mathcal{S}_{\text{in}}$, KT-COMPRESS++ outputs a coreset $\mathcal{S}_{\text{KT}} \subset \mathcal{S}_{\text{in}}$ with size $n_{\text{out}} \triangleq \sqrt{n} \ll n$. In this work, we leverage two guarantees of KT-COMPRESS++. Informally, $\mathcal{S}_{\text{KT}}$ satisfies (with high probability):

$$(L^\infty \text{ bound}) \quad \|(\mathbb{P}_{\text{in}} - \mathbb{Q}_{\text{out}})\mathbf{k}_{\text{ALG}}\|_\infty \leq C_1 \frac{\sqrt{d} \log n_{\text{out}}}{n_{\text{out}}} \tag{9}$$

$$(\text{MMD bound}) \quad \sup_{\|h\|_{\mathbf{k}_{\text{ALG}}} \leq 1} |(\mathbb{P}_{\text{in}} - \mathbb{Q}_{\text{out}})h| \leq C_2 \frac{\sqrt{\log n_{\text{out}} \cdot \log \mathcal{N}_{\mathbf{k}_{\text{ALG}}}(\mathbb{B}_2(\mathfrak{R}_{\text{in}}), 1/n_{\text{out}})}}{n_{\text{out}}}, \tag{10}$$

where $C_1, C_2 > 0$ are constants that depend on the properties of the input kernel $\mathbf{k}_{\text{ALG}}$ and the chosen failure probability of KT-COMPRESS++, $\mathfrak{R}_{\text{in}}$ characterizes the radius of $\{x_i\}_{i=1}^n$, and $\mathcal{N}_{\mathbf{k}_{\text{ALG}}}(\mathbb{B}_2(\mathfrak{R}_{\text{in}}), 1/n_{\text{out}})$ denotes the kernel covering number of $\mathcal{H}(\mathbf{k}_{\text{ALG}})$ over the ball $\mathbb{B}_2(\mathfrak{R}_{\text{in}}) \subset \mathbb{R}^d$ at a specified tolerance (see Sec. 4.2 for formal definitions).

At its highest level, KT provides good approximation of function averages. The bound (9) (formally stated in Lem. 1) controls the worst-case point-wise error, and is near-minimax optimal by Phillips and Tai [19, Thm. 3.1]. In the sequel, we leverage this type of result to derive generalization bounds for the kernel smoothing problem. The bound (10) (formally stated in Lem. 2) controls the integration error of functions in $\mathcal{H}(\mathbf{k}_{\text{ALG}})$ and is near-minimax optimal by Tolstikhin et al. [26, Thm. 1, 6]. In the sequel, we leverage this type of result to derive generalization bounds for the KRR problem.

---

[2]In the sequel, we use "KT" and "KT-COMPRESS++" interchangeably since the underlying algorithm (kernel halving [10, Alg. 1a]) and associated approximation guarantees are the same up to small constant factors.

### 3.3 Supervised kernel thinning

We show how the approximation results from kernel thinning can be extended to the regression setting. We construct two meta-kernels, the Nadaraya-Watson meta-kernel $\mathbf{k}_{\mathrm{NW}}$ and the ridge-regression meta-kernel $\mathbf{k}_{\mathrm{RR}}$, which take in a *base kernel* $\mathbf{k}$ (defined over $\mathcal{X}$ only) and return a new kernel (defined over $\mathcal{X} \times \mathcal{Y}$). When running KT, we set this new kernel as $\mathbf{k}_{\mathrm{ALG}}$.

#### 3.3.1 Kernel-thinned Nadaraya-Watson regression (KT-NW)

A tempting choice of kernel for KT-NW is the kernel $\mathbf{k}$ itself. That is, we can thin the input points using the kernel

$$\mathbf{k}_{\mathrm{ALG}}((x_1, y_1), (x_2, y_2)) \triangleq \mathbf{k}(x_1, x_2). \tag{11}$$

This choice is sub-optimal since it ignores any information in the response variable $y$. For our supervised learning set-up, perhaps another intuitive choice would be to use KT with

$$\mathbf{k}_{\mathrm{ALG}}((x_1, y_1), (x_2, y_2)) \triangleq \mathbf{k}((x_1, y_1), (x_2, y_2)), \tag{12}$$

where $(x, y)$ denotes the concatenation of $x$ and $y$. While this helps improve performance, there remains a better option as we illustrate next.

In fact, a simple but critical observation immediately reveals a superior choice of the kernel to be used in KT for NW estimator. We can directly observe that the NW estimator is a ratio of the averages of two functions:

$$f_{\mathrm{numer}}(x, y)(\cdot) \triangleq \mathbf{k}(x, \cdot)\langle y, 1\rangle_{\mathbb{R}}$$
$$\text{and} \quad f_{\mathrm{denom}}(x, y)(\cdot) \triangleq \mathbf{k}(x, \cdot),$$

over the empirical distribution $\mathbb{P}_{\mathrm{in}}$ (6). Recall that KT provides a good approximation of sample means of functions in an RKHS, so it suffices to specify a "correct" choice of the RKHS (or equivalently the "correct" choice of the reproducing kernel). We can verify that $f_{\mathrm{denom}}$ lies in the RKHS associated with kernel $\mathbf{k}(x_1, x_2)$ and $f_{\mathrm{numer}}$ lies in the RKHS associated with kernel $\mathbf{k}(x_1, x_2) \cdot y_1 y_2$. This motivates our definition for the Nadaraya-Watson kernel:

$$\mathbf{k}_{\mathrm{NW}}((x_1, y_1), (x_2, y_2)) \triangleq \mathbf{k}(x_1, x_2) + \mathbf{k}(x_1, x_2) \cdot y_1 y_2 \tag{13}$$

since then we do have $f_{\mathrm{denom}}, f_{\mathrm{numer}} \in \mathcal{H}(\mathbf{k}_{\mathrm{NW}})$. Intuitively, thinning with $\mathbf{k}_{\mathrm{RR}}$ should simultaneously provide good approximation of averages of $f_{\mathrm{denom}}$ and $f_{\mathrm{numer}}$ over $\mathbb{P}_{\mathrm{in}}$ (see the formal argument in Sec. 4.1). When $\mathcal{S}_{\mathrm{out}} = \text{KT-COMPRESS++}(\mathcal{S}_{\mathrm{in}}, \mathbf{k}_{\mathrm{NW}}, \delta)$, we call the resulting solution to (7) the kernel-thinned Nadaraya-Watson (KT-NW) estimator, denoted by $\widehat{f}_{\mathrm{KT}}$.

As we show in Fig. 1(a), this theoretically principled choice does provide practical benefits in MSE performance across sample sizes.

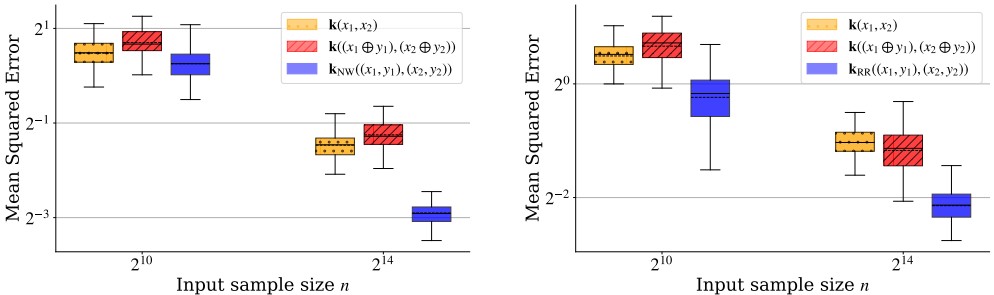

(a) NW ablation with Wendland(0) base kernel    (b) KRR ablation with Gaussian base kernel

Figure 1: **MSE vs choice of kernels**. For exact settings and further discussion see Sec. 5.1.

### 3.3.2 Kernel-thinned kernel ridge regression (KT-KRR)

While with NW estimator, the closed-form expression was a ratio of averages, the KRR estimate (4) can not be expressed as an easy function of averages. However, notice that $L_{\mathcal{S}_{\text{in}}}$ in (3) is an average of the function $\ell_f : \mathcal{X} \times \mathcal{Y} \to \mathbb{R}$ defined as

$$\ell_f(x, y) \triangleq f^2(x) - 2f(x)y + y^2 \quad \text{for} \quad f \in \mathcal{H}(\mathbf{k}).$$

Thus, there may be hope of deriving a KT-powered KRR estimator by thinning $L_{\mathcal{S}_{\text{in}}}$ with the appropriate kernel. Assuming $f \in \mathcal{H}(\mathbf{k})$, we can verify that $f^2$ lies in the RKHS associated with kernel $\mathbf{k}^2(x_1, x_2)$ and that $-2f(x)y$ lies in the RKHS associated with kernel $\mathbf{k}(x_1, x_2) \cdot y_1 y_2$. We now define the ridge regression kernel by

$$\mathbf{k}_{\text{RR}}((x_1, y_1), (x_2, y_2)) \triangleq \mathbf{k}^2(x_1, x_2) + \mathbf{k}(x_1, x_2) \cdot y_1 y_2 \tag{14}$$

and we can verify that $f^2(x) - 2f(x)y$ lies in the RKHS $\mathcal{H}(\mathbf{k}_{\text{RR}})$.[3] When $\mathcal{S}_{\text{out}} \triangleq$ KT-COMPRESS++$(\mathcal{S}_{\text{in}}, \mathbf{k}_{\text{RR}}, \delta)$, we call the resulting solution to (8) the kernel-thinned KRR (KT-KRR) estimator with regularization parameter $\lambda' > 0$, denoted $\widehat{f}_{\text{KT}, \lambda'}$. We note that the kernel $\mathbf{k}_{\text{RR}}$ also appears in [12, Lem. 4], except our subsequent analysis comes with generalization bounds for the KT-KRR estimator. Like for NW, in Fig. 1(b) we do a comparison for KRR-MSE across many kernel choices and conclude that the choice (14) is indeed a superior choice compared to the base kernel $\mathbf{k}$ and the concatenated kernel (12).

## 4 Main results

We derive generalization bounds of our two proposed estimators. In particular, we bound the mean squared error (MSE) defined by $\|f - f^\star\|_2^2 = \mathbb{E}_X \left[ (f(X) - f^\star(X))^2 \right]$. Our first assumption is that of a well-behaved density on the covariate space. This assumption mainly simplifies our analysis of Nadaraya-Watson and kernel ridge regression, but can in principle be relaxed.

**Assumption 1** (Compact support). *Suppose that $\mathcal{X} \subset \mathbb{B}_2(\mathfrak{R}_{\text{in}}) \subset \mathbb{R}^d$ for some $\mathfrak{R}_{\text{in}} > 0$ and that the density $p$ satisfies $0 < p_{\min} \leq p(x) \leq p_{\max}$ for all $x \in \mathcal{X}$.*

### 4.1 KT-NW

For the analysis of the NW estimator, we define function complexity in terms of Hölder smoothness following prior work [27].

**Definition 1.** *For $L > 0$ and $\beta \in (0, 1]$, a function $f : \mathcal{X} \to \mathbb{R}$ is $(\beta, L)$-Hölder if for all $x_1, x_2 \in \mathcal{X}$, $|f(x_1) - f(x_2)| \leq L\|x_1 - x_2\|^\beta$.*

Our next assumption is that on the kernel. Whereas typically the NW estimator does not require a reproducing kernel, our KT-NW estimator requires that $\mathbf{k}$ be reproducing to allow for valid analysis.

**Assumption 2** (Shift-invariant kernel). *$\mathbf{k}$ is a reproducing kernel function (i.e., symmetric and positive semidefinite) defined by $\mathbf{k}(x_1, x_2) \triangleq \kappa(\|x_1 - x_2\|/h)$, where $h > 0$ and $\kappa : \mathbb{R} \to \mathbb{R}$ is bounded by 1, $L_\kappa$-Lipschitz, square-integrable, and satisfies:*

$$\log(L_\kappa \cdot \kappa^\dagger(1/n)) = \mathcal{O}(\log n), \quad \text{where} \quad \kappa^\dagger(u) \triangleq \sup\{r : \kappa(r) \geq u\}; \tag{15}$$

*Additionally, there must exist constants $c_1, c_2 > 0$ such that*

$$2^{j\beta} \sup_{x \in \mathcal{X}} \int_{\|z\| \in [(2^{j-1} - \frac{1}{2})h, (2^j + \frac{1}{2})h]} \mathbf{k}(x, x - z) dz \leq c_1 \cdot \sup_{x \in \mathcal{X}} \int_{\|z\| \in [0, \frac{1}{2}h]} \mathbf{k}(x, x - z) dz \quad \text{and} \tag{16}$$

$$(2^j + \frac{1}{2})^d 2^{j\beta} \kappa(2^{j-1} - 1) \leq c_2 \cdot \int_0^{\frac{1}{2}} \kappa(u) u^{d-1} du \quad \text{for all} \quad j = 1, 2, \ldots. \tag{17}$$

When $\mathbf{k}$ is a compact kernel, such as Wendland, Sinc, and B-spline, Assum. 2 is easily satisfied. In App. G, we prove that Gaussian and Matérn (with $\nu > d/2 + 1$) kernels also satisfy Assum. 2. We now present our main result for the KT-NW estimator. See App. B for the proof.

---

[3] One might expect the ridge regression kernel to include a term that accounts for $y^2$. However, the generalization bounds turn out to be essentially the same regardless of whether we include this term when defining $\mathbf{k}_{\text{RR}}$.

**Theorem 1** (KT-NW). *Suppose that Assum. 1 and 2 hold and that $f^\star \in \Sigma(\beta, L_f)$ with $\beta \in (0, 1]$. Then for any fixed $\delta \in (0, 1]$, the KT-NW estimator (7) with $n_{\mathrm{out}} = \sqrt{n}$ and bandwidth $h = n^{-\frac{1}{2\beta+2d}}$ satisfies*

$$\|\widehat{f}_{\mathrm{KT}} - f^\star\|_2^2 \le C n^{-\frac{\beta}{\beta+d}} \log^2 n, \tag{18}$$

*with probability at least $1 - \delta$, for some positive constant $C$ that does not depend on $n$.*

Tsybakov and Tsybakov [27], Belkin et al. [3] show that FULL-NW achieves a rate of $\mathcal{O}(n^{-\frac{2\beta}{2\beta+d}})$, which is minimax optimal for the $(\beta, L)$-Hölder function class. Compared to the ST-NW rate of $n^{-\frac{\beta}{2\beta+d}}$, KT-NW achieves strictly better rates for all $\beta > 0$ and $d > 0$, while retaining ST-NW's fast query time of $\mathcal{O}(\sqrt{n})$. Note that our method KT-NW has a training time of $\mathcal{O}(n \log^3 n)$, which is not much more than simply storing the input points.

### 4.2 KT-KRR

We present our main result for KT-KRR using finite-rank kernels. This class of RKHS includes linear functions and polynomial function classes.

**Theorem 2** (KT-KRR for finite-dimensional RKHS). *Assume $f^\star \in \mathcal{H}(\mathbf{k})$, Assum. 1 is satisfied, and $\mathbf{k}$ has rank $m \in \mathbb{N}$. Let $\widehat{f}_{\mathrm{KT},\lambda'}$ denote the KT-KRR estimator with regularization parameter $\lambda' = \mathcal{O}(\frac{m \log n_{\mathrm{out}}}{n \wedge n_{\mathrm{out}}^2})$. Then with probability at least $1 - 2\delta - 2e^{-\frac{\|f^\star\|_{\mathbf{k}}^2}{c_1(\|f^\star\|_{\mathbf{k}}^2 + \sigma^2)}}$, the following holds:*

$$\|\widehat{f}_{\mathrm{KT},\lambda'} - f^\star\|_2^2 \le \frac{C m \cdot \log n_{\mathrm{out}}}{\min(n, n_{\mathrm{out}}^2)} [\|f^\star\|_{\mathbf{k}} + 1]^2 \tag{19}$$

*for some constant $C$ that does not depend on $n$ or $n_{\mathrm{out}}$.*

See App. C for the proof. Under the same assumptions, Wainwright [29, Ex. 13.19] showed that the Full-KRR estimator $\widehat{f}_{\mathrm{full},\lambda}$ achieves the minimax optimal rate of $O(m/n)$ in $O(n^3)$ runtime. When $n_{\mathrm{out}} = \sqrt{n} \log^c n$, the KT-KRR error rates from Thm. 2 match this minimax rate in $\widetilde{O}(n^{3/2})$ time, a (near) quadratic improvement over the Full-KRR. On the other hand, standard thinning-KRR with similar-sized output achieves a quadratically poor MSE of order $\frac{m}{\sqrt{n}}$.

Our method and theory also extend to the setting of infinte-dimensional kernels. To formalize this, we first introduce the notion of kernel covering number.

**Definition 2** (Covering number). *For a kernel $\mathbf{k} : \mathcal{Z} \times \mathcal{Z} \to \mathbb{R}$ with $\mathbb{B}_{\mathbf{k}} \triangleq \{f \in \mathcal{H} : \|f\|_{\mathcal{H}} \le 1\}$, a set $\mathcal{A} \subset \mathcal{Z}$ and $\epsilon > 0$, the covering number $\mathcal{N}_{\mathbf{k}}(\mathcal{A}, \epsilon)$ is the minimum cardinality of all sets $\mathcal{C} \subset \mathbb{B}_{\mathbf{k}}$ satisfying $\mathbb{B}_{\mathbf{k}} \subset \bigcup_{h \in \mathcal{C}} \{g \in \mathbb{B}_{\mathbf{k}} : \sup_{x \in \mathcal{A}} |h(x) - g(x)| \le \epsilon\}$.*

We consider two general classes of kernels.

**Assumption 3.** *For some $\mathfrak{C}_d > 0$, all $r > 0$ and $\epsilon \in (0, 1)$, and $\mathbb{B}_2(r) = \{x \in \mathbb{R}^d : \|x\|_2 \le r\}$, a kernel $\mathbf{k}$ is*

LOGGROWTH$(\alpha, \beta)$   *when*   $\log \mathcal{N}_{\mathbf{k}}(\mathbb{B}_2(r), \epsilon) \le \mathfrak{C}_d \log(e/\epsilon)^\alpha (r+1)^\beta$   *with*   $\alpha, \beta > 0$   *and*

POLYGROWTH$(\alpha, \beta)$   *when*   $\log \mathcal{N}_{\mathbf{k}}(\mathbb{B}_2(r), \epsilon) \le \mathfrak{C}_d (1/\epsilon)^\alpha (r+1)^\beta$   *with*   $\alpha < 2$.

We highlight that the definitions above cover several popular kernels: LOGGROWTH kernels include finite-rank kernels and analytic kernels, like Gaussian, inverse multiquadratic (IMQ), and sinc [9, Prop. 2], while POLYGROWTH kernels includes finitely-many continuously differentiable kernels, like Matérn and B-spline [9, Prop. 3]. For clarity, here we present our guarantee for LOGGROWTH kernels and defer the other case to App. E.

**Theorem 3** (KT-KRR guarantee for infinite-dimensional RKHS). *Suppose Assum. 1 is satisfied and $\mathbf{k}$ is LOGGROWTH$(\alpha, \beta)$ (Assum. 3). Then $\widehat{f}_{\mathrm{KT},\lambda'}$ with $\lambda' = \mathcal{O}(1/n_{\mathrm{out}})$ satisfies the following bound with probability at least $1 - 2\delta - 2e^{-\frac{\|f^\star\|_{\mathbf{k}}^2 \log^\alpha n}{c_1(\|f^\star\|_{\mathbf{k}}^2 + \sigma^2)}}$:*

$$\|\widehat{f}_{\mathrm{KT},\lambda'} - f^\star\|_2^2 \le C \left( \frac{\log^\alpha n}{n} + \frac{\sqrt{\log^\alpha n_{\mathrm{out}}}}{n_{\mathrm{out}}} \right) \cdot [\|f^\star\|_{\mathbf{k}} + 1]^2. \tag{20}$$

*for some constant $C$ that does not depend on $n$ or $n_{\mathrm{out}}$.*

See App. E for the proof. When $n_{\text{out}} = \sqrt{n}$, ST-KRR achieves an excess risk rate of $n^{-1/2} \log^\alpha n$ for $\mathbf{k}$ satisfying LogGrowth$(\alpha, \beta)$. While KT-KRR does not achieve a strictly better excess risk rate bound over ST-KRR, we see that in practice, KT-KRR still obtains an empirical advantage. Obtaining a sharper error rate for the infinite-dimensional kernel setting is an exciting venue for future work.

## 5 Experimental results

We now present experiments on simulated and real-world data. On real-world data, we compare our KT-KRR estimator with several state-of-the-art KRR methods, including Nyström subsampling-based methods and KRR pre-conditioning methods. All our experiments were run on a machine with 8 CPU cores and 100 GB RAM. Our code can be found at https://github.com/ag2435/npr.

### 5.1 Simulation studies

We begin with some simulation experiments. For simplicity, let $\mathcal{X} = \mathbb{R}$ and $\mathbb{P} = \text{Unif}[-\sqrt{3}, \sqrt{3}]$ so that $\text{Var}[X] = 1$. We set

$$f^\star(x) = 8\sin(8\pi x)\exp(x) \quad \text{and} \quad \sigma = 1 \qquad (21)$$

and follow (1) to generate $(y_i)_{i=1}^n$ (see Fig. 2). We let the input sample size $n$ vary between $2^8, 2^{10}, 2^{12}, 2^{14}$ and set the output coreset size to be $n_{\text{out}} = \sqrt{n}$ in all cases. For NW, we use the Wendland(0) kernel defined by

$$\mathbf{k}(x_1, x_2) \triangleq (1 - \tfrac{\|x_1 - x_2\|_2}{h})_+ \quad \text{for} \quad h > 0. \qquad (22)$$

For KRR, we use the Gaussian kernel defined by

Figure 2: **Simulated data.**

$$\mathbf{k}(x_1, x_2) \triangleq \exp(-\tfrac{\|x_1 - x_2\|_2^2}{2h^2}) \quad \text{for} \quad h > 0. \qquad (23)$$

We select the bandwidth $h$ and regularization parameter $\lambda'$ (for KRR) using grid search. Specifically, we use a held-out validation set of size $10^4$ and run each parameter configuration 100 times to estimate the validation MSE since KT-KRR and ST-KRR are random.

**Ablation study.** In Fig. 1, we compare thinning with our proposed meta-kernel $\mathbf{k}_{\text{ALG}} = \mathbf{k}_{\text{NW}}$ to thinning with the baseline meta-kernels (11) and (12). For our particular regression function (21), thinning with (11) outperforms thinning with (12). We hypothesize that the latter kernel is not robust to the scaling of the response variables. By inspecting (22), we see that $\|(x_1, y_1) - (x_2, y_2)\|_2$ is heavily determined by the $y_i$ values when they are large compared to the values of $x_i$—as is the case on the right side of Fig. 2 (when $X > 0$). Since $\mathbb{P}$ is a uniform distribution, thinning with (11) evenly subsamples points along the input domain $\mathcal{X}$, even though accurately learning the left side of Fig. 2 (when $X < 0$) is not needed for effective prediction since it is primarily noise. Validating our theory from Thm. 1, the best performance is obtained when thinning with $\mathbf{k}_{\text{NW}}$ (13), which avoids evenly subsampling points along the input domain and correctly exploits the dependence between $X$ and $Y$.

In Fig. 1, we perform a similar ablation for KRR. Again we observe that thinning with $\mathbf{k}_{\text{ALG}}((x_1, y_1), (x_2, y_2)) = \mathbf{k}(x_1, x_2)$ outperforms thinning with $\mathbf{k}_{\text{ALG}}((x_1, y_1), (x_2, y_2)) = \mathbf{k}((x_1, y_1), (x_2, y_2))$, while thinning with $\mathbf{k}_{\text{ALG}} = \mathbf{k}_{\text{RR}}$ achieves the best performance.

**Comparison with FULL, ST, RPCHOLESKY.** In Fig. 3(a), we compare the MSE of KT-NW to FULL-NW, ST-NW (a.k.a "Subsample"), and RPCHOLESKY-NW across four values of $n$. This last method uses the pivot points from RPCHOLESKY as the output coreset $\mathcal{S}_{\text{out}}$. At all $n$ we evaluated, KT-NW achieves lower MSE than ST-NW and RPCHOLESKY-NW. FULL-NW achieves the lowest MSE across the board, but it suffers from significantly worse run times, especially at test time. Owing to its $\mathcal{O}(n \log^3 n)$ runtime, KT-NW is significantly faster than RPCHOLESKY-NW for training and nearly matches ST-NW in both training and testing time. We hypothesize that RPCHOLESKY—while it provides a good low-rank approximation of the kernel matrix—is not designed to preserve averages.

In Fig. 3(b), we compare the MSE of KT-KRR to FULL-KRR, ST-KRR (a.k.a "Subsample"), and the RPCHOLESKY-KRR method from Chen et al. [6, Sec. 4.2.2], which uses RPCHOLESKY to select

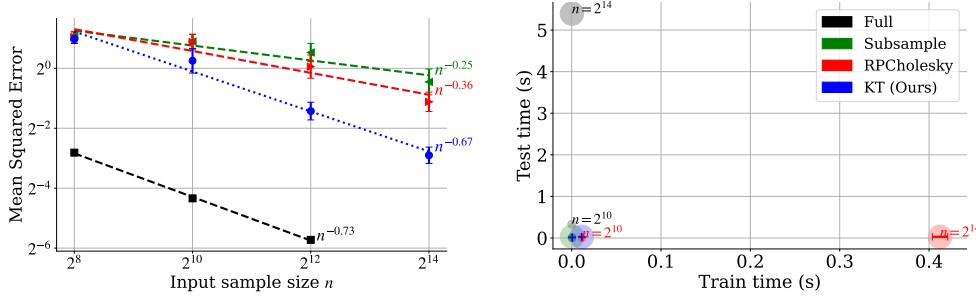

(a) Nadaraya-Watson estimator with Wendland(0) kernel (22).

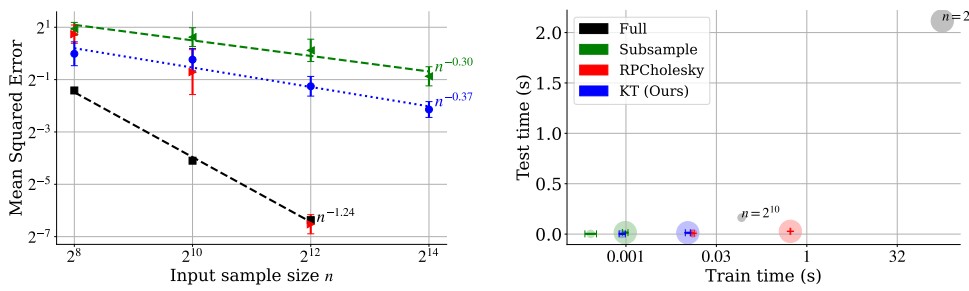

(b) Kernel ridge regression estimator with Gaussian kernel (23).

Figure 3: **MSE and runtime comparison on simulated data.** Each point plots the mean and standard deviation across 100 trials (after parameter grid search).

landmark points for the restricted KRR problem. We observe that KT-KRR achieves lower MSE than ST-KRR, but higher MSE than RPCHOLESKY-KRR and FULL-KRR. In Fig. 3(b), we also observe that KT-KRR is orders of magnitude faster than FULL-KRR across the board, with runtime comparable to ST-KRR and RPCHOLESKY-KRR in both training and testing.

## 5.2 Real data experiments

We now present experiments on real-world data using two popular datasets: the California Housing regression dataset from Pace and Barry [17] (https://scikit-learn.org/1.5/datasets/real_world.html#california-housing-dataset; BSD-3-Clause license) and the SUSY binary classification dataset from Baldi et al. [2] (https://archive.ics.uci.edu/dataset/279/susy; CC-BY-4.0 license).

**California Housing dataset** ($d = 8, N = 2 \times 10^4$). Tab. 1(a) compares the test MSE, train times, and test times. We normalize the input features by subtracting the mean and dividing by the standard deviation and use a 80-20 train-test split. For all methods, we use the Gaussian kernel (23) with bandwidth $h = 10$. We use $\lambda = \lambda' = 10^{-3}$ for FULL-KRR, ST-KRR, and KT-KRR and $\lambda = 10^{-5}$ for RPCHOLESKY-KRR. On this dataset, KT-KRR lies between ST-KRR and RPCHOLESKY-KRR in terms of test MSE. When $n_{\text{out}} = \sqrt{n}$, RPCHOLESKY pivot selection takes $\mathcal{O}(n^2)$ time by Chen et al. [6, Alg. 2], compared to KT-COMPRESS++, which compresses the input points in only $\mathcal{O}(n \log^3 n)$ time. This difference in big-O runtime is reflected in our empirical results, where we see KT-KRR take 0.0153s versus 0.3237s for RPCHOLESKY-KRR.

**SUSY dataset** ($d = 18, N = 5 \times 10^6$). Tab. 1(b) compares our proposed method KT-KRR (with $h = 10, \lambda' = 10^{-1}$) to several large-scale kernel methods, namely RPCholesky preconditioning [7], FALKON [22], and Conjugate Gradient (all with $h = 10, \lambda = 10^{-3}$) in terms of test classification error and training times. For the baseline methods, we use the Matlab implementation provided by Díaz et al. [7] (https://github.com/eepperly/Robust-randomized-preconditioning-for-kernel-ridge-regression). In

| Method | MSE (%) | Training time (s) | Prediction time (s) |
|---|---|---|---|
| Full | 0.4137 | 11.1095 | 0.7024 |
| ST-KRR | $0.5736 \pm 0.0018$ | $0.0018 \pm 0.0005$ | $0.0092 \pm 0.0006$ |
| RPCHOLESKY | $0.3503 \pm 0.0001$ | $0.3237 \pm 0.0094$ | $0.0060 \pm 0.0008$ |
| KT-KRR (Ours) | $0.5580 \pm 0.0015$ | $0.0153 \pm 0.0013$ | $0.0083 \pm 0.0003$ |

(a) California Housing regression dataset.

| Method | Test Error (%) | Training Time (s) |
|---|---|---|
| RPCholesky | $19.99 \pm 0.00$ | $3.46 \pm 0.03$ |
| FALKON | $19.99 \pm 0.00$ | $5.06 \pm 0.02$ |
| CG | $20.35 \pm 0.00$ | $6.16 \pm 0.03$ |
| ST-KRR | $22.71 \pm 0.30$ | $0.09 \pm 0.00$ |
| KT-KRR (Ours) | $22.00 \pm 0.21$ | $1.79 \pm 0.00$ |

(b) SUSY dataset.

Table 1: **Accuracy and runtime comparison on real-world data.** Each cell represents mean ± standard error across 100 trials.

our experiment, we use $4 \times 10^6$ points for training and the remaining $10^6$ points for testing. As is common practice for classification tasks, we use the Laplace kernel defined by $\mathbf{k}(x_1, x_2) \triangleq \exp(-\|x_1 - x_2\|_2 / h)$. All parameters are chosen with cross-validation.

We observe that KT-KRR achieves test MSE between ST-KRR and RPCHOLESKY preconditioning with training time almost half that of RPCHOLESKY preconditioning. Notably, our Cython implementation of KT-COMPRESS++ thinned the four million training samples in only 1.7 seconds on a single CPU core—with further speed-ups to be gained from parallelizing on a GPU in the future.

# 6 Conclusions

In this work, we introduce a meta-algorithm for speeding up two estimators from non-parametric regression, namely the Nadaraya-Watson and Kernel Ridge Regression estimators. Our method inherits the favorable computational efficiency of the underlying Kernel Thinning algorithm and stands to benefit from further advancements in unsupervised learning compression methods.

The KT guarantees provided in this work apply only when $f^\star \in \mathcal{H}(\mathbf{k})$ for some base kernel $\mathbf{k}$. In practice, choosing a good kernel $\mathbf{k}$ is indeed a challenge common to all prior work. Our framework is friendly to recent developments in kernel selection to handle this problem: Dwivedi and Mackey [9, Cor. 1] provide integration-error guarantees for KT when $f^\star \notin \mathcal{H}(\mathbf{k})$. Moreover, there are recent results on finding the best kernel (e.g., for hypothesis testing [8, Sec. 4.2]). Radhakrishnan et al. [20] introduce the Recursive Feature Machine, which uses a parameterized kernel $\mathbf{k}_M(x_1, x_2) \triangleq \exp(-(x_1 - x_2)^\top M(x_1 - x_2)/(2h^2))$, and propose an efficient method to learn the matrix parameter $M$ via the average gradient outer product estimator. An exciting future direction would be to combine these parameterized (or "learned") kernels with our proposed KT methods for non-parametric regression.

# 7 Acknowledgements

AG is supported with funding from the NewYork-Presbyterian Hospital.

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

## A  Background on KT-COMPRESS++

This section details the KT-COMPRESS++ algorithm of Shetty et al. [23, Ex. 6]. In a nutshell, KT-COMPRESS (Alg. 2) takes as input a point sequence of size $n$, a compression level $\mathfrak{g}$, a (reproducing) kernel functions $\mathbf{k}_{\mathrm{ALG}}$, and a failure probability $\delta$. KT-COMPRESS++ first runs the KT-COMPRESS($\mathfrak{g}$) algorithm of Shetty et al. [23, Ex. 4] to produce an intermediate coreset of size $2^{\mathfrak{g}}\sqrt{n}$. Next, the KT algorithm is run on the intermediate coreset to produce a final output of size $\sqrt{n}$.

KT-COMPRESS proceeds by calling the recursive procedure COMPRESS, which uses KT with kernels $\mathbf{k}_{\mathrm{ALG}}$ as an intermediate halving algorithm. The KT algorithm itself consists of two subroutines: (1) KT-SPLIT (Alg. 3a), which splits a given input point sequence into two equal halves with small approximation error in the $\mathbf{k}_{\mathrm{ALG}}$ reproducing kernel Hilbert space and (2) KT-SWAP (Alg. 3b), which selects the best approximation amongst the KT-SPLIT coresets and a baseline coreset (that simply selects every other point in the sequence) and then iteratively refines the selected coreset by swapping out each element in turn for the non-coreset point that most improves $\mathrm{MMD}_{\mathbf{k}_{\mathrm{ALG}}}$ error. As in Shetty et al. [23, Rem. 3], we symmetrize the output of KT by returning either the KT coreset or its complement with equal probability.

Following Shetty et al. [23, Ex. 6], we always default to $\mathfrak{g} = \lceil \log_2 \log n + 3.1 \rceil$ so that KT-COMPRESS++ has an overall runtime of $\mathcal{O}(n \log^3 n)$. For the sake of simplicity, we drop any dependence on $\mathfrak{g}$ in the main paper.

---

**Algorithm 1:** KT-COMPRESS++ – Identify coreset of size $\sqrt{n}$

**Input:** point sequence $\mathcal{S}_{\mathrm{in}}$ of size $n$, compression level $\mathfrak{g}$, kernel $\mathbf{k}_{\mathrm{ALG}}$, failure probability $\delta$

$\mathcal{S}_{\mathrm{C}} \quad \leftarrow \quad$ KT-COMPRESS($\mathfrak{g}, \mathcal{S}_{\mathrm{in}}, \delta$)      // coreset of size $2^{\mathfrak{g}}\sqrt{n}$

$\mathcal{S}_{\mathrm{C}++} \leftarrow \quad$ KT($\mathcal{S}_{\mathrm{C}}, \mathbf{k}_{\mathrm{ALG}}, \frac{\mathfrak{g}}{\mathfrak{g}+2^{\mathfrak{g}}(\beta_n+1)}\delta$)      // coreset of size $\sqrt{n}$

**return** $\mathcal{S}_{\mathrm{C}++}$

---

**Algorithm 2:** KT-COMPRESS – Identify coreset of size $2^{\mathfrak{g}}\sqrt{n}$

**Input:** point sequence $\mathcal{S}_{\mathrm{in}}$ of size $n$, compression level $\mathfrak{g}$, kernel $\mathbf{k}_{\mathrm{ALG}}$, failure probability $\delta$

**return** COMPRESS($\mathcal{S}_{\mathrm{in}}, \mathfrak{g}, \mathbf{k}_{\mathrm{ALG}}, \frac{\delta}{n4^{\mathfrak{g}+1}(\log_4 n - \mathfrak{g})}$)

---

**function** COMPRESS($\mathcal{S}, \mathfrak{g}, \mathbf{k}_{\mathrm{ALG}}, \delta$):

    **if** $|\mathcal{S}| = 4^{\mathfrak{g}}$ **then return** $\mathcal{S}$

    Partition $\mathcal{S}$ into four arbitrary subsequences $\{\mathcal{S}_i\}_{i=1}^4$ each of size $n/4$

    **for** $i = 1, 2, 3, 4$ **do**

        $\widetilde{\mathcal{S}}_i \leftarrow$ COMPRESS($\mathcal{S}_i, \mathfrak{g}, \mathbf{k}_{\mathrm{ALG}}, \delta$) // run COMPRESS recursively to return coresets of size $2^{\mathfrak{g}} \cdot \sqrt{\frac{|\mathcal{S}|}{4}}$

    **end**

    $\widetilde{\mathcal{S}} \leftarrow$ CONCATENATE($\widetilde{\mathcal{S}}_1, \widetilde{\mathcal{S}}_2, \widetilde{\mathcal{S}}_3, \widetilde{\mathcal{S}}_4$)      // combine the coresets to obtain a coreset of size $2 \cdot 2^{\mathfrak{g}} \cdot \sqrt{|\mathcal{S}|}$

**return** KT($\widetilde{\mathcal{S}}, \mathbf{k}_{\mathrm{ALG}}, |\widetilde{\mathcal{S}}|^2 \delta$)      // halve the coreset to size $2^{\mathfrak{g}}\sqrt{|\mathcal{S}|}$ via symmetrized kernel thinning

---

**function** KT($\mathcal{S}, \mathbf{k}_{\mathrm{ALG}}, \delta$):

    // Identify kernel thinning coreset containing $\lfloor |\mathcal{S}|/2 \rfloor$ input points

    $\mathcal{S}_{\mathrm{KT}} \leftarrow$ KT-SWAP($\mathbf{k}_{\mathrm{ALG}}$, KT-SPLIT($\mathbf{k}_{\mathrm{ALG}}, \mathcal{S}, \delta$))

**return** $\mathcal{S}_{\mathrm{KT}}$ with probability $\frac{1}{2}$ and the complementary coreset $\mathcal{S} \setminus \mathcal{S}_{\mathrm{KT}}$ otherwise

---

Define the event

$$\mathcal{E}_{\mathrm{KT},\delta} \triangleq \{\text{KT-COMPRESS++ succeeds}\}. \tag{24}$$

Dwivedi and Mackey [10, Thm. 1, Rmk. 4] show that

$$\mathbb{P}(\mathcal{E}_{\mathrm{KT},\delta}) \geq 1 - \delta.$$

We restate [10, Thm. 4] in our notation:

**Lemma 1** ($L^\infty$ *guarantee for* KT-COMPRESS++). *Let* $\mathcal{Z} \subset \mathbb{R}^d$ *and consider a reproducing kernel* $\mathbf{k}_{\mathrm{ALG}} : \mathcal{Z} \times \mathcal{Z} \to \mathbb{R}$. *Assume* $n/n_{\mathrm{out}} \in 2^{\mathbb{N}}$. *Let* $\mathcal{S}_{\mathrm{KT}} \triangleq$ KT-COMPRESS++($\mathcal{S}_{\mathrm{in}}, \mathfrak{g}, \mathbf{k}_{\mathrm{ALG}}, \delta$) *and*

---

**Algorithm 3a:** KT-SPLIT $-$ Divide points into candidate coresets of size $\lfloor n/2 \rfloor$

---

**Input:** kernel $\mathbf{k}_{\text{split}}$, point sequence $\mathcal{S}_{\text{in}} = (x_i)_{i=1}^n$, failure probability $\delta$

$\mathcal{S}^{(1)}, \mathcal{S}^{(2)} \leftarrow \{\}$    // Initialize empty coresets: $\mathcal{S}^{(1)}, \mathcal{S}^{(2)}$ have size $i$ after round $i$
$\sigma \leftarrow 0$                // Initialize swapping parameter
**for** $i = 1, 2, \ldots, \lfloor n/2 \rfloor$ **do**

> // Consider two points at a time
> $(x, x') \leftarrow (x_{2i-1}, x_{2i})$
>
> // Compute swapping threshold $\mathfrak{a}_i$
> $\mathfrak{a}_i, \sigma \leftarrow$ get_swap_params$(\sigma, \mathfrak{b}, \frac{\delta}{n})$ with $\mathfrak{b}^2 = \mathbf{k}_{\text{split}}(x,x) + \mathbf{k}_{\text{split}}(x',x') - 2\mathbf{k}_{\text{split}}(x,x')$
>
> // Assign one point to each coreset after probabilistic swapping
> $\theta \leftarrow \sum_{j=1}^{2i-2}(\mathbf{k}_{\text{split}}(x_j, x) - \mathbf{k}_{\text{split}}(x_j, x')) - 2\sum_{z \in \mathcal{S}^{(1)}}(\mathbf{k}_{\text{split}}(z, x) - \mathbf{k}_{\text{split}}(z, x'))$
> $(x, x') \leftarrow (x', x)$ *with probability* $\min(1, \frac{1}{2}(1 - \frac{\theta}{\mathfrak{a}_i})_+)$
> $\mathcal{S}^{(1)}$.append$(x)$;   $\mathcal{S}^{(2)}$.append$(x')$

**end**
**return** $(\mathcal{S}^{(1)}, \mathcal{S}^{(2)})$, candidate coresets of size $\lfloor n/2 \rfloor$

---

**function** get_swap_params$(\sigma, \mathfrak{b}, \delta)$:

> $\mathfrak{a}_i \leftarrow \max(\mathfrak{b}\sigma\sqrt{2\log(2/\delta)}, \mathfrak{b}^2)$
> $\sigma^2 \leftarrow \sigma^2 + \mathfrak{b}^2(1 + (\mathfrak{b}^2 - 2\mathfrak{a}_i)\sigma^2/\mathfrak{a}_i^2)_+$

**return** $(\mathfrak{a}_i, \sigma)$

---

---

**Algorithm 3b:** KT-SWAP $-$ Identify and refine the best candidate coreset

---

**Input:** kernel $\mathbf{k}_{\text{ALG}}$, point sequence $\mathcal{S}_{\text{in}} = (x_i)_{i=1}^n$, candidate coresets $(\mathcal{S}^{(1)}, \mathcal{S}^{(2)})$

$\mathcal{S}^{(0)} \leftarrow$ baseline_coreset$(\mathcal{S}_{\text{in}}, \texttt{size} = \lfloor n/2 \rfloor)$          // Compare to baseline (e.g., standard thinning)

$\mathcal{S}_{\text{KT}} \leftarrow \mathcal{S}^{(\ell^\star)}$ for $\ell^\star \leftarrow \operatorname{argmin}_{\ell \in \{0,1,2\}} \text{MMD}_{\mathbf{k}_{\text{ALG}}}(\mathcal{S}_{\text{in}}, \mathcal{S}^{(\ell)})$  // Select best coreset
// Swap out each point in $\mathcal{S}_{\text{KT}}$ for best alternative in $\mathcal{S}_{\text{in}}$ while ensuring no point is repeated in $\mathcal{S}_{\text{KT}}$
**for** $i = 1, \ldots, \lfloor n/2 \rfloor$ **do**

> $\mathcal{S}_{\text{KT}}[i] \leftarrow \operatorname{argmin}_{z \in \{\mathcal{S}_{\text{KT}}[i]\} \cup (\mathcal{S}_{\text{in}} \setminus \mathcal{S}_{\text{KT}})} \text{MMD}_{\mathbf{k}_{\text{ALG}}}(\mathcal{S}_{\text{in}}, \mathcal{S}_{\text{KT}} \text{ with } \mathcal{S}_{\text{KT}}[i] = z)$

**end**
**return** $\mathcal{S}_{\text{KT}}$, refined coreset of size $\lfloor n/2 \rfloor$

---

*define* $\mathbb{P}_{\text{in}} \triangleq \frac{1}{n}\sum_{z \in \mathcal{S}_{\text{in}}} \delta_z$ *and* $\mathbb{Q}_{\text{out}} \triangleq \frac{1}{n_{\text{out}}}\sum_{z \in \mathcal{S}_{\text{KT}}} \delta_z$. *Then on event* $\mathcal{E}_{KT,\delta}$, *the following bound holds:*

$$\|(\mathbb{P}_{\text{in}} - \mathbb{Q}_{\text{out}})\mathbf{k}_{\text{ALG}}\|_\infty \leq c\frac{\|\mathbf{k}_{\text{ALG}}\|_{\infty,\text{in}}}{n_{\text{out}}}\mathfrak{M}_{\mathbf{k}_{\text{ALG}}}(n, n_{\text{out}}, d, \delta, R), \quad where \tag{25}$$

$$\mathfrak{M}_{\mathbf{k}_{\text{ALG}}}(n, n_{\text{out}}, d, \delta, R) \triangleq \sqrt{\log\left(\frac{n_{\text{out}}\log_2(n/n_{\text{out}})}{\delta}\right)} \times \tag{26}$$

$$\left[\sqrt{\log\left(\frac{1}{\delta}\right)} + \sqrt{d\log\left(1 + \frac{L_{\mathbf{k}_{\text{ALG}}}}{\|\mathbf{k}_{\text{ALG}}\|_\infty}(R_{\mathbf{k}_{\text{ALG}},n} + R)\right)}\right],$$

$$L_{\mathbf{k}_{\text{ALG}}} \triangleq \sup_{z_1, z_2, z_3 \in \mathcal{Z}} \frac{|\mathbf{k}_{\text{ALG}}(z_1, z_2) - \mathbf{k}_{\text{ALG}}(z_1, z_3)|}{\|z_2 - z_3\|_2}, \quad and \tag{27}$$

$$R_{\mathbf{k}_{\text{ALG}},n} \triangleq \inf\left\{r : \sup_{\substack{z_1, z_2 \in \mathcal{Z} \\ \|z_1 - z_2\|_2 \geq r}} |\mathbf{k}_{\text{ALG}}(z_1, z_2)| \leq \frac{\|\mathbf{k}_{\text{ALG}}\|_\infty}{n}\right\}, \tag{28}$$

*for some universal positive constant c.*

*Proof.* The claim follows by replacing $\mathbf{k}$ [10, Thm. 4] with $\mathbf{k}_{\text{ALG}}$, replacing the sub-Gaussian constant of KT with that of KT-COMPRESS++ in [23, Ex. 5], and replacing $\|\mathbf{k}_{\text{ALG}}\|_\infty$ with $\|\mathbf{k}_{\text{ALG}}\|_{\infty,\text{in}} \triangleq \sup_{z \in \mathcal{S}_{\text{in}}} \mathbf{k}_{\text{ALG}}(z, z)$ throughout.                                               $\square$

We restate [9, Thm. 2] in our notation:

**Lemma 2** (MMD guarantee for KT-COMPRESS++). *Let $\mathcal{Z} \subset \mathbb{R}^d$ and consider a reproducing kernel $\mathbf{k}_{\mathrm{ALG}} : \mathcal{Z} \times \mathcal{Z} \to \mathbb{R}$. Assume $n/n_{\mathrm{out}} \in 2^{\mathbb{N}}$. Let $\mathcal{S}_{\mathrm{KT}} \triangleq \text{KT-COMPRESS++}(\mathbf{k}_{\mathrm{ALG}}, \mathfrak{g})(\mathcal{S}_{\mathrm{in}})$ and define $\mathbb{P}_{\mathrm{in}} \triangleq \frac{1}{n} \sum_{z \in \mathcal{S}_{\mathrm{in}}} \delta_z$ and $\mathbb{Q}_{\mathrm{out}} \triangleq \frac{1}{n_{\mathrm{out}}} \sum_{z \in \mathcal{S}_{\mathrm{KT}}} \delta_z$. Then on event $\mathcal{E}_{KT,\delta}$, the following bound holds:*

$$\sup_{\substack{h \in \mathcal{H}(\mathbf{k}_{\mathrm{ALG}}): \\ \|h\|_{\mathbf{k}_{\mathrm{ALG}}} \leq 1}} |(\mathbb{P}_{\mathrm{in}} - \mathbb{Q}_{\mathrm{out}})h| \leq \inf_{\substack{\epsilon \in (0,1) \\ \mathcal{S}_{\mathrm{in}} \subset \mathcal{A}}} \left\{ 2\epsilon + \frac{2\|\mathbf{k}_{\mathrm{ALG}}\|_{\infty,\mathrm{in}}^{1/2}}{n_{\mathrm{out}}} \mathfrak{W}_{\mathbf{k}_{\mathrm{ALG}}}(n, n_{\mathrm{out}}, \delta, \mathcal{A}, \epsilon) \right\} \quad \text{where}$$

$$\mathfrak{W}_{\mathbf{k}_{\mathrm{ALG}}}(n, n_{\mathrm{out}}, \delta, R, \epsilon) \triangleq c \sqrt{\log\left(\frac{n_{\mathrm{out}} \log(n/n_{\mathrm{out}})}{\delta}\right) \cdot \left[\log\left(\frac{1}{\delta}\right) + \log \mathcal{N}_{\mathbf{k}_{\mathrm{ALG}}}(\mathcal{A}, \epsilon)\right]}. \tag{29}$$

*for some universal positive constant c.*

*Proof.* The claim follows from replacing $\mathbf{k}$ in [9, Thm. 2] with $\mathbf{k}_{\mathrm{ALG}}$ and replacing the sub-Gaussian constant of KT with that of KT-COMPRESS++ in [23, Ex. 5]. $\qquad\square$

# B   Proof of Thm. 1: KT-NW

Our primary goal is to bound $\mathbb{E}_{\mathcal{S}_{\mathrm{in}}}[(\widehat{f}_{\mathrm{KT}}(x_0) - f^\star(x_0))^2]$ for a fixed $x_0 \in \mathcal{X}$. Once we have this bound, bounding $\|\widehat{f}_{\mathrm{KT}} - f^\star\|_2^2$ is as straightforward as integrating over $x_0 \in \mathcal{X}$.

Consider the following decomposition:

$$\mathbb{E}_{\mathcal{S}_{\mathrm{in}}}\left[\left(\widehat{f}_{\mathrm{KT}}(x_0) - f^\star(x_0)\right)^2\right] = \mathbb{E}_{\mathcal{S}_{\mathrm{in}}}\left[\left(\widehat{f}_{\mathrm{KT}}(x_0) - \widehat{f}(x_0) + \widehat{f}(x_0) - f^\star(x_0)\right)^2\right]$$

$$\leq 2\,\mathbb{E}_{\mathcal{S}_{\mathrm{in}}}\left[\left(\widehat{f}_{\mathrm{KT}}(x_0) - \widehat{f}(x_0)\right)^2\right] \tag{30}$$

$$+ 2\,\mathbb{E}_{\mathcal{S}_{\mathrm{in}}}\left[\left(\widehat{f}(x_0) - f^\star(x_0)\right)^2\right]. \tag{31}$$

Define the random variables

$$\eta_i \triangleq \mathbf{1}\left\{\frac{\|X_i - x_0\|}{h} \leq 1\right\} \quad \text{for} \quad i = 1, 2, \ldots, n.$$

Also define the event

$$\mathcal{E} \triangleq \left\{\sum_{i=1}^n \eta_i > 0\right\}. \tag{32}$$

Since $X_i$ are i.i.d. samples from $\mathbb{P}$, it follows that $\eta_i$ are i.i.d. Bernoulli random variables with parameter

$$\overline{p} \triangleq \mathbb{P}(\eta_i = 1) \geq c_0 p_{\min} h^d, \tag{33}$$

where $c_0 > 0$ depends only on $d$ and $\kappa$ (see Assum. 2). Denote the denominator terms in $\widehat{f}$ and $\widehat{f}_{\mathrm{KT}}$ by

$$\widehat{p}(\cdot) \triangleq \frac{1}{n} \sum_{i=1}^n \mathbf{k}(\cdot, x_i) \quad \text{and} \quad \widehat{p}_{\mathrm{KT}}(\cdot) \triangleq \frac{1}{n_{\mathrm{out}}} \sum_{j=1}^{n_{\mathrm{out}}} \mathbf{k}(\cdot, x_i'), \tag{34}$$

respectively, and the numerator terms in $\widehat{f}$ and $\widehat{f}_{\mathrm{KT}}$ by

$$\widehat{A}(\cdot) \triangleq \frac{1}{n} \sum_{i=1}^n \mathbf{k}(\cdot, x_i) y_i \quad \text{and} \quad \widehat{A}_{\mathrm{KT}}(\cdot) \triangleq \frac{1}{n_{\mathrm{out}}} \sum_{j=1}^{n_{\mathrm{out}}} \mathbf{k}(\cdot, x_i') y_i', \tag{35}$$

respectively.

We now consider two cases depending on the event $\mathcal{E}$.

*Case I:* Suppose event $\mathcal{E}^c$ is satisfied. It follows from (34) that $\widehat{p}(x_0) = 0$, in which case $\widehat{f}(x_0) = 0$. Since $\mathcal{S}_{\mathrm{out}} \subset \mathcal{S}_{\mathrm{in}}$, it necessarily follows that $\widehat{p}_{\mathrm{KT}}(x_0) = 0$ and $\widehat{f}_{\mathrm{KT}}(x_0) = 0$. Thus, we can bound (30) and (31) by

$$\mathbb{E}_{\mathcal{S}_{\mathrm{in}}}\left[\left(\widehat{f}_{\mathrm{KT}}(x_0) - \widehat{f}(x_0)\right)^2 \mathbb{I}[\mathcal{E}^c]\right] = 0 \quad \text{and}$$

$$\mathbb{E}_{\mathcal{S}_{\mathrm{in}}}\left[\left(\widehat{f}(x_0) - f^\star(x_0)\right)^2 \mathbb{I}[\mathcal{E}^c]\right] = \mathbb{E}_{\mathcal{S}_{\mathrm{in}}}\left[\left(0 - f^\star(x_0)\right)^2 \mathbb{I}[\mathcal{E}^c]\right]$$

$$\leq (f^\star)^2(x_0)\mathbb{P}(\mathcal{E}^c)$$

$$\leq (f^\star)^2(x_0)(1 - \overline{p})^n$$

$$\leq (f^\star)^2(x_0)\exp\{-Cnh^d\}$$

for some positive constant $C$ that does not depend on $n$. Note that these are low-order terms compared to the rest of the calculations, so we may ignore them in the final bound.

*Case II:* Otherwise, we may assume event $\mathcal{E}$ is satisfied. Let us first bound (30). On event $\mathcal{E}_{\mathrm{KT},\delta}$ (24), we claim that

$$\mathbb{E}_{\mathcal{S}_{\mathrm{in}}}\left[\left(\widehat{f}_{\mathrm{KT}}(x_0) - \widehat{f}(x_0)\right)^2\mathbb{I}[\mathcal{E}]\right] \leq \frac{Cd\log^2 n}{nh^{2d}} \quad \text{whenever} \quad \overline{p} = \omega\left(\sqrt{\tfrac{d}{n}}\log n\right). \tag{36}$$

We defer the proof to App. B.1.

Letting $X \triangleq (X_1, \ldots, X_n)$ and $Y \triangleq (Y_1, \ldots, Y_n)$ denote the $x$ and $y$ components of $\mathcal{S}_{\mathrm{in}}$, respectively, we can further decompose (31) by

$$\mathbb{E}_{\mathcal{S}_{\mathrm{in}}}\left[\left(\widehat{f}(x_0) - f^\star(x_0)\right)^2 \mathbb{I}[\mathcal{E}]\right] = \mathbb{E}_X\left[\mathbb{E}_{Y|X}\left[\left(\widehat{f}(x_0) - \mathbb{E}_{Y|X}\left[\widehat{f}(x_0)\right]\right)^2\right]\mathbb{I}[\mathcal{E}]\right]$$

$$+ \mathbb{E}_X\left[\left(\mathbb{E}_{Y|X}\left[\widehat{f}(x_0)\right] - f^\star(x_0)\right)^2\mathbb{I}[\mathcal{E}]\right],$$

where the first RHS term corresponds to the variance and the second RHS term corresponds to the bias. We claim that

$$\mathbb{E}_X\left[\mathbb{E}_{Y|X}\left[\left(\widehat{f}(x_0) - \mathbb{E}_{Y|X}\left[\widehat{f}(x_0)\right]\right)^2\right]\mathbb{I}[\mathcal{E}]\right] \leq \sigma_\xi^2\left(n\exp\{-Cnh^d\} + \tfrac{C}{nh^d}\right) \quad \text{and} \tag{37}$$

$$\mathbb{E}_X\left[\left(\mathbb{E}_{Y|X}\left[\widehat{f}(x_0)\right] - f^\star(x_0)\right)^2\mathbb{I}[\mathcal{E}]\right] \leq C \cdot L_f^2 h^{2\beta}\log^2 n, \tag{38}$$

for some constant $C > 0$ that does not depend on either $n$ or $h$. We defer the proofs to App. B.2 and B.3. Combining (36) to (38), we have

$$\mathbb{E}_{\mathcal{S}_{\mathrm{in}}}\left[\left(\widehat{f}_{\mathrm{KT}}(x_0) - f^\star(x_0)\right)^2\mathbb{I}[\mathcal{E}]\right] \leq \underbrace{\frac{Cd\log^2 n}{nh^{2d}}}_{\text{KT bound}} + \underbrace{2\sigma_\xi^2\left(ne^{-Cnh^d} + \tfrac{C}{nh^d}\right)}_{\text{Variance bound}} + \underbrace{2CL_f^2 h^{2\beta}\log^2 n}_{\text{Bias bound}}. \tag{39}$$

Note that $h^d \leq 1$, so the $\frac{Cd\log^2 n}{nh^{2d}}$ term dominates the $\frac{C}{nh^d}$ term. Thus, the optimal choice of bandwidth $h$ comes from balancing

$$\frac{C}{nh^{2d}} \sim 2L_f^2 h^{2\beta} \implies h = cn^{-\frac{1}{2\beta+2d}}. \tag{40}$$

Finally, we must verify our growth rate assumption on $\overline{p}$ in (36) is satisfied. Since $\beta > 0$, we have

$$\overline{p} \overset{(33)}{\geq} c_0 p_{\min}h^d \overset{(40)}{=} c_0' n^{-\frac{d}{2\beta+2d}} \implies \lim_{n\to\infty}\frac{\overline{p}}{\sqrt{\tfrac{d}{n}}\log n} = \infty.$$

Plugging (40) into (39) yields the advertised bound (18).

## B.1 Proof of claim (36)

We first provide a generic result for approximating the numerator and denominator terms defined in (34) and (35).

**Lemma 3** (Simultaneous $L^\infty$ bound using KT-COMPRESS++ with $\mathbf{k}_{\mathrm{NW}}$). *Suppose* $\mathbf{k}$ *satisfies Assum. 2. Given $\mathcal{S}_{\mathrm{in}}$, the following bounds hold on the event $\mathcal{E}_{\mathrm{KT},\delta}$:*

$$\|\widehat{p} - \widehat{p}_{\mathrm{KT}}\|_\infty \leq c_p\sqrt{\tfrac{d}{n}}(\log n + \log(1/\delta)) \tag{41}$$

$$\|\widehat{A} - \widehat{A}_{\mathrm{KT}}\|_\infty \leq c_p\sqrt{\tfrac{d}{n}}(\log n + \log(1/\delta)), \tag{42}$$

*where $c_a, c_p > 0$ are constants that do not depend on $d$ or $n$.*

See App. B.1.1 for the proof. In the sequel, we will simply treat the $\log(1/\delta)$ term as a constant, meaning the $\log n$ terms dominate in the expressions.

With this lemma in hand, let us prove the claim (36). Define the following events:

$$\mathcal{A} \triangleq \{\widehat{p}_{\mathrm{KT}}(x_0) = 0\} \qquad \mathcal{B} \triangleq \{\widehat{p}_{\mathrm{KT}}(x_0) \neq 0\} \qquad \mathcal{C} \triangleq \left\{\widehat{p}(x_0) \geq \tfrac{\overline{p}}{2}\right\}.$$

On event $\mathcal{E}$, consider the following decomposition:

$$\mathbb{E}_{\mathcal{S}_{\mathrm{in}}}\left[\left(\widehat{f}_{\mathrm{KT}}(x_0) - \widehat{f}(x_0)\right)^2 \mathbb{I}[\mathcal{E}_{\mathrm{KT},\delta}]\right] = \mathbb{E}_{\mathcal{S}_{\mathrm{in}}}\left[\left(\widehat{f}_{\mathrm{KT}}(x_0) - \widehat{f}(x_0)\right)^2 \mathbb{I}[\mathcal{E}_{\mathrm{KT},\delta} \cap \mathcal{C}^c]\right] \tag{43}$$

$$+ \mathbb{E}_{\mathcal{S}_{\mathrm{in}}}\left[\left(\widehat{f}_{\mathrm{KT}}(x_0) - \widehat{f}(x_0)\right)^2 \mathbb{I}[\mathcal{E}_{\mathrm{KT},\delta} \cap \mathcal{A} \cap \mathcal{C}]\right] \tag{44}$$

$$+ \mathbb{E}_{\mathcal{S}_{\mathrm{in}}}\left[\left(\widehat{f}_{\mathrm{KT}}(x_0) - \widehat{f}(x_0)\right)^2 \mathbb{I}[\mathcal{E}_{\mathrm{KT},\delta} \cap \mathcal{B} \cap \mathcal{C}]\right]. \tag{45}$$

**Bounding (43).** Note that almost surely, we have

$$|\widehat{f}(x_0)| \leq Y_{\max} \quad \text{and} \quad |\widehat{f}_{\mathrm{KT}}(x_0)| \leq Y_{\max}.$$

Thus, we have

$$\mathbb{E}_{\mathcal{S}_{\mathrm{in}}}\left[\left(\widehat{f}_{\mathrm{KT}}(x_0) - \widehat{f}(x_0)\right)^2 \mathbb{I}[\mathcal{C}^c]\right] \leq 4Y_{\max}^2 \, \mathbb{P}\left(n\,\widehat{p}(x_0) < \tfrac{n\overline{p}}{2}\right)$$

$$\stackrel{(i)}{=} 4Y_{\max}^2 \mathbb{P}\left(\textstyle\sum_{i=1}^{n} \eta_i - n\overline{p} < \tfrac{n\overline{p}}{2} - n\overline{p}\right)$$

$$\stackrel{(ii)}{\leq} c_0 \exp\{-c_1 n h^d\},$$

where (i) follows from subtracting $n\overline{p}$ from both sides of the probability statement and (ii) follows from concentration of Bernoulli random variables (see App. B.2).

**Bounding (44).** Note that on event $\mathcal{E}_{\mathrm{KT},\delta} \cap \mathcal{C}$, we have

$$\widehat{p}_{\mathrm{KT}}(x_0) \geq \widehat{p}(x_0) - \|\widehat{p} - \widehat{p}_{\mathrm{KT}}\|_\infty$$

$$\stackrel{(i)}{\geq} \tfrac{\overline{p}}{2} - c_p \sqrt{\tfrac{d}{n}} \log n$$

$$\stackrel{(36)}{\geq} c_1 \overline{p} \stackrel{(33)}{\geq} c_2 p_{\min} h^d > 0. \tag{46}$$

where step (i) follows from applying (41) and substituting $\widehat{p} \geq \tfrac{\overline{p}}{2}$. Hence the events $\mathcal{E}_{\mathrm{KT},\delta}$ and $\mathcal{A} \cap \mathcal{C}$ are mutually exclusive with probability 1, thereby yielding

$$\mathbb{E}_{\mathcal{S}_{\mathrm{in}}}\left[\left(\widehat{f}_{\mathrm{KT}}(x_0) - \widehat{f}(x_0)\right)^2 \mathbb{I}[\mathcal{E}_{\mathrm{KT},\delta} \cap \mathcal{A} \cap \mathcal{C}]\right] = 0.$$

**Bounding (45).** On the event $\mathcal{B} \cap \mathcal{C}$, we have $\widehat{f}_{\mathrm{KT}}(x_0) = \frac{\widehat{A}_{\mathrm{KT}}(x_0)}{\widehat{p}_{\mathrm{KT}}(x_0)}$ and $\widehat{f}(x_0) = \frac{\widehat{A}(x_0)}{\widehat{p}(x_0)}$, which yields

$$\widehat{f}_{\mathrm{KT}}(x_0) - \widehat{f}(x_0) = \frac{\widehat{A}_{\mathrm{KT}}}{\widehat{p}_{\mathrm{KT}}} - \frac{\widehat{A}(x_0)}{\widehat{p}(x_0)} = \frac{\widehat{A}_{\mathrm{KT}}(x_0) \cdot \widehat{p}(x_0) - \widehat{A}(x_0) \cdot \widehat{p}_{\mathrm{KT}}(x_0)}{\widehat{p}(x_0) \cdot \widehat{p}_{\mathrm{KT}}(x_0)}$$

$$= \frac{(\widehat{A}_{\mathrm{KT}}(x_0) - \widehat{A}(x_0)) \cdot \widehat{p}(x_0) + \widehat{A}(x_0) \cdot (\widehat{p}(x_0) - \widehat{p}_{\mathrm{KT}}(x_0))}{\widehat{p}(x_0) \cdot \widehat{p}_{\mathrm{KT}}(x_0)}$$

$$\leq \frac{\left|\widehat{A}_{\mathrm{KT}}(x_0) - \widehat{A}(x_0)\right| \cdot \widehat{p}(x_0) + \widehat{A}(x_0) \cdot \left|\widehat{p}(x_0) - \widehat{p}_{\mathrm{KT}}(x_0)\right|}{\widehat{p}(x_0) \cdot \widehat{p}_{\mathrm{KT}}(x_0)}$$

We can invoke (41) and (42) to bound $|\widehat{p}(x_0) - \widehat{p}_{\mathrm{KT}}(x_0)|$ and $|\widehat{A}_{\mathrm{KT}}(x_0) - \widehat{A}(x_0)|$ respectively. Thus, we have

$$\mathbb{E}_{\mathcal{S}_{\mathrm{in}}}\left[\left(\widehat{f}_{\mathrm{KT}}(x_0) - \widehat{f}(x_0)\right)^2 \mathbb{I}[\mathcal{E}_{\mathrm{KT},\delta} \cap \mathcal{B} \cap \mathcal{C}]\right] \leq \left(\frac{c_a \sqrt{\frac{d}{n}} \log n \cdot \widehat{p}(x_0) + c_p \sqrt{\frac{d}{n}} \log n \cdot \widehat{A}(x_0)}{\widehat{p}(x_0) \cdot \widehat{p}_{\mathrm{KT}}(x_0)}\right)^2$$

$$\leq \frac{2d \cdot \log^2 n}{n} \left[ \left( \frac{c_a}{\widehat{p}_{\mathrm{KT}}(x_0)} \right)^2 + \left( \frac{\widehat{A}(x_0)}{\widehat{p}(x_0)} \right)^2 \left( \frac{c_p}{\widehat{p}_{\mathrm{KT}}(x_0)} \right)^2 \right]$$

$$\overset{(i)}{\leq} \frac{2d \cdot \log^2 n}{n} \left[ \frac{c_a^2 + Y_{\max}^2 c_p^2}{\widehat{p}_{\mathrm{KT}}(x_0)} \right]^2$$

$$\overset{(ii)}{\leq} \frac{Cd \log^2 n}{nh^{2d}},$$

for some positive constant $C$ that does not depend on $n$, where step (i) uses the fact that $\frac{\widehat{A}(x_0)}{\widehat{p}(x_0)} \leq Y_{\max}$ and step (ii) uses the lower bound on $\widehat{p}_{\mathrm{KT}}(x_0)$ from (46). Combining (43) to (45), we have

$$\mathbb{E}_{\mathcal{S}_{\mathrm{in}}} \left[ \left( \widehat{f}_{\mathrm{KT}}(x_0) - \widehat{f}(x_0) \right)^2 \mathbb{I}[\mathcal{E}_{\mathrm{KT},\delta}] \right] \leq c_0 \exp\{-c_1 nh^d\} + \frac{Cd \log n}{nh^{2d}}.$$

Note that the second term dominates so that we may drop the first term with slight change to the value of the constant $C$ in the bound (36).

### B.1.1 Proof of Lem. 3: Simultaneous $L^\infty$ bound using KT-COMPRESS++ with $\mathbf{k}_{\mathrm{NW}}$

We first decompose $\mathbf{k}_{\mathrm{NW}}$ as

$$\mathbf{k}_{\mathrm{NW}}((x_1, y_1), (x_2, y_2)) = \mathbf{k}_1((x_1, y_1), (x_2, y_2)) + \mathbf{k}_2((x_1, y_1), (x_2, y_2)), \quad \text{where}$$

$$\mathbf{k}_1((x_1, y_1), (x_2, y_2)) \triangleq \mathbf{k}(x_1, x_2) \quad \text{and} \tag{47}$$

$$\mathbf{k}_2((x_1, y_1), (x_2, y_2)) \triangleq \mathbf{k}(x_1, x_2) \cdot y_1 y_2. \tag{48}$$

and note that

$$\mathcal{H}(\mathbf{k}_{\mathrm{NW}}) = \mathcal{H}(\mathbf{k}_1) \oplus \mathcal{H}(\mathbf{k}_2). \tag{49}$$

This fact will be useful later for proving simultaneous $L^\infty$ approximation guarantees for $\widehat{A}$ and $\widehat{p}$.

Given that $\mathbf{k}$ satisfies Assum. 2, we want to show that $\mathbf{k}_{\mathrm{NW}}$ defined by (13) satisfies the Lipschitz and tail decay properties, so that we may apply Lem. 1. Note that

$$\|\mathbf{k}_{\mathrm{NW}}\|_\infty = \|\mathbf{k}\|_\infty (1 + Y_{\max}^2). \tag{50}$$

We claim that kernel $\mathbf{k}_{\mathrm{NW}}$ satisfies

$$L_{\mathbf{k}_{\mathrm{NW}}} \leq L_{\mathbf{k}} + Y_{\max}(\|\mathbf{k}\|_\infty + L_{\mathbf{k}} Y_{\max}) \quad \text{and} \tag{51}$$

$$R_{\mathbf{k}_{\mathrm{NW}}, n} \leq R_{\mathbf{k}, n} + 2Y_{\max} \tag{52}$$

By [10, Rmk. 8], we have

$$\frac{L_{\mathbf{k}}}{\|\mathbf{k}\|_\infty} \leq \frac{L_\kappa}{h} \quad \text{and} \quad R_{\mathbf{k}, n} \leq h\kappa^\dagger(1/n), \tag{53}$$

where $\kappa^\dagger$ is defined by (15). Applying (50) and (51), we have

$$\frac{L_{\mathbf{k}_{\mathrm{NW}}}}{\|\mathbf{k}_{\mathrm{NW}}\|_\infty} \leq \frac{L_{\mathbf{k}} + Y_{\max}(\|\mathbf{k}\|_\infty + L_{\mathbf{k}} Y_{\max})}{\|\mathbf{k}\|_\infty (1 + Y_{\max}^2)}$$

$$\leq \frac{L_{\mathbf{k}}}{\|\mathbf{k}\|_\infty} + \frac{1}{Y_{\max}} + \frac{L_{\mathbf{k}}}{\|\mathbf{k}\|_\infty}$$

$$\overset{(53)}{\leq} \frac{2L_\kappa}{h} + \frac{1}{Y_{\max}}.$$

Finally, we have

$$\frac{L_{\mathbf{k}_{\mathrm{NW}}} R_{\mathbf{k}_{\mathrm{NW}}, n}}{\|\mathbf{k}_{\mathrm{NW}}\|_\infty} \leq \left( \frac{2L_\kappa}{h} + \frac{1}{Y_{\max}} \right) \left( h\kappa^\dagger(1/n) + 2Y_{\max} \right)$$

$$= 2L_\kappa \kappa^\dagger(1/n) + \frac{4L_\kappa Y_{\max}}{h} + \frac{h\kappa^\dagger(1/n)}{Y_{\max}} + 2$$

$$\leq 4 \max\{1, L_\kappa Y_{\max}\} \cdot \frac{\kappa^\dagger(1/n)}{h}$$

$$\leq 4 \max\{1, L_\kappa Y_{\max}\} \cdot c' n^\alpha, \tag{54}$$

where the last inequality follows from Assum. 2 for some universal positive constant $c'$.

Since Assum. 1 is satisfied, $R$ is constant. Applying (54) to $\mathfrak{M}_{\mathbf{k}_{\mathrm{NW}}}(n, n_{\mathrm{out}}, d, \delta, R)$ as defined by (26), we have the bound

$$\mathfrak{M}_{\mathbf{k}_{\mathrm{NW}}}(n, n_{\mathrm{out}}, d, \delta, R) \leq c'' \sqrt{\log\left(\tfrac{n_{\mathrm{out}}}{\delta}\right)}\left[\sqrt{\log\left(\tfrac{8}{\delta}\right)} + 5\sqrt{d\log n}\right]$$

for some positive constant $c''$. Substituting this into (25), we have

$$\|(\mathbb{P}_{\mathrm{in}} - \mathbb{Q}_{\mathrm{out}})\mathbf{k}_{\mathrm{NW}}\|_\infty \leq c_1 \frac{\|\mathbf{k}\|_\infty (1+Y_{\max}^2)}{n_{\mathrm{out}}} \sqrt{\log\left(\tfrac{n_{\mathrm{out}}}{\delta}\right)}\left[\sqrt{\log\left(\tfrac{8}{\delta}\right)} + 5\sqrt{d\log n}\right]$$
$$\leq c_2 \frac{\|\mathbf{k}\|_\infty (1+Y_{\max}^2)}{n_{\mathrm{out}}} \sqrt{d}(\sqrt{\log n} + \sqrt{\log(1/\delta)})^2,$$

for some positive constants $c_1, c_2$. By definition,

$$\|(\mathbb{P}_{\mathrm{in}} - \mathbb{Q}_{\mathrm{out}})\mathbf{k}_{\mathrm{NW}}\|_\infty = \sup_{z\in\mathcal{Z}} \langle (\mathbb{P}_{\mathrm{in}} - \mathbb{Q}_{\mathrm{out}})\mathbf{k}_{\mathrm{NW}}, \mathbf{k}_{\mathrm{NW}}(\cdot, z)\rangle_{\mathbf{k}_{\mathrm{NW}}}.$$

Define $\mathbf{k}_1$ and $\mathbf{k}_2$ by (47) and (48), respectively, and note that $\mathbf{k}_1(\cdot, z), \mathbf{k}_2(\cdot, z) \in \mathcal{H}(\mathbf{k}_{\mathrm{NW}})$ for all $z \in \mathcal{Z}$. We want to show that

$$\sup_{z\in\mathcal{Z}} \langle (\mathbb{P}_{\mathrm{in}} - \mathbb{Q}_{\mathrm{out}})\mathbf{k}_{\mathrm{NW}}, \mathbf{k}_1(\cdot, z)\rangle_{\mathbf{k}_{\mathrm{NW}}} \leq c_2 \frac{\|\mathbf{k}\|_\infty (1+Y_{\max}^2)}{n_{\mathrm{out}}} \sqrt{d}(\sqrt{\log n} + \sqrt{\log(1/\delta)})^2 \quad \text{and}$$
$$\sup_{z\in\mathcal{Z}} \langle (\mathbb{P}_{\mathrm{in}} - \mathbb{Q}_{\mathrm{out}})\mathbf{k}_{\mathrm{NW}}, \mathbf{k}_2(\cdot, z)\rangle_{\mathbf{k}_{\mathrm{NW}}} \leq c_2 \frac{\|\mathbf{k}\|_\infty (1+Y_{\max}^2)}{n_{\mathrm{out}}} \sqrt{d}(\sqrt{\log n} + \sqrt{\log(1/\delta)})^2,$$

which would imply (41) and (42) (after simplifying all terms besides $n$, $d$, and $\delta$).

The first inequality follows from replacing all occurrences of the test function $\mathbf{k}_{\mathrm{NW}}(\cdot, (x, y))$ in the proof of Lem. 1 with the function $\mathbf{k}_1(\cdot, x)$ and noting that $\langle \mathbf{k}_{\mathrm{NW}}(\cdot, (x_i, y_i)), \mathbf{k}_1(\cdot, (x, y))\rangle_{\mathbf{k}_{\mathrm{NW}}} = \langle \mathbf{k}_1(\cdot, x_i), \mathbf{k}_1(\cdot, x)\rangle_{\mathbf{k}_1}$ from the fact that $\mathcal{H}(\mathbf{k}_{\mathrm{NW}}) = \mathcal{H}(\mathbf{k}_1) \oplus \mathcal{H}(\mathbf{k}_2)$ (49).

The second inequality follows from replacing all occurrences of the test function $\mathbf{k}_{\mathrm{NW}}(\cdot, (x, y))$ in the proof of Lem. 1 with the function $\mathbf{k}_2(\cdot, x)$ and noting that $\langle \mathbf{k}_{\mathrm{NW}}(\cdot, (x_i, y_i)), \mathbf{k}_2(\cdot, (x, y))\rangle_{\mathbf{k}_{\mathrm{NW}}} = \langle \mathbf{k}_2(\cdot, (x_i, y_i)), \mathbf{k}_2(\cdot, (x, y))\rangle_{\mathbf{k}_2}$, again from the fact that $\mathcal{H}(\mathbf{k}_{\mathrm{NW}}) = \mathcal{H}(\mathbf{k}_1) \oplus \mathcal{H}(\mathbf{k}_2)$ (49).

**Proof of claim (51).** We leverage the fact that the Lipschitz constants defined by (27) satisfies the following additivity property. Letting $\mathcal{Z} = \mathcal{S}_{\mathrm{in}}$, we have

$$L_{\mathbf{k}_{\mathrm{NW}}} = \sup_{z_1, z_2, z_3 \in \mathcal{Z}} \frac{|\mathbf{k}_{\mathrm{NW}}(z_1, z_2) - \mathbf{k}_{\mathrm{NW}}(z_1, z_3)|}{\|z_2 - z_3\|_2}$$
$$\leq \sup_{z_1, z_2, z_3 \in \mathcal{Z}} \frac{|\mathbf{k}_1(z_1, z_2) - \mathbf{k}_1(z_1, z_3)|}{\|z_2 - z_3\|_2} + \sup_{z_1, z_2, z_3 \in \mathcal{Z}} \frac{|\mathbf{k}_2(z_1, z_2) - \mathbf{k}_2(z_1, z_3)|}{\|z_2 - z_3\|_2}$$
$$= L_{\mathbf{k}_1} + L_{\mathbf{k}_2}.$$

We proceed to bound $L_{\mathbf{k}_1}$ and $L_{\mathbf{k}_2}$ separately. Note that

$$L_{\mathbf{k}_1} = L_{\mathbf{k}}.$$

Applying the definition (27) to $L_{\mathbf{k}_2}$, we have

$$L_{\mathbf{k}_2} = \sup_{\substack{z_1=(x_1,y_1)\\z_2=(x_2,y_2)\\z_3=(x_3,y_3)}} \frac{|\mathbf{k}(x_1,x_2)y_1y_2 - \mathbf{k}(x_1,x_3)y_1y_3|}{\sqrt{\|x_2-x_3\|^2 + \|y_2-y_3\|^2}}$$
$$= \sup_{\substack{z_1=(x_1,y_1)\\z_2=(x_2,y_2)\\z_3=(x_3,y_3)}} \frac{|y_1| \cdot |\mathbf{k}(x_1,x_2)y_2 - \mathbf{k}(x_1,x_2)y_3 + \mathbf{k}(x_1,x_2)y_3 - \mathbf{k}(x_1,x_3)y_3|}{\sqrt{\|x_2-x_3\|^2 + \|y_2-y_3\|^2}}$$
$$= \sup_{\substack{z_1=(x_1,y_1)\\z_2=(x_2,y_2)\\z_3=(x_3,y_3)}} \frac{|y_1| \cdot |\mathbf{k}(x_1,x_2)(y_2-y_3) + (\mathbf{k}(x_1,x_2) - \mathbf{k}(x_1,x_3))y_3|}{\sqrt{\|x_2-x_3\|^2 + \|y_2-y_3\|^2}}$$
$$\leq \sup_{\substack{z_1=(x_1,y_1)\\z_2=(x_2,y_2)\\z_3=(x_3,y_3)}} \frac{|y_1| \cdot |\mathbf{k}(x_1,x_2)(y_2-y_3)|}{\sqrt{\|x_2-x_3\|^2 + \|y_2-y_3\|^2}} + \sup_{\substack{z_1=(x_1,y_1)\\z_2=(x_2,y_2)\\z_3=(x_3,y_3)}} \frac{|y_1| \cdot |(\mathbf{k}(x_1,x_2) - \mathbf{k}(x_1,x_3))y_3|}{\sqrt{\|x_2-x_3\|^2 + \|y_2-y_3\|^2}}$$
$$\leq Y_{\max}\|\mathbf{k}\|_\infty + L_{\mathbf{k}} Y_{\max}^2.$$

Putting together the pieces yields the claimed bound.

**Proof of claim (52).** We aim to show that $R_{\mathbf{k}_{\mathrm{NW}},n}$ is not much larger than $R_{\mathbf{k},n}$. Note that $\mathbf{k}_{\mathrm{NW}}$ can be rewritten as

$$\mathbf{k}_{\mathrm{NW}}((x_1, y_1), (x_2, y_2)) = (1 + y_1 y_2)\mathbf{k}(x_1, x_2).$$

We define the sets

$$\Gamma \triangleq \left\{ r : \sup_{\substack{x_1, x_2: \\ \|x_1 - x_2\|_2 \geq r}} |\mathbf{k}(x_1, x_2)| \leq \frac{\|\mathbf{k}\|_\infty}{n} \right\} \quad \text{and} \tag{55}$$

$$\Gamma^\star \triangleq \left\{ r^\star : \sup_{\substack{(x_1, y_1), (x_2, y_2): \\ \|x_1 - x_2\|^2 + \|y_1 - y_2\|^2 \geq (r^\star)^2}} |\mathbf{k}(x_1, x_2) \cdot y_1 y_2| \leq \frac{\|\mathbf{k}_\star\|_\infty}{n} \right\}, \tag{56}$$

noting that

$$R_{\mathbf{k},n} = \inf \Gamma \quad \text{and} \quad R_{\mathbf{k}_{\mathrm{NW}},n} = \inf \Gamma^\star$$

by definition (28).

Suppose $r \in \Gamma$. Then for any $(x_1, y_1), (x_2, y_2)$ such that

$$\|x_1 - x_2\|^2 + \|y_1 - y_2\|^2 \geq r^2 + 4Y_{\max}^2,$$

it must follow that $\|x_1 - x_2\|^2 \geq r^2$ (since $\|y_1 - y_2\|^2 \leq 4Y_{\max}^2$ by triangle inequality). Since $r$ satisfies (55), it must follow that

$$|\mathbf{k}(x_1, x_2) \cdot (y_1 y_2 + 1)| \leq |\mathbf{k}(x_1, x_2)|(Y_{\max}^2 + 1) \leq \frac{\|\mathbf{k}\|_\infty}{n}(Y_{\max}^2 + 1) \overset{(50)}{=} \frac{\|\mathbf{k}_{\mathrm{NW}}\|_\infty}{n},$$

meaning $\sqrt{r^2 + 4Y_{\max}^2} \in \Gamma^\star$, where recall $\Gamma^\star$ is defined by (56). Thus, we have

$$R_{\mathbf{k}_{\mathrm{NW}},n} \leq \sqrt{R_{\mathbf{k},n} + 4Y_{\max}^2} \leq R_{\mathbf{k},n} + 2Y_{\max}$$

as desired.

## B.2 Proof of claim (37)

Suppose event $\mathcal{E}$ (32) is satisfied. Define the shorthand for the variance:

$$\sigma^2(x_0; X) \triangleq \mathbb{E}_{Y|X}\left[ \left( \widehat{f}(x_0) - \mathbb{E}_{Y|X}\left[ \widehat{f}(x_0) \right] \right)^2 \right].$$

Conditioned on $X_1 = x_1, X_2 = x_2, \ldots, X_n = x_n$, we have

$$\mathbb{E}_{Y|X}\left[ \widehat{f}(x_0) \right] = \mathbb{E}_{Y_1|X_1, \ldots, Y_n|X_n}\left[ \frac{\sum_{i=1}^n Y_i \mathbf{k}(X_i, x_0)}{\sum_{i=1}^n \mathbf{k}(X_i, x_0)} \right] = \frac{\sum_{i=1}^n f^\star(X_i)\mathbf{k}(X_i, x_0)}{\sum_{i=1}^n \mathbf{k}(X_i, x_0)}, \tag{57}$$

where we have used the fact that $\mathbb{E}[Y \mid X = \cdot] = f^\star(\cdot)$ by assumption (1).

Note that on event $\mathcal{E}$, we have

$$\sigma^2(x_0; X) \overset{(57)}{=} \mathbb{E}_{Y|X}\left[ \left( \frac{\sum_{i=1}^n Y_i \mathbf{k}(X_i, x_0)}{\sum_{i=1}^n \mathbf{k}(X_i, x_0)} - \frac{\sum_{i=1}^n f^\star(X_i)\mathbf{k}(X_i, x_0)}{\sum_{i=1}^n \mathbf{k}(X_i, x_0)} \right)^2 \right]$$

$$= \mathbb{E}_{Y|X}\left[ \left( \frac{\sum_{i=1}^n v_i \mathbf{k}(X_i, x_0)}{\sum_{i=1}^n \mathbf{k}(X_i, x_0)} \right)^2 \right]$$

$$= \mathrm{Var}[v_1] \cdot \sum_{i=1}^n \frac{\mathbf{k}(X_i, x_0)^2}{\left( \sum_{i=1}^n \mathbf{k}(X_i, x_0) \right)^2},$$

where recall $v_1, \ldots, v_n$ are i.i.d. random variables with $\mathrm{Var}[v_i] = \sigma^2$ by (1). Taking the expectation w.r.t. $X_1, \ldots, X_n$ and leveraging symmetry, we have

$$\mathbb{E}_X\left[ \sigma^2(x_0; X)\, \mathbb{I}[\mathcal{E}] \right] = n\sigma^2 \cdot \sigma_X^2, \quad \text{where} \quad \sigma_X^2 \triangleq \mathbb{E}_X\left[ \frac{\mathbf{k}^2(X_1, x_0)}{\left( \sum_{i=1}^n \mathbf{k}(X_i, x_0) \right)^2} \right]. \tag{58}$$

$\sigma_X^2$ can be bounded by

$$\sigma_X^2 \leq \mathbb{E}_X\left[ \frac{\mathbf{k}^2(X_1, x_0)}{\left( \sum_{i=1}^n \mathbf{k}(X_i, x_0) \right)^2} \mathbb{I}\left[ \sum_{i=1}^n \eta_i \leq \frac{n\bar{p}}{2} \right] \right] + \left( \frac{2}{n\bar{p}} \right)^2 \mathbb{E}_X\left[ \mathbf{k}^2(X_1, x_0) \right]$$

$$\overset{(i)}{\leq} \mathbb{P}\Big(\sum_{i=1}^n \eta_i \leq \tfrac{n\overline{p}}{2}\Big) + \Big(\tfrac{2}{n\overline{p}}\Big)^2 \int_{\mathcal{X}} \mathbf{k}^2(x_1, x_0)p(dx_1),$$

where step (i) follows from the fact that $\frac{\mathbf{k}^2(X_1, x_0)}{\left(\sum_{i=1}^n \mathbf{k}(X_i, x_0)\right)^2} \leq 1$. Using Bernstein's inequality [29, Prop. 2.14], the first term can be bounded by

$$
\begin{aligned}
\mathbb{P}\Big(\sum_{i=1}^n \eta_i \leq \tfrac{n\overline{p}}{2}\Big) &= \mathbb{P}\Big(\sum_{i=1}^n \eta_i - n\overline{p} \leq -\tfrac{n\overline{p}}{2}\Big) \\
&\leq \exp\Big\{-\tfrac{(n\overline{p})^2}{2(n\overline{p}(1-\overline{p})+n\overline{p}/3)}\Big\} \leq \exp\{-c_1 n h^d\}
\end{aligned}
$$

for some universal positive constant $c$. Applying the fact that $p$ is bounded by Assum. 1 and $\kappa$ is square-integrable by Assum. 2, we can bound the second term by

$$
\begin{aligned}
\Big(\tfrac{2}{n\overline{p}}\Big)^2 \int_{\mathcal{X}} \mathbf{k}^2(x_1, x_0)p(dx_1) &\overset{(i)}{\leq} \Big(\tfrac{2}{n\overline{p}}\Big)^2 p_{\max} h^d \int_{\mathbb{R}^d} \kappa^2(u)du \\
&\overset{(ii)}{\leq} \tfrac{4}{(n\overline{p})^2}\big(c_1 h^d\big) \leq \tfrac{c_2}{n^2 h^d},
\end{aligned}
$$

for some positive constant $c_2$ that does not depend on either $h$ or $n$. Substituting these expressions into (58) yields a bound on $\mathbb{E}_X\big[\sigma^2(x_0; X)\,\mathbb{I}[\mathcal{E}]\big]$ as desired.

## B.3  Proof of claim (38)

Define the following shorthand for the bias:

$$b(x_0; X) \triangleq \mathbb{E}_{Y|X}\Big[\widehat{f}(x_0)\Big] - f^\star(x_0).$$

We state a more detailed version of the claim.

**Lemma 4** (Bias of Nadaraya-Watson). *Suppose Assum. 1 and 2 are satisfied and the event $\mathcal{E}$ (32) holds. If $f^\star \in \Sigma(\beta, L_f)$ for $\beta \in (0, 1]$, $L_f > 0$, then with high probability, the following statements hold true uniformly for all $x_0 \in \mathcal{X}$:*

  *(I.a)  If $\mathbf{k}$ is compactly supported, then $b(x_0; X) \leq L_f h^\beta$.*

  *(I.b)  Otherwise, if $\mathbf{k}$ is non-compactly supported, we have $b(x_0; X) \leq c \cdot L_f h^\beta \log n$ for some positive constant $c$ that does not depend on $n$.*

For completeness, we first state the proof of claim (I.a) from Belkin et al. [3, Lem. 2]. On the event $\mathcal{E}$ (32), we have

$$b(x_0; X) \overset{(57)}{=} \tfrac{\sum_{i=1}^n (f^\star(X_i) - f^\star(x_0))\mathbf{k}(X_i, x_0)}{\sum_{i=1}^n \mathbf{k}(X_i, x_0)} \overset{(i)}{\leq} \tfrac{\sum_{i=1}^n L_f \|X_i\|^\beta \mathbf{k}(X_i, x_0)}{\sum_{i=1}^n \mathbf{k}(X_i, x_0)} \overset{(ii)}{\leq} L_f h^\beta,$$

where step (i) follows from our assumption that $f^\star \in \Sigma(\beta, L_f)$ and step (ii) follows from our assumption that $\mathbf{k}$ is compactly supported, so $\mathbf{k}(x, x_0) = 0$ whenever $\|x - x_0\| > h$. We now proceed to proving claim (I.b).

**Step I: Decomposing the bias.**    For any $x_0 \in \mathcal{X}$, define the random variables

$$A_0(x_0) \triangleq \tfrac{1}{n}\sum_{i=1}^n \mathbf{k}(X_i, x_0)\mathbb{I}[\|X_i - x_0\| \leq h] \quad \text{and} \tag{59}$$

$$A_j(x_0) \triangleq \tfrac{1}{n}\sum_{i=1}^n \mathbf{k}(X_i, x_0)\mathbb{I}\big[2^{j-1}h < \|X_i - x_0\| \leq 2^j h\big] \quad \text{for} \quad j = 1, 2, \ldots, T, \tag{60}$$

where

$$T \triangleq \Big\lceil \tfrac{\log R}{\log h} \Big\rceil = \mathcal{O}(\log n) \tag{61}$$

when $p_X(\cdot)$ has compact support under Assum. 1 and $h$ is defined by (40). We call these random variables the *kernel empirical means*. We can directly verify that

$$\sum_{j=0}^T A_0(x_0) = \tfrac{1}{n}\sum_{i=1}^n \mathbf{k}(X_i, x_0) \quad \text{for all} \quad x_0 \in \mathcal{X}.$$

With these definitions in hand, we may rewrite $b(x_0; X)$ in terms of $A_0(x_0), A_1(x_0), \ldots, A_T(x_0)$ as follows:

$$
\begin{aligned}
b(x_0; X) &= \frac{\frac{1}{n}\sum_{i=1}^{n} \mathbf{k}(X_i, x_0)(f^\star(X_i) - f^\star(x_0))}{\frac{1}{n}\sum_{i=1}^{n} \mathbf{k}(X_i, x_0)} \\
&= \frac{\frac{1}{n}\sum_{i=1}^{n} \mathbf{k}(X_i, x_0)\mathbb{I}[\|X_i - x_0\| \leq h](f^\star(X_i) - f^\star(x_0))}{A_0(x_0) + \sum_{j=1}^{T} A_j(x_0)} \\
&\quad + \frac{\sum_{j=1}^{T} \frac{1}{n}\sum_{i=1}^{n} \mathbf{k}(X_i, x_0)\mathbb{I}[2^{j-1}h < \|X_i - x_0\| \leq 2^j h](f^\star(X_i) - f^\star(x_0))}{A_0(x_0) + \sum_{j=1}^{T} A_j(x_0)} \\
&\overset{(i)}{\leq} \frac{\frac{1}{n}\sum_{i=1}^{n} \mathbf{k}(X_i, x_0)\mathbb{I}[\|X_i - x_0\| \leq h]L_f\|X_i - x_0\|^\beta}{A_0(x_0) + \sum_{j=1}^{T} A_j(x_0)} \\
&\quad + \frac{\sum_{j=1}^{T} \frac{1}{n}\sum_{i=1}^{n} \mathbf{k}(X_i, x_0)\mathbb{I}[2^{j-1}h < \|X_i - x_0\| \leq 2^j h]L_f\|X_i - x_0\|^\beta}{A_0(x_0) + \sum_{j=1}^{T} A_j(x_0)} \\
&\overset{(ii)}{\leq} \frac{\frac{1}{n}\sum_{i=1}^{n} \mathbf{k}(X_i, x_0)\mathbb{I}[\|X_i - x_0\| \leq h]L_f h^\beta}{A_0(x_0) + \sum_{j=1}^{T} A_j(x_0)} \quad (62) \\
&\quad + \frac{\sum_{j=1}^{T} \frac{1}{n}\sum_{i=1}^{n} \mathbf{k}(X_i, x_0)\mathbb{I}[2^{j-1}h < \|X_i - x_0\| \leq 2^j h]L_f (2^j h)^\beta}{A_0(x_0) + \sum_{j=1}^{T} A_j(x_0)}, \quad (63)
\end{aligned}
$$

where step (i) follows from the fact that $f^\star(X_i) - f^\star(x_0) \leq L_f\|X_i - x_0\|^\beta$ by Def. 1 and step (ii) follows from the fact that $\|X_i - x_0\| \leq 2^j h$ under the indicator functions. Applying (59) to (62) and (60) to (63), we have

$$
b(x_0; X) \leq \frac{A_0(x_0)L_f h^\beta}{A_0(x_0) + \sum_{j=1}^{T} A_j(x_0)} + \frac{\sum_{j=1}^{T} A_j(x_0)L_f(2^j h)^\beta}{A_0(x_0) + \sum_{j=1}^{T} A_j(x_0)} \leq L_f h^\beta \left( \sum_{i=0}^{T} \frac{A_i(x_0)2^{i\beta}}{\sum_{j=0}^{T} A_j(x_0)} \right). \quad (64)
$$

For any $z \in \mathcal{X}$, define the following shorthands:

$$
a_0(z) \triangleq \mathbb{E}_X[\mathbf{k}(X, z)\mathbb{I}[\|X - z\| \leq h]] \quad \text{and} \quad (65)
$$

$$
a_j(z) \triangleq \mathbb{E}_X[\mathbf{k}(X, x)\mathbb{I}[2^{j-1}h < \|X - z\| \leq 2^j h]] \quad \text{for} \quad j = 1, 2, \ldots, T. \quad (66)
$$

We can directly verify that

$$
\mathbb{E}_X[A_0(z)] = a_0(z) \quad \text{and} \quad \mathbb{E}_X[A_j(z)] = a_j(z) \quad \text{for all} \quad j = 1, 2, \ldots, T.
$$

Under Assum. 1, $a_i(z)$ can be computed directly from the decay properties of $\mathbf{k}$ as follows:

$$
a_0(z) = \int_{\|x-z\| \leq h} \mathbf{k}(x, z)p_X(dx) \geq p_{\min} \int_{\|x-z\| \leq h} \mathbf{k}(x, z)dx
$$

$$
a_j(z) = \int_{2^{j-1} < \|x-z\| \leq 2^j h} \mathbf{k}(x, z)p_X(dx) \leq p_{\max} \int_{2^{j-1} < \|x-z\| \leq 2^j h} \mathbf{k}(x, z)dx \quad \text{for} \quad j = 1, 2, \ldots, T.
$$

**Remark 1.** *Note that for fixed $x_0 \in \mathcal{X}$, $A_i(x_0)$, Hoeffding's inequality implies that $A_i(x_0)$ concentrates around $a_i(x_0)$ for all $i = 0, 1, \ldots, T$ with high probability. Assuming that $\mathbf{k}$ decays sufficiently rapidly (so that $a_j(x_0)2^{j\beta} = \mathcal{O}(a_0(x_0))$ for all $j \in [T]$), we have*

$$
b(x_0; X) \leq L_f h^\beta \left( \sum_{i=1}^{T} O(1) \right) = \mathcal{O}(L_f h^\beta \log n).
$$

*It remains to show that this bound holds uniformly over all $x_0 \in \mathcal{X}$ with high probability.*

**Step II: Uniform concentration of empirical kernel means.** Fix $\epsilon \in (0, \frac{h}{2}]$ and let $\mathcal{C}_0$ be a $\epsilon$-cover of $\mathcal{X}$ w.r.t. the Euclidean norm (see [29, Def. 5.1] for the detailed definition). We will optimize the choice of $\epsilon$ at the end of the proof. By [29, Lem. 5.7], we have

$$
|\mathcal{C}_0| \leq (1 + \tfrac{2R}{\epsilon})^d \quad (67)
$$

We now introduce a slightly modified version of (65) and (66) defined as

$$
a_0^\epsilon(z) \triangleq \mathbb{E}_X[\mathbf{k}(X, z)\mathbb{I}[\|X - z\| \leq h - \epsilon]] \quad \text{and}
$$

$$
a_j^\epsilon(z) \triangleq \mathbb{E}_X[\mathbf{k}(X, z)\mathbb{I}[2^{j-1}h - \epsilon < \|X - z\| \leq 2^j h + \epsilon]] \quad \text{for} \quad j = 1, 2, \ldots, T,
$$

respectively. Using the fact that $\epsilon \leq h/2$, Assum. 1 and 2, and a change of variables, we have

$$
a_0^\epsilon(z) \geq p_{\min} \int_{\|x-z\| \leq \frac{h}{2}} \mathbf{k}(x, z)dx = p_{\min} h^d \omega_{d-1} \int_0^{\frac{1}{2}} \kappa(u)u^{d-1}du, \quad \text{and}
$$

$$a_j^\epsilon(z) \leq p_{\max} \int_{(2^{j-1}-\frac{1}{2})h < \|x-z\| \leq (2^j+\frac{1}{2})h} \mathbf{k}(x, x_0') dx = p_{\max} h^d \omega_{d-1} \int_{2^{j-1}-\frac{1}{2}}^{2^j+\frac{1}{2}} \kappa(u) u^{d-1} du \quad \text{for} \quad j \in [T],$$

where $\omega_{d-1} \triangleq 2\pi^{d/2}/\Gamma(d/2)$ denotes the surface area of the unit sphere in $\mathbb{R}^d$. Moreover, we consider the following concentration events:

$$\mathcal{A}_0^\epsilon(z) \triangleq \left\{ \frac{1}{n} \sum_{i=1}^n \mathbf{k}(X_i, z) \mathbb{I}[\|X_i - z\| \leq h - \epsilon] \geq a_0^\epsilon(z) - \Delta_0 \right\} \quad \text{and} \tag{68}$$

$$\mathcal{A}_j^\epsilon(z) \triangleq \left\{ \frac{1}{n} \sum_{i=1}^n \mathbf{k}(X_i, z) \mathbb{I}[2^{j-1}h - \epsilon < \|X_i - z\| \leq 2^j h + \epsilon] \leq a_j^\epsilon(z) + \Delta_j \right\} \quad \text{for} \quad j \in [T], \tag{69}$$

where $\Delta_0, \Delta_1, \ldots, \Delta_T$ are positive scalars to be chosen later.

For any scalars $0 < a < b$ and $z \in \mathcal{X}$, further denote the empirical mass contained within the ball or annulus centered at point $z$ by

$$\text{Mass}_z(a) \triangleq \frac{1}{n} \sum_{i=1}^n \mathbb{I}[\|X_i - z\| \leq a] \quad \text{and}$$

$$\text{Mass}_z(a, b) \triangleq \frac{1}{n} \sum_{i=1}^n \mathbb{I}[a < \|X_i - z\| \leq b].$$

We now introduce a lemma that bounds the empirical kernel mean restricted to each ball and annuli:

**Lemma 5** (Uniform concentration of empirical kernel means). *Suppose event $\bigcap_{z \in \mathcal{C}_0} \bigcap_{i=0}^T \mathcal{A}_i^\epsilon(z)$ is satisfied. Then for every $x_0 \in \mathcal{X}$, there exists $x_0' \in \mathcal{C}_0$ such that $\|x_0 - x_0'\| \leq \epsilon$ and the following bounds hold simultaneously:*

$$A_0(x_0) \geq a_0^\epsilon(x_0') - \Delta_0 - \text{Mass}_{x_0'}(h - \epsilon) \cdot L_{\mathbf{k}} \epsilon \quad \textit{and} \tag{70}$$

$$A_j(x_0) \leq a_j^\epsilon(x_0') + \Delta_j + \text{Mass}_{x_0'}(2^{j-1}h - \epsilon, 2^j h + \epsilon) \cdot \kappa(\tfrac{2^{j-1}h - 2\epsilon}{h}) \quad \textit{for} \quad j \in [T]. \tag{71}$$

*Here, $L_{\mathbf{k}}$ and $\kappa(\cdot)$ are properties of $\mathbf{k}$ from Assum. 2.*

See App. B.3.1 for the proof. This result suggests that we can control the error between the empirical kernel mean, $A_i(x_0)$, and its (shifted) population counterpart, $a_j^\epsilon(x_0')$, by choosing $\epsilon$ to be small and leveraging the decay property of $\kappa$. To bound the $\text{Mass}_{x_0'}(h - \epsilon)$ and $\text{Mass}_{x_0'}(2^{j-1}h - \epsilon, 2^j h + \epsilon)$ terms, we define the following events:

$$\mathcal{M}_0(x_0') \triangleq \left\{ \text{Mass}_{x_0'}(h - \epsilon) \leq \mathbb{E}_X \left[ \text{Mass}_{x_0'}(h - \epsilon) \right] + t_0 \right\}, \quad \text{and} \tag{72}$$

$$\mathcal{M}_j(x_0') \triangleq \left\{ \text{Mass}_{x_0'}(2^{j-1}h - \epsilon, 2^j h + \epsilon) \leq \mathbb{E}_X \left[ \text{Mass}_{x_0'}(2^{j-1}h - \epsilon, 2^j h + \epsilon) \right] + t_j \right\} \quad \text{for} \quad j \in [T], \tag{73}$$

where $t_0, t_1, \ldots, t_T$ are positive scalars to be chosen later. For any scalars $0 < a < b$, we may apply Assum. 1 to obtain

$$\mathbb{E}_X \left[ \text{Mass}_{x_0'}(a) \right] \leq p_{\max} \text{Vol}(a) \quad \text{and} \quad \mathbb{E}_X \left[ \text{Mass}_{x_0'}(a, b) \right] \leq p_{\max} \text{Vol}(b), \tag{74}$$

where $\text{Vol}(r) \triangleq \pi^d/\Gamma(d/2 + 1) r^d$ denotes the volume of the Euclidean ball $\mathbb{B}_2(r) \subset \mathbb{R}^d$.

**Step III: Putting things together.** For any $z \in \mathcal{C}_0$, note that

$$\mathbf{k}(X_i, z) \mathbb{I}[\|X_i - z\| \leq h - \epsilon] \in [0, \|\mathbf{k}\|_\infty] \quad \text{and}$$

$$\mathbf{k}(X_i, z) \mathbb{I}[2^{j-1}h - \epsilon < \|X_i - z\| \leq 2^j h + \epsilon] \in [0, \|\mathbf{k}\|_\infty]$$

and therefore these random variables are sub-Gaussian with parameter $\sigma = \frac{1}{2} \|\mathbf{k}\|_\infty$. Thus, we may bound (68) and (69) using Hoeffding's bound [29, Prop. 2.5], obtaining

$$\mathbb{P}(\mathcal{A}_i^\epsilon(z)^c) \leq \exp\left\{ -\frac{2n\Delta_0^2}{\|\mathbf{k}\|_\infty^2} \right\} \quad \text{for all} \quad i = 0, 1, \ldots, T \quad \text{and} \quad z \in \mathcal{C}_0, \tag{75}$$

Again by Hoeffding's bound [29, Prop. 2.5], we have

$$\mathbb{P}(\mathcal{M}_i(z)^c) \leq \exp\left\{ -2nt_i^2 \right\} \quad \text{for all} \quad i = 0, 1, \ldots, T \quad \text{and} \quad z \in \mathcal{C}_0.$$

Next, it remains to choose the values of $\epsilon, \Delta_0, \ldots, \Delta_T$ and $t_0, \ldots, t_T$. Consider

$$\Delta_0 = \frac{1}{2} \cdot p_{\min} h^d \omega_{d-1} \int_0^{\frac{1}{2}} \kappa(u) u^{d-1} du, \tag{76}$$

$$\Delta_j = 2^{-j\beta} \cdot p_{\max} h^d \omega_{d-1} \int_0^{\frac{1}{2}} \kappa(u) u^{d-1} du \quad \text{for} \quad j = 1, 2, \ldots, T, \tag{77}$$

$$t_0 = \tfrac{1}{2} p_{\max} \mathrm{Vol}(h - \epsilon), \quad \text{and} \tag{78}$$

$$t_j = \tfrac{1}{2} p_{\max} \mathrm{Vol}(2^j h + \epsilon) \quad \text{for} \quad j = 1, 2, \ldots, T. \tag{79}$$

Combining (70), (72), (74), (76), and (78), we have on the event $\mathcal{E}_0 \triangleq \bigcap_{z \in \mathcal{C}_0} (\mathcal{A}_0^\epsilon(z) \cap \mathcal{M}_0(z))$ the following bound:

$$A_0(x_0) \geq \tfrac{1}{2} p_{\min} h^d \omega_{d-1} \int_0^{\frac{1}{2}} \kappa(u) u^{d-1} du - p_{\max} \frac{\pi^d}{\Gamma(\frac{d}{2}+1)} (h - \epsilon)^d \cdot L_k \epsilon$$

$$\geq h^d \omega_{d-1} \Big( \tfrac{1}{2} p_{\min} \int_0^{\frac{1}{2}} \kappa(u) u^{d-1} du - p_{\max} \frac{\pi^d}{\Gamma(\frac{d}{2}+1)\omega_{d-1}} L_{\mathbf{k}} \epsilon \Big)$$

$$\overset{(i)}{\geq} \tfrac{1}{4} p_{\min} h^d \omega_{d-1} \int_0^{\frac{1}{2}} \kappa(u) u^{d-1} du,$$

where step (i) follows from setting

$$\epsilon = \min \left\{ \tfrac{1}{4} \Big[ p_{\min} \int_0^{\frac{1}{2}} \kappa(u) u^{d-1} du \Big] \Big( p_{\max} \frac{\pi^d}{\Gamma(\frac{d}{2}+1)\omega_{d-1}} L_{\mathbf{k}} \Big)^{-1}, \frac{h}{2} \right\}. \tag{80}$$

Similarly, for each $j = 1, 2, \ldots, T$, we may combine (71), (73), (74), (77), and (79) to obtain a bound on $A_j(x_0)$. Hence, on the $\mathcal{E}_j \triangleq \bigcap_{z \in \mathcal{C}_0} (\mathcal{A}_j^\epsilon(z) \cap \mathcal{M}_j(z))$, we have

$$A_j(x_0) \leq p_{\max} h^d \omega_{d-1} \Big( \int_{2^{j-1}-\frac{1}{2}}^{2^j+\frac{1}{2}} \kappa(u) u^{d-1} du + 2^{-j\beta} \int_0^{\frac{1}{2}} \kappa(u) u^{d-1} du \Big) + p_{\max} \frac{\pi^d}{\Gamma(\frac{d}{2}+1)} (2^j h + \epsilon)^d \cdot \kappa(\tfrac{2^{j-1}h-2\epsilon}{h})$$

$$\overset{(i)}{\leq} p_{\max} h^d \omega_{d-1} \Big( \int_{2^{j-1}-\frac{1}{2}}^{2^j+\frac{1}{2}} \kappa(u) u^{d-1} du + 2^{-j\beta} \int_0^{\frac{1}{2}} \kappa(u) u^{d-1} du + \frac{\pi^d (2^j+\frac{1}{2})^d}{\Gamma(\frac{d}{2}+1)\omega_{d-1}} \kappa(2^{j-1} - 1) \Big)$$

where step (i) employs the fact that $\epsilon \leq h/2$. Note that for $j = 1, 2, \ldots, T$:

$$\frac{A_j(x_0) 2^{j\beta}}{A_0(x_0)} \leq \frac{4 p_{\max}}{p_{\min}} \left( \frac{2^{j\beta} \int_{2^{j-1}-\frac{1}{2}}^{2^j+\frac{1}{2}} \kappa(u) u^{d-1} du + \int_0^{\frac{1}{2}} \kappa(u) u^{d-1} du + \frac{\pi^d (2^j+\frac{1}{2})^d 2^{j\beta}}{\Gamma(\frac{d}{2}+1)\omega_{d-1}} \kappa(2^{j-1}-1)}{\int_0^{\frac{1}{2}} \kappa(u) u^{d-1} du} \right) = \mathcal{O}(1), \tag{81}$$

where the last inequality comes from the fact that

$$\frac{2^{j\beta} \int_{2^{j-1}-\frac{1}{2}}^{2^j+\frac{1}{2}} \kappa(u) u^{d-1} du}{\int_0^{\frac{1}{2}} \kappa(u) u^{d-1} du} = \mathcal{O}(1) \quad \text{and} \quad \frac{\frac{\pi^d (2^j+\frac{1}{2})^d 2^{j\beta}}{\Gamma(\frac{d}{2}+1)\omega_{d-1}} \kappa(2^{j-1}-1)}{\int_0^{\frac{1}{2}} \kappa(u) u^{d-1} du} = \mathcal{O}(1)$$

by Assum. 2. Substituting this bound into (64), we have

$$b(x_0; X) \overset{(64)}{\leq} L_f h^\beta \Big( \sum_{i=0}^T \frac{A_i(x_0) 2^{j\beta}}{\sum_{j=0}^T A_j(x_0)} \Big) \leq L_f h^\beta \Big( \sum_{i=0}^T \frac{A_i(x_0) 2^{j\beta}}{A_0(x_0)} \Big) \overset{(81)}{\leq} L_f h^\beta TC \overset{(61)}{\leq} \mathcal{O}(L_f h^\beta \log n)$$

as desired. Finally, it remains to control the probability that this bound on $b(x_0; X)$ holds. Applying union-bound, we have

$$\mathbb{P}(\mathcal{E}_0^c \cup \ldots \cup \mathcal{E}_T^c) \overset{(67),(75)}{\leq} (1 + \tfrac{2R}{\epsilon})^d \sum_{i=0}^T \Big( \exp\Big\{ -\frac{2n\Delta_i^2}{\|\mathbf{k}\|_\infty^2} \Big\} + \exp\{-2nt_i^2\} \Big)$$

$$\leq (1 + \tfrac{2R}{\epsilon})^d \sum_{j=0}^T \big( \exp\{-c_1 n h^{2d} 4^{-j\beta}\} + \exp\{-c_2 n h^{jd}\} \big)$$

$$\leq (1 + \tfrac{2R}{\epsilon})^d (T+1) \big( \exp\{-c_1 n h^{2d} 4^{-T\beta}\} + \exp\{-c_2 n h^{Td}\} \big),$$

for some universal positive constants $c_1, c_2$ that do not depend on $n$. Note that by (80), we have $(1 + \tfrac{2R}{\epsilon})^d = \mathcal{O}((1 + \tfrac{1}{h})^d)$. Setting $h = cn^{-\frac{1}{2\beta+2d}}$ as in (40) and recalling that $T = \mathcal{O}(\log n)$, we have $\mathbb{P}(\mathcal{E}_0^c \cup \ldots \cup \mathcal{E}_T^c) \ll 1$ for sufficiently large $n$. This completes the proof of Lem. 4.

### B.3.1 Proof of Lem. 5: Uniform concentration of empirical kernel means

Fix any $x_0 \in \mathcal{X}$. By definition of $\mathcal{C}_0$, there exists $x_0' \in \mathcal{C}_0$ such that $\|x_0 - x_0'\| \leq \epsilon$. Consider any $x \in \mathbb{B}_2(h - \epsilon; x_0')$[4]. By triangle inequality, $\|x - x_0\| \leq \|x - x_0'\| + \|x_0' - x_0\| \leq (h - \epsilon) + \epsilon$. Therefore, we have

$$\mathbb{B}_2(h - \epsilon; x_0') \subset \mathbb{B}_2(h; x_0). \tag{82}$$

We now introduce the following lemma, which is a finite-sample version of [10, Lem. 8]:

---

[4]We use the notation $\mathbb{B}_2(r; x)$ to denote the Euclidean ball of radius $r$ centered at $x \in \mathbb{R}^d$.

**Lemma 6** ($L^\infty$ bound on $L^2$ kernel error (finite sample))**.** *Consider any bounded kernel $\mathbf{k} \in L^{2,\infty}$, points $x_0, x_0' \in \mathcal{X}$, and function $g \in L^2(\mathbb{P}_n)$. For any $r, a, b \geq 0$ with $a + b = 1$, points $x_1, \ldots, x_n \in \mathcal{X}$, and set $A \subset \mathcal{X}$, we have*

$$\left| \tfrac{1}{n} \textstyle\sum_{i=1}^n g(x_i)(\mathbf{k}(x_i, x_0) - \mathbf{k}(x_i, x_0'))\mathbf{1}_A(x_i) \right|$$

$$\leq \|g\|_{A, L^2(\mathbb{P}_n)} \cdot \left[ \|\mathbf{k}(\cdot, x_0) - \mathbf{k}(\cdot, x_0')\|_{A,\infty} \mathrm{Mass}_A^{\frac{1}{2}}(r) + 2\tau_{\mathbf{k},A}(ar) + 2\|\mathbf{k}\|_{A, L^{2,\infty}} \mathbb{I}[\|x_0 - x_0'\| \geq br] \right].$$

*where*

$$\|f\|_{A, L^2(\mathbb{P}_n)} \triangleq \left( \tfrac{1}{n} \textstyle\sum_{i=1}^n f^2(x_i)\mathbf{1}_A(x_i) \right)^{\frac{1}{2}}$$

$$\|f\|_{A,\infty} \triangleq \max_{i \in [n]} |f(x_i)\mathbf{1}_A(x_i)|$$

$$\mathrm{Mass}_A(r) \triangleq \tfrac{1}{n} \textstyle\sum_{i=1}^n \mathbb{I}[\|x_i - x_0'\| \leq r, x_i \in A]$$

$$\tau_{A,\mathbf{k}}(r) \triangleq \sup_{x' \in \{x_0, x_0'\}} \left( \tfrac{1}{n} \textstyle\sum_{i=1}^n \mathbf{k}^2(x_i, x')\mathbb{I}[\|x_i - x'\| \geq r, x_i \in A] \right)^{\frac{1}{2}}$$

$$\|\mathbf{k}\|_{A, L^{2,\infty}} \triangleq \sup_{x' \in \{x_0, x_0'\}} \|\mathbf{k}(\cdot, x')\|_{A, L^2(\mathbb{P}_n)}.$$

See App. B.3.2 for the proof. We now use this lemma to prove each of the claims in Lem. 5.

**Proof of claim (70).** We consider the following basic decomposition:

$$\tfrac{1}{n} \textstyle\sum_{i=1}^n \mathbf{k}(X_i, x_0)\mathbb{I}[\|X_i - x_0\| \leq h] \tag{83}$$

$$= \tfrac{1}{n} \textstyle\sum_{i=1}^n \mathbf{k}(X_i, x_0')\mathbb{I}[\|X_i - x_0'\| \leq h - \epsilon] \tag{84}$$

$$- \left( \tfrac{1}{n} \textstyle\sum_{i=1}^n \mathbf{k}(X_i, x_0')\mathbb{I}[\|X_i - x_0'\| \leq h - \epsilon] - \tfrac{1}{n} \textstyle\sum_{i=1}^n \mathbf{k}(X_i, x_0)\mathbb{I}[\|X_i - x_0\| \leq h] \right). \tag{85}$$

Note that the term (84) can be directly bounded using the definition of (68). Next, we bound the term (85). Define

$$g \triangleq \mathbf{1} \quad \text{and} \quad A \triangleq \mathbb{B}_2(h - \epsilon; x_0'),$$

so that $\|g\|_{A, L^2(\mathbb{P}_n)} = \mathrm{Mass}_{x_0'}^{\frac{1}{2}}(h - \epsilon)$. By property (82), we have

$$\mathbb{I}[\|X_i - x_0'\| \leq h - \epsilon] \leq \mathbb{I}[\|X_i - x_0\| \leq h] \quad \text{for all} \quad i = 1, 2, \ldots, n. \tag{86}$$

Observe that

$$\tfrac{1}{n} \textstyle\sum_{i=1}^n g(X_i)(\mathbf{k}(X_i, x_0) - \mathbf{k}(x_i, x_0'))\mathbf{1}_A(x_i) \tag{87}$$

$$= \tfrac{1}{n} \textstyle\sum_{i=1}^n (\mathbf{k}(X_i, x_0') - \mathbf{k}(X_i, x_0))\mathbb{I}[\|X_i - x_0'\| \leq h - \epsilon]$$

$$= \tfrac{1}{n} \textstyle\sum_{i=1}^n \mathbf{k}(X_i, x_0')\mathbb{I}[\|X_i - x_0'\| \leq h - \epsilon] - \tfrac{1}{n} \textstyle\sum_{i=1}^n \mathbf{k}(X_i, x_0)\mathbb{I}[\|X_i - x_0'\| \leq h - \epsilon]$$

$$\overset{(86)}{\geq} \tfrac{1}{n} \textstyle\sum_{i=1}^n \mathbf{k}(X_i, x_0')\mathbb{I}[\|X_i - x_0'\| \leq h - \epsilon] - \tfrac{1}{n} \textstyle\sum_{i=1}^n \mathbf{k}(X_i, x_0)\mathbb{I}[\|X_i - x_0\| \leq h],$$

To upper bound (87), we now apply Lem. 6 with $r = h$, $a = \frac{1}{2}$, and $b = \frac{1}{2}$ to obtain

$$\left| \tfrac{1}{n} \textstyle\sum_{i=1}^n g(x_i)(\mathbf{k}(x_i, x_0) - \mathbf{k}(x_i, x_0'))\mathbf{1}_A(x_i) \right|$$

$$\leq \mathrm{Mass}_{x_0'}^{\frac{1}{2}}(h - \epsilon) \cdot \left[ \|\mathbf{k}(\cdot, x_0) - \mathbf{k}(\cdot, x_0')\|_{A,\infty} \mathrm{Mass}_A^{\frac{1}{2}}(h) + 2\tau_{\mathbf{k},A}(\tfrac{h}{2}) + 2\|\mathbf{k}\|_{A, L^{2,\infty}} \mathbb{I}[\|x_0 - x_0'\| > \tfrac{h}{2}] \right]$$

$$\leq \mathrm{Mass}_{x_0'}^{\frac{1}{2}}(h - \epsilon) \cdot \left[ L_{\mathbf{k}}\epsilon \cdot \mathrm{Mass}_{x_0'}^{\frac{1}{2}}(h - \epsilon) \right], \tag{88}$$

where the last inequality uses the fact that

$$\|\mathbf{k}(\cdot, x_0) - \mathbf{k}(\cdot, x_0')\|_{A,\infty} \leq L_{\mathbf{k}}\|x_0 - x_0'\| \leq L_{\mathbf{k}}\epsilon$$

from the Lipschitz property of $\mathbf{k}$ (see Assum. 2) and the fact that

$$\mathrm{Mass}_A^{\frac{1}{2}}(h) = \mathrm{Mass}_{x_0'}^{\frac{1}{2}}(h - \epsilon), \qquad \tau_{\mathbf{k},A}(\tfrac{h}{2}) = 0, \quad \text{and} \quad \|x_i - x_0\| \leq \epsilon \leq \tfrac{h}{2}.$$

Substituting the bounds (68), (87), and (88) into the decomposition (83) yields the desired claim.

**Proof of claim (71).** We consider the basic decomposition:

$$\frac{1}{n}\sum_{i=1}^{n}\mathbf{k}(X_i,x_0)\mathbb{I}\big[2^{j-1}h-\epsilon<\|X_i-x_0\|\le 2^j h+\epsilon\big]\tag{89}$$

$$=\frac{1}{n}\sum_{i=1}\mathbf{k}(X_i,x_0')\mathbb{I}\big[2^{j-1}h-\epsilon<\|X_i-x_0'\|\le 2^j h+\epsilon\big]\tag{90}$$

$$+\big(\frac{1}{n}\sum_{i=1}^{n}\mathbf{k}(X_i,x_0')\mathbb{I}\big[2^{j-1}h<\|X_i-x_0\|\le 2^j h\big]$$

$$-\frac{1}{n}\sum_{i=1}^{n}\mathbf{k}(X_i,x_0')\mathbb{I}\big[2^{j-1}h-\epsilon<\|X_i-x_0'\|\le 2^j h+\epsilon\big]\big)\tag{91}$$

Note that the term (90) can be directly bounded using the definition of (69). Next, we bound the term (91). By triangle inequality, we have

$$\|x-x_0'\|\le\|x-x_0\|+\|x_0-x_0'\|\le 2^j h+\epsilon,\quad\text{for all}\quad x\in\mathbb{B}_2(2^j h;x_0),$$

which implies that $\mathbb{B}_2(2^j h;x_0)\subset\mathbb{B}_2(2^j h+\epsilon;x_0')$. Moreover, we can verify that $\mathbb{B}_2(2^{j-1}h-\epsilon;x_0')\subset\mathbb{B}_2(2^{j-1}h;x_0)$. Hence,

$$\mathbb{I}\big[2^{j-1}h<\|X_i-x_0\|\le 2^j h\big]\le\mathbb{I}\big[2^{j-1}h-\epsilon<\|X_i-x_0'\|\le 2^j h+\epsilon\big]\quad\text{for all}\quad i=1,2,\dots,n.\tag{92}$$

Now define

$$g\triangleq 1\quad\text{and}\quad A\triangleq\mathbb{B}_2(2^j h+\epsilon;x_0')\setminus\mathbb{B}_2(2^{j-1}h-\epsilon;x_0'),\tag{93}$$

so that $\|g\|_{A,L^2(\mathbb{P}_n)}=\mathrm{Mass}_{x_0'}^{\frac{1}{2}}(2^{j-1}h-\epsilon,2^j h+\epsilon)$. Hence, (91) can be bounded by

$$\frac{1}{n}\sum_{i=1}^{n}\mathbf{k}(X_i,x_0')\mathbb{I}\big[2^{j-1}h<\|X_i-x_0\|\le 2^j h\big]-\frac{1}{n}\sum_{i=1}^{n}\mathbf{k}(X_i,x_0')\mathbb{I}\big[2^{j-1}h-\epsilon<\|X_i-x_0'\|\le 2^j h+\epsilon\big]$$

$$\overset{(92)}{\le}\frac{1}{n}\sum_{i=1}^{n}\mathbf{k}(X_i,x_0')\mathbb{I}\big[2^{j-1}h-\epsilon<\|X_i-x_0'\|\le 2^j h+\epsilon\big]$$

$$-\frac{1}{n}\sum_{i=1}^{n}\mathbf{k}(X_i,x_0')\mathbb{I}\big[2^{j-1}h-\epsilon<\|X_i-x_0'\|\le 2^j h+\epsilon\big]$$

$$\overset{(93)}{\le}\frac{1}{n}\sum_{i=1}^{n}g(X_i)(\mathbf{k}(X_i,x_0')-\mathbf{k}(X_i,x_0'))\mathbf{1}_A(x_i)$$

$$\le\mathrm{Mass}_{x_0'}^{\frac{1}{2}}(2^{j-1}h-\epsilon,2^j h+\epsilon)\cdot\big[\|\mathbf{k}(\cdot,x_0)-\mathbf{k}(\cdot,x_0')\|_{A,\infty}\mathrm{Mass}_A^{\frac{1}{2}}(2(2^j h+\epsilon))$$

$$+2\tau_{\mathbf{k},A}(2^j h+\epsilon)+2\|\mathbf{k}\|_{A,L^2,\infty}\mathbb{I}\big[\|x_0-x_0'\|\ge 2^j h+\epsilon\big]\big]\tag{94}$$

where the last inequality follows from applying Lem. 6 with $r=2(2^j h+\epsilon)$ and $a=b=\frac{1}{2}$. By triangle inequality, we have

$$\|X_i-x_0\|\ge\|X_i-x_0'\|-\|x_0'-x_0\|\overset{(i)}{\ge}(2^{j-1}h-\epsilon)-\epsilon\ge 2^{j-1}h-2\epsilon,$$

where step (i) follows from the definition of $A$ (93). Since $\kappa$ is monotonically decreasing by Assum. 2, we have $\kappa(\frac{\|X_i-x_0\|}{h})\le\kappa(\frac{2^{j-1}h-2\epsilon}{h})$. Thus, we have

$$\|\mathbf{k}(\cdot,x_0)-\mathbf{k}(\cdot,x_0')\|_{A,\infty}\le\max\{\|\mathbf{k}(\cdot,x_0)\|_{A,\infty},\|\mathbf{k}(\cdot,x_0')\|_{A,\infty}\}$$

$$\le\max\Big\{\kappa(\tfrac{2^{j-1}h-2\epsilon}{h}),\kappa(\tfrac{2^{j-1}h-\epsilon}{h})\Big\}$$

$$\overset{(i)}{\le}\kappa(\tfrac{2^{j-1}h-2\epsilon}{h}),\tag{95}$$

where step (i) again uses the fact that $\kappa$ is monotonically decreasing. Moreover, note that

$$\mathrm{Mass}_A(2(2^j h+\epsilon))=\mathrm{Mass}_{x_0'}(2^{j-1}h-\epsilon,2^j h+\epsilon),$$

$$\tau_{A,\mathbf{k}}(2^j h+\epsilon)=0,\quad\text{and}$$

$$\|x_0-x_0'\|\le\epsilon<2^j h+\epsilon.$$

Substituting the bounds (69), (94), and (95) into the decomposition (89) yields the desired claim.

### B.3.2 Proof of Lem. 6: $L^\infty$ bound on $L^2$ kernel error (finite sample)

Define the restrictions

$$g_r(x)=g(x)\mathbf{1}_{\mathbb{B}_2(r;x_0')}(x),\qquad g_r^{(c)}=g-g_r,\qquad \mathbf{k}_r(x,z)\triangleq\mathbf{k}(x,z)\cdot\mathbf{1}_{\mathbb{B}_2(r;x_0')}(z),\quad\text{and}\quad\mathbf{k}_r^{(c)}\triangleq\mathbf{k}-\mathbf{k}_r,$$

so that $\mathbf{k}(z, x_0) = \mathbf{k}_r(z, x_0) + \mathbf{k}_r^{(c)}(z, x_0)$ and $\mathbf{k}(z, x_0') = \mathbf{k}_r(z, x_0') + \mathbf{k}_r^{(c)}(z, x_0')$. We now apply the triangle inequality and Hölder's inequality to obtain

$$\left| \frac{1}{n} \sum_{i=1}^n g(x_i)(\mathbf{k}(x_i, x_0) - \mathbf{k}(x_i, x_0')) \mathbf{1}_A(x_i) \right|$$

$$= \left| \frac{1}{n} \sum_{i=1}^n g_r(x_i)(\mathbf{k}_r(x_i, x_0) - \mathbf{k}_r(x_i, x_0')) \mathbf{1}_A(x_i) + \frac{1}{n} \sum_{i=1}^n g_r^{(c)}(x_i)(\mathbf{k}_r^{(c)}(x_i, x_0) - \mathbf{k}_r^{(c)}(x_i, x_0')) \mathbf{1}_A(x_i) \right|$$

$$\leq \left| \frac{1}{n} \sum_{i=1}^n g_r(x_i)(\mathbf{k}_r(x_i, x_0) - \mathbf{k}_r(x_i, x_0')) \mathbf{1}_A(x_i) \right|$$

$$+ \left| \frac{1}{n} \sum_{i=1}^n g_r^{(c)}(x_i)(\mathbf{k}_r^{(c)}(x_i, x_0) - \mathbf{k}_r^{(c)}(x_i, x_0')) \mathbf{1}_A(x_i) \right|$$

$$\leq \|g_r \cdot \mathbf{1}_A\|_{L^1(\mathbb{P}_n)} \cdot \|(\mathbf{k}(\cdot, x_0) - \mathbf{k}(\cdot, x_0')) \mathbf{1}_A(\cdot)\|_\infty \tag{96}$$

$$+ \left| \frac{1}{n} \sum_{i=1}^n g_r^{(c)}(x_i)(\mathbf{k}(x_i, x_0) - \mathbf{k}(x_i, x_0')) \mathbf{1}_A(x_i) \right|, \tag{97}$$

where $\|\cdot\|_{L^1(\mathbb{P}_n)}$ is defined by $\|f\|_{L^1(\mathbb{P}_n)} \triangleq \frac{1}{n} \sum_{i=1}^n |f(x_i)|$. To bound the term (96), we apply Cauchy-Schwarz to $g_r \in L^1(\mathbb{P}_n) \cap L^2(\mathbb{P}_n)$ to obtain

$$\|g_r \cdot \mathbf{1}_A\|_{L^1(\mathbb{P}_n)} \leq \|g_r \cdot \mathbf{1}_A\|_{L^2(\mathbb{P}_n)} \cdot \sqrt{\mathrm{Mass}_A(r)} \leq \|g\|_{A, L^2(\mathbb{P}_n)} \cdot \sqrt{\mathrm{Mass}_A(r)}.$$

Next, we bound the term (97). For any $x^\star \in \{x_0, x_0'\}$ and $x_i$ satisfying $\|x_i - x_0'\| \geq r$ and scalars $a, b \in [0, 1]$ such that $a + b = 1$, either $\|x^\star - x_i\| \geq ar$ or $\|x^\star - x_0'\| > br$. Hence,

$$\left| \frac{1}{n} \sum_{i=1}^n g_r^{(c)}(\mathbf{k}(x_i, x_0) - \mathbf{k}(x_i, x_0') \mathbf{1}_A(x_i) \right|$$

$$\leq \left| \frac{1}{n} \sum_{i=1}^n \mathbb{I}[\|x_i - x_0\| \geq r] g_r^{(c)}(x_i) \sum_{x^\star \in \{x_0, x_0'\}} \mathbf{k}(x_i, x^\star) \mathbf{1}_A(x_i)(\delta_{x_0}(x^\star) - \delta_{x_0'}(x^\star)) \right|$$

$$\leq \underbrace{\left| \frac{1}{n} \sum_{i=1}^n \sum_{x^\star \in \{x_0, x_0'\}} \mathbb{I}[\|x_i - x^\star\| \geq ar] g_r^{(c)}(x_i) \mathbf{k}(x_i, x^\star) \mathbf{1}_A(x_i)(\delta_{x_0}(x^\star) - \delta_{x_0'}(x^\star)) \right|}_{\triangleq T_1} \tag{98}$$

$$+ \underbrace{\left| \frac{1}{n} \sum_{i=1}^n \sum_{x^\star \in \{x_0, x_0'\}} \mathbb{I}[\|x^\star - x_0\| > br] g_r^{(c)}(x_i) \mathbf{k}(x_i, x^\star) \mathbf{1}_A(x_i)(\delta_{x_0}(x^\star) - \delta_{x_0'}(x^\star)) \right|}_{\triangleq T_2},$$

$$\tag{99}$$

where we define $\delta_{x_0}(z) \triangleq \mathbb{I}[z = x_0]$ and $\delta_{x_0'}(z) \triangleq \mathbb{I}[z = x_0']$.

**Bounding (98).** Exchanging the order of the summations and applying Cauchy-Schwarz, we have

$$T_1 = \left| \sum_{x^\star \in \{x_0, x_0'\}} \frac{1}{n} \sum_{i=1}^n \mathbb{I}[\|x^\star - x_i\| \geq ar] g_r^{(c)}(x_i) \mathbf{k}(x^\star, x_i) \mathbf{1}_A(x_i)(\delta_{x_0}(x^\star) - \delta_{x_0'}(x^\star)) \right|$$

$$\leq \sum_{x^\star \in \{x_0, x_0'\}} \left| \frac{1}{n} \sum_{i=1}^n \mathbb{I}[\|x_i - x^\star\| \geq ar] g_r^{(c)}(x_i) \mathbf{k}(x^\star, x_i) \mathbf{1}_A(x_i)(\delta_{x_0}(x^\star) - \delta_{x_0'}(x^\star)) \right|$$

$$\leq \sum_{x^\star \in \{x_0, x_0'\}} \left( \frac{1}{n} \sum_{i=1}^n \mathbb{I}[\|x_i - x^\star\| \geq ar] g_r^{(c)}(x_i)^2 \mathbf{1}_A(x_i) \right)^{\frac{1}{2}}$$

$$\cdot \left( \frac{1}{n} \sum_{i=1}^n \mathbb{I}[\|x_i - x^\star\| \geq ar] \mathbf{k}^2(x_i, x^\star) \mathbf{1}_A(x_i) \right)^{\frac{1}{2}} \cdot \left| \delta_{x_0}(x^\star) - \delta_{x_0'}(x^\star) \right|$$

$$\leq \sum_{x^\star \in \{x_0, x_0'\}} \left( \frac{1}{n} \sum_{i=1}^n \mathbb{I}[\|x_i - x^\star\| \geq ar] g_r^{(c)}(x_i)^2 \mathbf{1}_A(x_i) \right)^{\frac{1}{2}}$$

$$\cdot \sup_{x' \in \{x_0, x_0'\}} \left( \frac{1}{n} \sum_{i=1}^n \mathbb{I}[\|x_i - x'\| \geq ar] \mathbf{k}^2(x_i, x') \mathbf{1}_A(x_i) \right)^{\frac{1}{2}} \cdot \left| \delta_{x_0}(x^\star) - \delta_{x_0'}(x^\star) \right|$$

$$= \sum_{x^\star \in \{x_0, x_0'\}} \|g \cdot \mathbf{1}_A\|_{L^2(\mathbb{P}_n)} \cdot \tau_\mathbf{k}(ar) \cdot \left| \delta_{x_0}(x^\star) - \delta_{x_0'}(x^\star) \right|$$

$$= 2 \|g \cdot \mathbf{1}_A\|_{L^2(\mathbb{P}_n)} \cdot \tau_\mathbf{k}(ar)$$

**Bounding (99).** Rearranging the terms and applying Cauchy-Schwarz, we have

$$T_2 = \left| \sum_{x^\star \in \{x_0, x_0'\}} \mathbb{I}[\|x^\star - x_0'\| > br] \frac{1}{n} \sum_{i=1}^n g_r^{(c)}(x_i) \mathbf{k}(x_i, x^\star) \mathbf{1}_A(x_i)(\delta_{x_0}(x^\star) - \delta_{x_0'}(x^\star)) \right|$$

$$\leq \sum_{x^\star \in \{x_0, x_0'\}} \mathbb{I}[\|x^\star - x_0'\| > br] \cdot \left| \frac{1}{n} \sum_{i=1}^n g_r^{(c)}(x_i) \mathbf{k}(x_i, x^\star) \mathbf{1}_A(x_i) \right| \cdot \left| \delta_{x_0}(x^\star) - \delta_{x_0'}(x^\star) \right|$$

$$\leq \sum_{x^\star \in \{x_0, x_0'\}} \mathbb{I}[\|x^\star - x_0'\| > br] \cdot \|g \cdot \mathbf{1}_A\|_{L^2(\mathbb{P}_n)} \sup_{x' \in \{x_0, x_0'\}} \|\mathbf{k}(\cdot, x') \cdot \mathbf{1}_A\|_{L^2(\mathbb{P}_n)}$$

$$\cdot \left| \delta_{x_0}(x^\star) - \delta_{x_0'}(x^\star) \right|$$
$$= \|g \cdot \mathbf{1}_A\|_{L^2(\mathbb{P}_n)} \sup_{x' \in \{x_0, x_0'\}} \|\mathbf{k}(\cdot, x') \cdot \mathbf{1}_A\|_{L^2(\mathbb{P}_n)} \cdot \mathbb{I}[\|x_0 - x_0'\| > br].$$

## C  Proof of Thm. 2: KT-KRR for finite-dimensional RKHS

We rely on the localized Gaussian/Rademacher analysis of KRR from prior work [29]. Define the *Gaussian critical radius* $\varepsilon_n > 0$ to be the smallest positive solution to the inequality

$$\widehat{\mathcal{G}}_n(\varepsilon; \mathbb{B}_{\mathcal{H}}(3)) \leq \tfrac{R}{2\sigma}\varepsilon^2, \quad \text{where} \quad \widehat{\mathcal{G}}_n(\varepsilon; \mathcal{F}) \triangleq \mathbb{E}_w \left[ \sup_{\substack{f \in \mathcal{F}: \\ \|f\|_n \leq \varepsilon}} \left| \tfrac{1}{n} \sum_{i=1}^n w_i f(x_i) \right| \right], \quad (100)$$

$\mathbb{B}_{\mathcal{H}}(3)$ is the $\|\cdot\|_{\mathbf{k}}$-ball of radius 3 and $w_i \overset{\text{i.i.d.}}{\sim} \mathcal{N}(0, 1)$.

**Assumption 4.** *Assume that $\|f^\star\|_{\mathbf{k}} \in \mathbb{B}_{\mathbf{k}}(R)$ and $\widehat{f}_{\mathrm{KT},\lambda'} \in \mathbb{B}_{\mathbf{k}}(c_\dagger R)$, for some constant $c_\dagger > 0$.*

Note that for any $g \in \mathbb{B}_{\mathbf{k}}(c_\dagger R)$, we have

$$\|g\|_\infty \leq \sup_{x \in \mathcal{X}} \langle g, \mathbf{k}(\cdot, x) \rangle_{\mathbf{k}} \overset{(i)}{\leq} \sup_{x \in \mathcal{X}} \|g\|_{\mathbf{k}} \|\mathbf{k}(\cdot, x)\|_{\mathbf{k}} \leq \|g\|_{\mathbf{k}} \sqrt{\|\mathbf{k}\|_\infty} \leq c_\dagger R \sqrt{\|\mathbf{k}\|_\infty} \triangleq B,$$

where step (i) follows from Cauchy-Schwarz. Thus, the function class $\mathbb{B}_{\mathbf{k}}(c_\dagger R)$ is $B$-uniformly bounded. Now define the *Rademacher critical radius* $\delta_n > 0$ to be the smallest positive solution to the inequality

$$\mathcal{R}_n(\delta; \mathcal{H}) \leq \delta^2, \quad \text{where} \quad \mathcal{R}_n(\delta; \mathcal{F}) \triangleq \mathbb{E}_{x,\nu} \left[ \sup_{\substack{f \in \mathcal{F}: \\ \|f\|_2 \leq \delta}} \left| \tfrac{1}{n} \sum_{i=1}^n \nu_i f(x_i) \right| \right] \quad (101)$$

and $\nu_i = \pm 1$ each with probability $1/2$.

Finally, we use the following shorthand to control the KT approximation error term,

$$\eta_{n,\mathbf{k}} \triangleq \tfrac{\mathfrak{a}^2}{n_{\mathrm{out}}} (2 + \mathfrak{W}_{\mathbf{k}}(n, n_{\mathrm{out}}, \delta, \mathfrak{R}_{\mathrm{in}}, \tfrac{\mathfrak{a}}{n_{\mathrm{out}}})), \quad \text{where} \quad (102)$$

$$\mathfrak{R}_{\mathrm{in}} \triangleq \max_{x \in \mathcal{S}_{\mathrm{in}}} \|x\|_2 \quad \text{and} \quad \mathfrak{a} \triangleq \|\mathbf{k}\|_{\infty,\mathrm{in}} + Y_{\max}^2 \quad (103)$$

and $\mathfrak{W}_{\mathbf{k}}$ is an *inflation factor* defined in (29) that scales with the covering number $\mathcal{N}_{\mathbf{k}}$ (see Def. 2). With these definitions in place, we are ready to state a detailed version of Thm. 2:

**Theorem 4** (KT-KRR for finite-dimensional RKHS, detailed). *Suppose the kernel operator associated with $\mathbf{k}$ and $\mathbb{P}$ has eigenvalues $\mu_1 \geq \ldots \geq \mu_m > 0$ (by Mercer's theorem). Define $C_m \triangleq 1/\mu_m$. Let $\varepsilon_n$ and $\delta_n$ denote the solutions to (101) and (100), respectively. Further assume* [5]

$$n\delta_n^2 > \log(4\log(1/\delta_n)). \quad (104)$$

*Let $\widehat{f}_{\mathrm{KT},\lambda'}$ denote the KT-KRR estimator with regularization parameter*

$$\lambda' \geq 2\xi_n^2 \quad \text{where} \quad \xi_n \triangleq \varepsilon_n \vee \delta_n \vee 4\sqrt{C_m}(\|f^\star\|_{\mathbf{k}} + 1)\eta_{n,\mathbf{k}}.$$

*Then with probability at least $1 - 2\delta - 2e^{-\frac{n\delta_n^2}{c_1(b^2 + \sigma^2)}}$, we have*

$$\|\widehat{f}_{\mathrm{KT},\lambda'} - f^\star\|_2^2 \leq c\{\xi_n^2 + \lambda'\}\|f^\star\|_{\mathbf{k}}^2 + c\delta_n^2. \quad (105)$$

*where recall $\delta$ is the success probability of* KT-COMPRESS++ (24).

See App. D for the proof. We set $\lambda' = 2\xi_n^2$, so that (105) becomes

$$\|\widehat{f}_{\mathrm{KT},\lambda'} - f^\star\|_2^2 \leq 3c\xi_n^2\|f^\star\|_{\mathbf{k}}^2 + c\delta_n^2. \quad (106)$$

It remains to bound the quantities $\varepsilon_n$ (100), $\delta_n$ (101), and $\eta_{n,\mathbf{k}}$ (102). We claim that

$$\varepsilon_n \leq c_0 \tfrac{\sigma}{R} \sqrt{\tfrac{m}{n}} \quad (107)$$

---

[5]Note that when $\mathbf{k}$ is finite-rank, this condition is automatically satisfied.

$$\delta_n \leq c_1 b \sqrt{\tfrac{m}{n}} \tag{108}$$

$$\eta_{n,\mathbf{k}} \leq c_2 \frac{\sqrt{m \cdot \log n_{\mathrm{out}} \cdot \log(1/\delta)}}{n_{\mathrm{out}}}. \tag{109}$$

for some universal positive constants $c_0, c_1, c_2$. Now set

$$R = \|f^\star\|_{\mathbf{k}} \qquad \delta = e^{-1/R^4}.$$

Thus, we have

$$\xi_n \leq c' \big( \tfrac{\sigma}{\|f^\star\|_{\mathbf{k}}} \vee b \vee \tfrac{4\sqrt{C_m}}{\|f^\star\|_{\mathbf{k}}} \big) \frac{\sqrt{m}}{\sqrt{n} \wedge n_{\mathrm{out}}}.$$

for some universal positive constant $c'$. Substituting this into (106) leads to the advertised bound (19).

**Proof of claim (107).** For finite rank kernels, $\hat{\mu}_j = 0$ for $j > m$. Thus, we have $\sqrt{\tfrac{2}{n}} \sqrt{\sum_{j=1}^n \min\{\varepsilon^2, \hat{\mu}_j\}} = \sqrt{\tfrac{2}{n}} \sqrt{m\varepsilon^2}$. From the critical radius condition (100), we want $\sqrt{\tfrac{2}{n}} \sqrt{m\varepsilon^2} \leq \tfrac{R}{4\sigma} \varepsilon^2$, so we may set $\varepsilon_n \simeq \tfrac{\sigma}{R} \sqrt{\tfrac{m}{n}}$.

**Proof of claim (108).** By similar logic as above, we have $\sqrt{\tfrac{2}{n}} \sqrt{\sum_{j=1}^n \min\{\delta^2, \mu_j\}} = \sqrt{\tfrac{2}{n}} \sqrt{m\delta^2}$. From the critical radius condition (101), we want $\sqrt{\tfrac{2}{n}} \sqrt{m\delta^2} \leq \tfrac{1}{b} \delta^2$, so we may set $\delta_n \simeq b\sqrt{2} \sqrt{\tfrac{m}{n}}$.

**Proof of claim (109).** Consider the linear operator $T : \mathcal{H} \to \mathbb{R}^m$ that maps a function to the coefficients in the vector space spanned by $\{\phi_i\}_{i=1}^m$. Note that

$$\|T\| = \frac{\|Tf\|_\infty}{\|f\|_{\mathbf{k}}} \leq \sqrt{\|\mathbf{k}\|_\infty}$$

Since the image of $T$ has dimension $m$, we have $\mathrm{rank}(T) \leq m$. Moreover, $\|\mathbf{k}\|_\infty \leq \mu_1 \cdot \mathfrak{R}_{\mathrm{in}}^2$. Now we can invoke [25, Eq. 14] with $\epsilon = \mathfrak{a}/n_{\mathrm{out}}$ to obtain

$$\mathcal{N}_{\mathbf{k}}(\mathbb{B}_2^d(\mathfrak{R}_{\mathrm{in}}), \mathfrak{a}/n_{\mathrm{out}}) \leq \mathcal{N}(T, \mathfrak{a}/n_{\mathrm{out}}) \leq (1 + \mu_1 \mathfrak{R}_{\mathrm{in}}^2 n_{\mathrm{out}}/\mathfrak{a})^m.$$

Taking the log on both sides and substituting this bound into (102), we have

$$\eta_{n,\mathbf{k}} = \tfrac{\mathfrak{a}^2}{n_{\mathrm{out}}} \big( 2 + \mathfrak{W}_{\mathbf{k}}(n, n_{\mathrm{out}}, \delta, \mathfrak{R}_{\mathrm{in}}, \tfrac{\mathfrak{a}}{n_{\mathrm{out}}}) \big)$$

$$\leq \tfrac{\mathfrak{a}^2}{n_{\mathrm{out}}} \Big( 2 + \sqrt{\log\Big( \tfrac{n_{\mathrm{out}} \log(n/n_{\mathrm{out}})}{\delta} \Big) \cdot \Big[ \log\big(\tfrac{1}{\delta}\big) + \log \mathcal{N}_{\mathbf{k}}(\mathbb{B}_2^d(\mathfrak{R}_{\mathrm{in}}), \tfrac{\mathfrak{a}}{n_{\mathrm{out}}}) \Big]} \Big)$$

$$\leq \tfrac{\mathfrak{a}^2}{n_{\mathrm{out}}} \Big( 2 + \sqrt{\log\Big( \tfrac{n_{\mathrm{out}} \log(n/n_{\mathrm{out}})}{\delta} \Big) \cdot \Big[ \log\big(\tfrac{1}{\delta}\big) + m \log\Big( 1 + \tfrac{2\|T\| n_{\mathrm{out}}}{\mathfrak{a}} \Big) \Big]} \Big)$$

$$\leq c \frac{\sqrt{m \cdot \log n_{\mathrm{out}} \cdot \log(1/\delta)}}{n_{\mathrm{out}}}$$

for some positive constant $c$ that doesn't depend on $m, n_{\mathrm{out}}, \delta$.

# D  Proof of Thm. 4: KT-KRR for finite-dimensional RKHS, detailed

We rescale our observation model (1) by $\|f^\star\|_{\mathbf{k}}$, so that the noise variance is $(\sigma/\|f^\star\|_{\mathbf{k}})^2$ and our new regression function satisfies $\|f^\star\|_{\mathbf{k}} = 1$. Our final prediction error should then be multiplied by $\|f^\star\|_{\mathbf{k}}^2$ to recover a result for the original problem. For simplicity, denote

$$\widetilde{\sigma} = \sigma/\|f^\star\|_{\mathbf{k}}.$$

For notational convenience, define an event

$$\mathcal{E} = \big\{ \|\widehat{f}_{\mathrm{KT}, \lambda'} - f^\star\|_2^2 \leq c(\xi_n^2 + \lambda') \big\},$$

and our goal is to show that $\mathcal{E}$ occurs with high-probability in terms of $\mathbb{P}$, the probability regarding all the randomness. For that end, we introduce several events that are used throughout,

$$\mathcal{E}_{\mathrm{conc}} \triangleq \Big\{ \sup_{g \in \mathcal{H}} \big| \|g\|_n - \|g\|_2 \big| \leq \tfrac{\delta_n}{2} \Big\} \quad \text{and} \quad \mathcal{E}_{\mathrm{lower}} \triangleq \big\{ \|\widehat{f}_{\mathrm{KT}, \lambda'} - f^\star\|_2 > \delta_n \big\}, \tag{110}$$

where $\delta_n$ is defined in (101) and $\mathcal{H}$ is the RKHS generated by $\mathbf{k}$ hence star-shaped. Further, we introduce two technical events $\mathcal{A}_{\mathrm{KT}}(u), \mathcal{B}_{\mathrm{KT}}$ defined in (123) and (136) respectively, which are proven to occur with small probability, and define a shorthand

$$\mathcal{E}_{\mathrm{good}} \triangleq \mathcal{A}_{\mathrm{KT}}^c(\xi_n) \cap \mathcal{B}_{\mathrm{KT}}^c \cap \mathcal{E}_{\mathrm{conc}} \cap \mathcal{E}_{\mathrm{KT},\delta}.$$

Equipped with these shorthands, observe the following inequality,

$$
\begin{aligned}
\mathbb{P}(\mathcal{E}) &= \mathbb{P}(\mathcal{E} \cap \mathcal{E}_{\mathrm{lower}}) + \mathbb{P}(\mathcal{E} \cap \mathcal{E}_{\mathrm{lower}}^c) \\
&\geq \mathbb{P}(\mathcal{E} \cap \mathcal{E}_{\mathrm{lower}}) + \mathbb{P}(\mathcal{E}_{\mathrm{lower}}^c)
\end{aligned}
\tag{111}
$$

where the second inequality is because $\mathcal{E}_{\mathrm{lower}}^c \subseteq \mathcal{E}_{\mathrm{lower}}^c \cap \mathcal{E}$ due to the assumption $\lambda' \geq 2\xi_n^2 \geq 2\delta_n^2$.

If we are able to show the set inclusion $\{\mathcal{E}_{\mathrm{good}} \cap \mathcal{E}_{\mathrm{lower}}\} \subseteq \{\mathcal{E} \cap \mathcal{E}_{\mathrm{lower}}\}$ and that $\mathbb{P}(\mathcal{E}_{\mathrm{good}}^c)$ is small, we are able refine (111) to the following

$$\mathbb{P}(\mathcal{E}) \geq \mathbb{P}(\mathcal{E}_{\mathrm{good}} \cap \mathcal{E}_{\mathrm{lower}}) + \mathbb{P}(\mathcal{E}_{\mathrm{lower}}^c) \geq 1 - \mathbb{P}(\mathcal{E}_{\mathrm{good}}^c) - \mathbb{P}(\mathcal{E}_{\mathrm{lower}}^c) + \mathbb{P}(\mathcal{E}_{\mathrm{lower}}) = 1 - \mathbb{P}(\mathcal{E}_{\mathrm{good}}^c),$$

where the last quantity $1 - \mathbb{P}(\mathcal{E}_{\mathrm{good}}^c)$ would be large.

To complete this proof strategy, we claim the set inclusion

$$\{\mathcal{E}_{\mathrm{good}} \cap \mathcal{E}_{\mathrm{lower}}\} \subseteq \{\mathcal{E} \cap \mathcal{E}_{\mathrm{lower}}\}
\tag{112}
$$

to hold and prove it in App. D.1 and further claim

$$\mathbb{P}(\mathcal{E}_{\mathrm{good}}^c) \leq c'' \{\delta + e^{-c' n \delta_n^2 / (B_{\mathcal{H}}^2 \wedge \widetilde{\sigma}^2)}\}
\tag{113}
$$

which verify in App. D.2.

Putting the pieces together, claims (112) and (113) collectively implies

$$\mathbb{P}(\mathcal{E}) \geq 1 - c'' \{\delta + e^{-c' n \delta_n^2 / (B_{\mathcal{H}}^2 \wedge \widetilde{\sigma}^2)}\}$$

as desired.

### D.1 Proof of claim (112)

There are several intermediary steps we take to show the set inclusion of interest (112). We introduce the shorthand

$$\widehat{\Delta}_{\mathrm{KT}} \triangleq \widehat{f}_{\mathrm{KT},\lambda'} - f^\star.$$

By invoking Propositions and basic inequalities to come, we successively show the following chain of set inclusions

$$
\begin{aligned}
\mathcal{E}_{\mathrm{good}} \cap \mathcal{E}_{\mathrm{lower}} &\subseteq \mathcal{E}_{\mathrm{good}} \cap \mathcal{E}_{\mathrm{lower}} \cap \{\|\widehat{\Delta}_{\mathrm{KT}}\|_{n_{\mathrm{out}}}^2 \leq c(\xi_n^2 + \lambda')\} \tag{114} \\
&\subseteq \mathcal{E}_{\mathrm{good}} \cap \mathcal{E}_{\mathrm{lower}} \cap \{\|\widehat{\Delta}_{\mathrm{KT}}\|_n^2 \leq c(\xi_n^2 + \lambda')\} \tag{115} \\
&\subseteq \mathcal{E}_{\mathrm{good}} \cap \mathcal{E}_{\mathrm{lower}} \cap \mathcal{E} \tag{116} \\
&\subseteq \mathcal{E}_{\mathrm{lower}} \cap \mathcal{E}. \tag{117}
\end{aligned}
$$

Note that step (117) is achieved trivially by dropping $\mathcal{E}_{\mathrm{good}}$. Further note that (115) is the crucial intermediary step after which we may apply uniform concentration across $n$ independent samples. Proof of (115) leverages on the Proposition to come (Prop. 1) that allows $\|\cdot\|_n$ and $\|\cdot\|_{n_{\mathrm{out}}}$ to be exchangeable for finite rank kernels.

**Recovering step (114)**  Since $\widehat{f}_{\mathrm{KT},\lambda'}$ and $f^\star$ are optimal and feasible, respectively for the central optimization problem of interest

$$\min_{f \in \mathcal{H}(\mathbf{k})} \frac{1}{n_{\mathrm{out}}} \sum_{i=1}^{n_{\mathrm{out}}} (y_i' - f(x_i'))^2 + \lambda' \|f\|_{\mathbf{k}}^2,$$

we have the basic inequality

$$\frac{1}{n_{\mathrm{out}}} \sum_{i=1}^{n_{\mathrm{out}}} \left(y_i' - \widehat{f}_{\mathrm{KT},\lambda'}(x_i')\right)^2 + \lambda' \|\widehat{f}_{\mathrm{KT},\lambda'}\|_{\mathbf{k}}^2 \leq \frac{1}{n_{\mathrm{out}}} \sum_{i=1}^{n_{\mathrm{out}}} (y_i' - f^\star(x_i'))^2 + \lambda' \|f^\star\|_{\mathbf{k}}^2, \tag{118}$$

With some algebra , may refine (118) to

$$\tfrac{1}{2}\|\widehat{\Delta}_{\mathrm{KT}}\|_{n_{\mathrm{out}}}^2 \le \left|\tfrac{1}{n_{\mathrm{out}}}\sum_{i=1}^{n_{\mathrm{out}}} v_i'\widehat{\Delta}_{\mathrm{KT}}(x_i')\right| + \lambda\Big\{\|f^\star\|_{\mathbf{k}}^2 - \|\widehat{f}_{\mathrm{KT},\lambda'}\|_{\mathbf{k}}^2\Big\}. \tag{119}$$

where $\widehat{\Delta}_{\mathrm{KT}} = \widehat{f}_{\mathrm{KT},\lambda'} - f^\star$. Suppose that $\|\widehat{\Delta}_{\mathrm{KT}}\|_{n_{\mathrm{out}}} < \xi_n$, then we trivially recover (114) by adding $\lambda' > 0$. Thus, we assume that $\|\widehat{\Delta}_{\mathrm{KT}}\|_{n_{\mathrm{out}}} \ge \xi_n$.

Under the assumption $\|\widehat{\Delta}_{\mathrm{KT}}\|_{n_{\mathrm{out}}} \ge \xi_n$, which is without loss of generality, we utilize the basic inequality (119) and control its stochastic component

$$\left|\tfrac{1}{n_{\mathrm{out}}}\sum_{i=1}^{n_{\mathrm{out}}} v_i'\widehat{\Delta}_{\mathrm{KT}}(x_i')\right|,$$

with a careful case work to follow, which is technical by nature.

Case where $\|\widehat{f}_{\mathrm{KT},\lambda'}\|_{\mathbf{k}} \le 2$: Under such case, we introduce a technical event

$$\mathcal{A}_{\mathrm{KT}}(u) \triangleq \left\{\exists g \in \mathcal{F}\setminus\mathbb{B}_2(\delta_n)\cap\{\|g\|_{n_{\mathrm{out}}} \ge u\}\text{ such that }\left|\tfrac{1}{n_{\mathrm{out}}}\sum_{i=1}^{n_{\mathrm{out}}} v_i'g(x_i')\right| \ge 3\|g\|_{n_{\mathrm{out}}}u\right\},$$

for any star-shaped function class $\mathcal{F}\subset\mathcal{H}$. Since $\|f^\star\|_{\mathbf{k}} = 1$, triangle inequality implies $\|\widehat{\Delta}_{\mathrm{KT}}\|_{\mathbf{k}} \le \|\widehat{f}_{\mathrm{KT},\lambda'}\|_{\mathbf{k}} + \|f^\star\|_{\mathbf{k}} \le 3$. Moreover, on the event $\mathcal{E}_{\mathrm{lower}}$ ($\subseteq \mathcal{E}_{\mathrm{good}}$), we have $\|\widehat{\Delta}_{\mathrm{KT}}\|_2 > \delta_n$. Thus, we may apply $\widehat{\Delta}_{\mathrm{KT}}$ to the event $\mathcal{A}_{\mathrm{KT}}^c(\xi_n)$ with $\mathcal{F} = \mathbb{B}_{\mathcal{H}}(3)$ (i.e., the $\mathcal{H}$-ball of radius 3) to attain

$$\left|\tfrac{1}{n_{\mathrm{out}}}\sum_{i=1}^{n_{\mathrm{out}}} v_i'\widehat{\Delta}_{\mathrm{KT}}(x_i')\right| \le c_0\xi_n\|\widehat{\Delta}_{\mathrm{KT}}\|_{n_{\mathrm{out}}}\text{ on the event }\mathcal{A}_{\mathrm{KT}}^c(\xi_n)\cap\mathcal{E}_{\mathrm{lower}}. \tag{120}$$

Upper bounding the stochastic component of the basic inequality (119) by (120) and dropping the $-\|\widehat{f}_{\mathrm{KT},\lambda'}\|_{\mathbf{k}}^2$ term in (119), we have

$$\tfrac{1}{2}\|\widehat{\Delta}_{\mathrm{KT}}\|_{n_{\mathrm{out}}}^2 \le c_0\xi_n\|\widehat{\Delta}_{\mathrm{KT}}\|_{n_{\mathrm{out}}} + \lambda'.$$

As a last step under the case $\|\widehat{f}_{\mathrm{KT},\lambda'}\|_{\mathbf{k}} \le 2$, apply the quadratic formula (specifically, if $a, b \ge 0$ and $x^2 - ax - b \le 0$, then $x \le a^2 + b$) to obtain

$$\|\widehat{\Delta}_{\mathrm{KT}}\|_{n_{\mathrm{out}}}^2 \le 4c_0^2\xi_n^2 + 2\lambda'.$$

Case where $\|\widehat{f}_{\mathrm{KT},\lambda'}\|_{\mathbf{k}} > 2$: Under such case, by assumption we have $\|\widehat{f}_{\mathrm{KT},\lambda'}\|_{\mathbf{k}} > 2 > 1 \ge \|f^\star\|_{\mathbf{k}}$. Thus, we may derive the following

$$\|f^\star\|_{\mathbf{k}} - \|\widehat{f}_{\mathrm{KT},\lambda'}\|_{\mathbf{k}} < 0 \quad\text{and}\quad \|f^\star\|_{\mathbf{k}} + \|\widehat{f}_{\mathrm{KT},\lambda'}\|_{\mathbf{k}} > 1,$$

which further implies the following inequality

$$\|f^\star\|_{\mathbf{k}}^2 - \|\widehat{f}_{\mathrm{KT},\lambda'}\|_{\mathbf{k}}^2 = \{\|f^\star\|_{\mathbf{k}} - \|\widehat{f}_{\mathrm{KT},\lambda'}\|_{\mathbf{k}}\}\{\|f^\star\|_{\mathbf{k}} + \|\widehat{f}_{\mathrm{KT},\lambda'}\|_{\mathbf{k}}\} \le \|f^\star\|_{\mathbf{k}} - \|\widehat{f}_{\mathrm{KT},\lambda'}\|_{\mathbf{k}}. \tag{121}$$

Further writing $\widehat{f}_{\mathrm{KT},\lambda'} = f^\star + \widehat{\Delta}_{\mathrm{KT}}$ and noting that $\|\widehat{\Delta}_{\mathrm{KT}}\|_{\mathbf{k}} - \|f^\star\|_{\mathbf{k}} \le \|\widehat{f}_{\mathrm{KT},\lambda'}\|_{\mathbf{k}}$ holds through triangle inequality, we may further refine (121) as

$$\|f^\star\|_{\mathbf{k}} - \|\widehat{f}_{\mathrm{KT},\lambda'}\|_{\mathbf{k}} \le 2\|f^\star\|_{\mathbf{k}} - \|\widehat{\Delta}_{\mathrm{KT}}\|_{\mathbf{k}} \le 2 - \|\widehat{\Delta}_{\mathrm{KT}}\|_{\mathbf{k}},$$

so that the basic inequality in (119) reduces to

$$\tfrac{1}{2}\|\widehat{\Delta}_{\mathrm{KT}}\|_{n_{\mathrm{out}}}^2 \le \left|\tfrac{1}{n_{\mathrm{out}}}\sum_{i=1}^{n_{\mathrm{out}}} v_i'\widehat{\Delta}_{\mathrm{KT}}(x_i')\right| + \lambda'\{2 - \|\widehat{\Delta}_{\mathrm{KT}}\|_{\mathbf{k}}\}. \tag{122}$$

We again introduce a technical event that controls the stochastic component of (122), which is

$$\mathcal{B}_{\mathrm{KT}} \triangleq \Bigg\{\exists g \in \mathcal{F}\setminus\mathbb{B}_2(\delta_n)\cap\{\|g\|_{\mathbf{k}} \ge 1\}: \tag{123}$$

$$\left|\tfrac{1}{n_{\mathrm{out}}}\sum_{i=1}^{n_{\mathrm{out}}} v_i'g(x_i')\right| > 4\xi_n\|g\|_{n_{\mathrm{out}}} + 2\xi_n^2\|g\|_{\mathbf{k}} + \tfrac{1}{4}\|g\|_{n_{\mathrm{out}}}^2\Bigg\},$$

for a star-shaped function class $\mathcal{F} \subset \mathcal{H}$.

By triangle inequality, we have $\|\widehat{\Delta}_{\mathrm{KT}}\|_{\mathbf{k}} \geq \|\widehat{f}_{\mathrm{KT},\lambda'}\|_{\mathbf{k}} - \|f^\star\|_{\mathbf{k}} > 1$, and on event $\mathcal{E}_{\mathrm{lower}}$ ($\subset \mathcal{E}_{\mathrm{good}}$), we have $\|\widehat{\Delta}_{\mathrm{KT}}\|_2 > \delta_n$. Thus, we may apply $g = \widehat{\Delta}_{\mathrm{KT}}$ to the event $\mathcal{B}_{\mathrm{KT}}^c$, and the resulting refined basic inequality is

$$\tfrac{1}{2}\|\widehat{\Delta}_{\mathrm{KT}}\|_{n_{\mathrm{out}}}^2 \leq 4\xi_n\|\widehat{\Delta}_{\mathrm{KT}}\|_{n_{\mathrm{out}}} + (2\xi_n^2 - \lambda')\|\widehat{\Delta}_{\mathrm{KT}}\|_{\mathbf{k}} + 2\lambda'$$
$$\leq 4\xi_n\|\widehat{\Delta}_{\mathrm{KT}}\|_{n_{\mathrm{out}}} + 2\lambda' \quad \text{on the event } \mathcal{B}_{\mathrm{KT}}^c \cap \mathcal{E}_{\mathrm{lower}}$$

where the second inequality is due to the assumption that $\lambda' \geq 2\xi_n^2$. We apply the quadratic formula (specifically, if $a, b \geq 0$ and $x^2 - ax - b \leq 0$, then $x \leq a^2 + b$) to obtain

$$\|\widehat{\Delta}_{\mathrm{KT}}\|_{n_{\mathrm{out}}}^2 \leq 4c_0^2\xi_n^2 + 2\lambda'.$$

Putting the pieces together, we have shown

$$\|\widehat{\Delta}_{\mathrm{KT}}\|_{n_{\mathrm{out}}}^2 \leq c(\xi_n^2 + \lambda') \quad \text{on the event } \mathcal{A}_{\mathrm{KT}}^c(\xi_n) \cap \mathcal{B}_{\mathrm{KT}}^c \cap \mathcal{E}_{\mathrm{lower}},$$

which is sufficient to recover (114).

**Recovering step (115)** We now upgrade events

$$\{\|\widehat{\Delta}_{\mathrm{KT}}\|_{n_{\mathrm{out}}}^2 \leq c(\xi_n^2 + \lambda')\} \quad \Longrightarrow \quad \{\|\widehat{\Delta}_{\mathrm{KT}}\|_n^2 \leq c'(\xi_n^2 + \lambda')\}$$

by exploiting the events $\mathcal{E}_{\mathrm{conc}} \cap \mathcal{E}_{\mathrm{KT},\delta}$ (subset of $\mathcal{E}_{\mathrm{good}}$) that were otherwise not used when recovering (114). For this end, the following result is a crucial ingredient, which shows that $\|\cdot\|_{n_{\mathrm{out}}}$ and $\|\cdot\|_n$ are essentially exchangeable with high-probability,

**Proposition 1** (Multiplicative guarantee for KT-COMPRESS++ with $\mathbf{k}_{\mathrm{RR}}$)**.** *Let $C_m \triangleq 1/\mu_m$ and suppose $\delta_n$ satisfies* (104). *Then on event $\mathcal{E}_{KT,\delta} \cap \mathcal{E}_{conc}$, where $\mathcal{E}_{KT,\delta}$ and $\mathcal{E}_{conc}$ are defined in* (24) *and* (110) *respectively, we have*

$$(1 - 4C_m \cdot \eta_{n,\mathbf{k}})\|g\|_n \leq \|g\|_{n_{\mathrm{out}}} \leq (1 + 4C_m \cdot \eta_{n,\mathbf{k}})\|g\|_n \tag{124}$$

*uniformly over all $g \in \mathcal{H}$ such that $\|g\|_2 > \delta_n$.*

See App. D.3 for the proof.

An immediate consequence of Prop. 1 is that

$$\{\|\widehat{\Delta}_{\mathrm{KT}}\|_{n_{\mathrm{out}}}^2 \leq c(\xi_n^2 + \lambda')\} \quad \Longrightarrow \quad \{\|\widehat{\Delta}_{\mathrm{KT}}\|_n^2 \leq c'(\xi_n^2 + \lambda')\}$$

on the event $\mathcal{E}_{\mathrm{KT},\delta} \cap \mathcal{E}_{\mathrm{conc}} \cap \mathcal{E}_{\mathrm{lower}}$,, which is sufficient to recover (115).

**Recovering step (116)** Our last step is to show

$$\{\|\widehat{\Delta}_{\mathrm{KT}}\|_n^2 \leq c'(\xi_n^2 + \lambda')\} \quad \Longrightarrow \quad \{\|\widehat{\Delta}_{\mathrm{KT}}\|_2^2 \leq c''(\xi_n^2 + \lambda')\}.$$

Such result can be immediately shown on the event $\mathcal{E}_{\mathrm{conc}}$ by observing that $\widehat{\Delta}_{\mathrm{KT}} \in \mathcal{H}$, by our assumption that $f^\star \in \mathcal{H}$ and by the definition

$$\widehat{f}_{\mathrm{KT},\lambda'} \in \mathrm{argmin}_{f \in \mathcal{H}(\mathbf{k})} L_{n_{\mathrm{out}}}(f) + \lambda'\|f\|_{\mathbf{k}}^2.$$

## D.2 Proof of claim (113)

It suffices to show the appropriate bounds for the following four probability terms

$$\mathbb{P}(\mathcal{A}_{\mathrm{KT}}(\xi_n)), \quad \mathbb{P}(\mathcal{B}_{\mathrm{KT}}), \quad \mathbb{P}(\mathcal{E}_{\mathrm{conc}}^c), \quad \mathbb{P}(\mathcal{E}_{\mathrm{KT},\delta}^c).$$

Fix the shorthand

$$B_{\mathcal{H}} \triangleq \|\mathbf{k}\|_\infty^2 R^2 < \infty.$$

We know from [10] that $\mathbb{P}(\mathcal{E}_{\mathrm{KT},\delta}^c | \mathcal{S}_{\mathrm{in}}) \leq \delta$ and then we may apply [29, Thm. 14.1] to obtain a high probability statement,

$$\mathbb{P}(\mathcal{E}_{\mathrm{conc}}^c) \leq e^{-c'n\delta_n^2/B_{\mathcal{H}}^2}. \tag{125}$$

Now we present two Lemmas that bound the $\mathbb{P}(\cdot \mid \mathcal{S}_{\mathrm{in}})$ probability of events $\mathcal{A}_{\mathrm{KT}}(\xi_n)$ and $\mathcal{B}_{\mathrm{KT}}$,

**Lemma 7** (Controlling bad event when $\|\widehat{f}_{\mathrm{KT},\lambda'}\|_{\mathbf{k}} \leq 2$). *Suppose $u \geq \xi_n$. Then for some constant $c > 0$,*

$$\mathbb{P}(\mathcal{A}_{\mathrm{KT}}(u) \mid \mathcal{S}_{\mathrm{in}}) \leq \delta + e^{-cn\delta_n^2/B_{\mathcal{H}}^2} + e^{-cnu^2/\widetilde{\sigma}^2} \tag{126}$$

*where $\widetilde{\sigma} = \sigma/\|f^\star\|_{\mathbf{k}}$.*

See App. D.4 for the proof. Note that by plugging in $\xi_n$ into (126) results in a probability that depends on $\mathcal{S}_{\mathrm{in}}$ (as $\xi_n$ depends on $\mathcal{S}_{\mathrm{in}}$). By invoking the definition of $\xi_n$, we may further refine the probability bound of $\mathcal{A}_{\mathrm{KT}}(\xi_n)$ by

$$\mathbb{P}(\mathcal{A}_{\mathrm{KT}}(\xi_n) \mid \mathcal{S}_{\mathrm{in}}) \leq \delta + e^{-cn\delta_n^2/B_{\mathcal{H}}^2} + e^{-cn\delta_n^2/\widetilde{\sigma}^2} \tag{127}$$

**Lemma 8** (Controlling bad event when $\|\widehat{f}_{\mathrm{KT},\lambda'}\|_{\mathbf{k}} > 2$). *For some constants $c, c' > 0$,*

$$\mathbb{P}(\mathcal{B}_{\mathrm{KT}} \mid \mathcal{S}_{\mathrm{in}}) \leq \delta + e^{-cn\delta_n^2/B_{\mathcal{H}}^2} + ce^{-n\xi_n^2/(c'\widetilde{\sigma}^2)} \tag{128}$$

*where $\widetilde{\sigma} = \sigma/\|f^\star\|_{\mathbf{k}}$.*

See App. D.5 for the proof. It is notable that $\xi_n$ in the probability bound of (128) contains a term $\varepsilon_n$ defined in (100) that is a function of $\mathcal{S}_{\mathrm{in}}$. Invoking the definition of $\xi_n$, we observe the probability upper bound (128) can be refined to

$$\mathbb{P}(\mathcal{B}_{\mathrm{KT}} \mid \mathcal{S}_{\mathrm{in}}) \leq \delta + e^{-cn\delta_n^2/B_{\mathcal{H}}^2} + ce^{-n\delta_n^2/(c'\widetilde{\sigma}^2)}, \tag{129}$$

which does not depend on $\mathcal{S}_{\mathrm{in}}$.

Putting the pieces together, we have the following probability bound for some constants $c, c' > 0$,

$$\mathbb{P}(\mathcal{E}_{\mathrm{good}}^c \mid \mathcal{S}_{\mathrm{in}}) \leq \mathbb{P}(\mathcal{A}_{\mathrm{KT}}(\xi_n) \mid \mathcal{S}_{\mathrm{in}}) + \mathbb{P}(\mathcal{B}_{\mathrm{KT}} \mid \mathcal{S}_{\mathrm{in}}) + \mathbb{P}(\mathcal{E}_{\mathrm{conc}}^c \mid \mathcal{S}_{\mathrm{in}}) + \mathbb{P}(\mathcal{E}_{\mathrm{KT},\delta}^c \mid \mathcal{S}_{\mathrm{in}})$$

$$\overset{(125)(127)(129)}{\leq} c\big\{\delta + e^{-c'n\delta_n^2/(B_{\mathcal{H}}^2 \wedge \widetilde{\sigma}^2)}\big\}$$

thereby implying $\mathbb{P}(\mathcal{E}_{\mathrm{good}}^c) \leq c''\big\{\delta + e^{-c'n\delta_n^2/(B_{\mathcal{H}}^2 \wedge \widetilde{\sigma}^2)}\big\}$ for some constant $c''$.

### D.3 Proof of Prop. 1: Multiplicative guarantee for KT-COMPRESS++ with $\mathbf{k}_{\mathrm{RR}}$

Fix $g \in \mathcal{H}$. Denote $\langle g, h \rangle = \int g(x)h(x)dx$ as the inner product in the $L^2$ sense. By Mercer's theorem [29, Cor. 12.26], the $\mathbf{k}$-norm of $g$ has a basis expansion $\|g\|_{\mathbf{k}}^2 = \sum_{i=1}^m \langle g, \phi_i \rangle^2/\lambda_i$ so that

$$\|g\|_{\mathbf{k}}^2 \leq \sum_{i=1}^m \langle g, \phi_i \rangle^2/\lambda_m = C_m\|g\|_2^2 \quad \text{since} \quad C_m = 1/\lambda_m. \tag{130}$$

The assumption $\|g\|_2 \geq \delta_n$ implies that on the event $\mathcal{E}_{\mathrm{conc}}$ (110), we have

$$\tfrac{1}{2}\delta_n \leq \|g\|_2 - \tfrac{1}{2}\delta_n \leq \|g\|_n \tag{131}$$

Moreover, $g$ must be a non-zero function. Note that $g^2 \in \mathcal{H}(\mathbf{k}_{\mathrm{RR}})$ (see App. F.3). Thus, we may apply Lem. 14 to $f_1 = f_2 = g$ and $a = 1, b = 0$ to obtain

$$\big|\|g\|_n^2 - \|g\|_{n_{\mathrm{out}}}^2\big| = \Big|\tfrac{1}{n}\sum_{i=1}^n g^2(x_i) - \tfrac{1}{n_{\mathrm{out}}}\sum_{i=1}^{n_{\mathrm{out}}} g^2(x_i')\Big| \leq \|g\|_{\mathbf{k}}^2 \cdot \eta_{n,\mathbf{k}}. \tag{132}$$

The LHS can be expanded as

$$\big|\|g\|_n^2 - \|g\|_{n_{\mathrm{out}}}^2\big| = \big|\|g\|_n - \|g\|_{n_{\mathrm{out}}}\big| \cdot \underbrace{\big|\|g\|_n + \|g\|_{n_{\mathrm{out}}}\big|}_{>0 \text{ by } (131)}.$$

Thus, we may rearrange (132) and combine with (130) to obtain

$$\big|\|g\|_n - \|g\|_{n_{\mathrm{out}}}\big| \leq \frac{C_m\|g\|_2^2}{\|g\|_n + \|g\|_{n_{\mathrm{out}}}} \cdot \eta_{n,\mathbf{k}}. \tag{133}$$

On event $\mathcal{E}_{\mathrm{conc}}$, we have

$$\|g\|_2^2 \overset{(110)}{\leq} (\tfrac{1}{2}\delta_n + \|g\|_n)^2 \overset{(i)}{\leq} \tfrac{\delta_n^2}{4} + \delta_n\|g\|_n + \|g\|_n^2. \tag{134}$$

Thus, we have

$$
\frac{\|g\|_2^2}{\|g\|_n+\|g\|_{n_{\mathrm{out}}}} \overset{(134)}{\le} \frac{\delta_n^2}{4\big|\|g\|_n+\|g\|_{n_{\mathrm{out}}}\big|} + \frac{\delta_n\|g\|_n}{\|g\|_n+\|g\|_{n_{\mathrm{out}}}} + \frac{\|g\|_n^2}{\|g\|_n+\|g\|_{n_{\mathrm{out}}}}
$$

$$
\overset{(131)}{\le} \frac{\delta_n^2}{2\delta_n} + \delta_n + \|g\|_n \cdot \frac{\|g\|_n}{\|g\|_n+\|g\|_{n_{\mathrm{out}}}}
$$

$$
\le \tfrac{3}{2}\delta_n + \|g\|_n
$$

$$
\overset{(131)}{\le} 3\|g\|_n + \|g\|_n = 4\|g\|_n. \tag{135}
$$

Using (135) to refine (133), we have on event $\mathcal{E}_{\mathrm{KT},\delta} \cap \mathcal{E}_{\mathrm{conc}}$:

$$
\big|\|g\|_n - \|g\|_{n_{\mathrm{out}}}\big| \le 4C_m \|g\|_n \cdot \eta_{n,\mathbf{k}}.
$$

With some algebra, this implies with probability at least $1 - \delta - \exp(-c'n\delta_n^2/B_{\mathcal{F}}^2)$:

$$
(1 - 4C_m \cdot \eta_{n,\mathbf{k}})\|g\|_n \le \|g\|_{n_{\mathrm{out}}} \le (1 + 4C_m \cdot \eta_{n,\mathbf{k}})\|g\|_n
$$

uniformly over all non-zero $g \in \mathcal{H}$ such that $\|g\|_2 > \delta_n$.

### D.4 Proof of Lem. 7: Controlling bad event when $\|\widehat{f}_{\mathrm{KT},\lambda'}\|_{\mathbf{k}} \le 2$

Recall $\mathcal{E}_{\mathrm{KT},\delta}$ and $\mathcal{E}_{\mathrm{conc}}$ defined by (24) and (110). Also recall that $\mathcal{E}_{\mathrm{KT},\delta} \cap \mathcal{E}_{\mathrm{conc}}$ combined with the assumption $\|g\|_2 \ge \delta_n$ invokes the event (124). Our aim is to show

$$
\mathcal{A}_{\mathrm{KT}}(u) \cap \mathcal{E}_{\mathrm{KT},\delta} \cap \mathcal{E}_{\mathrm{conc}} \subseteq \{Z_n(2u) \ge 2u^2\}, \quad \text{where} \quad Z_n(t) \triangleq \sup_{\substack{g \in \mathcal{F}: \\ \|g\|_n \le t}} \Big|\frac{\widetilde{\sigma}}{n}\sum_{i=1}^n w_i g(x_i)\Big| \tag{136}
$$

so that we have a probability bound

$$
\mathbb{P}(\mathcal{A}_{\mathrm{KT}}(u)) \le \mathbb{P}(\mathcal{E}_{\mathrm{KT},\delta}^c) + \mathbb{P}(\mathcal{E}_{\mathrm{conc}}^c) + \mathbb{P}(Z_n(2u) \ge 2u^2).
$$

The first RHS term can be bounded by $\delta$ (see (24)). The second RHS term can bounded by (125). The third term can be bounded by

$$
\mathbb{P}(Z_n(2u) \ge 2u^2) = \mathbb{P}\big(Z_n(u) \ge u^2/2 + u^2/2\big) \overset{(i)}{\le} \mathbb{P}\big(Z_n(u) \ge u\varepsilon_n/2 + u^2/2\big) \overset{(ii)}{\le} e^{-\frac{nu^2}{8\widetilde{\sigma}^2}},
$$

where (i) follows from our assumption that $u \ge \varepsilon_n$ and (ii) follows from applying generic concentration bounds on $Z_n(u)$ (see [29, Thm. 2.26, Eq. 13.66]). Putting together the pieces yields our desired probability bound (126).

**Proof of claim (136).** Consider the event $\mathcal{A}_{\mathrm{KT}}(u) \cap \mathcal{E}_{\mathrm{KT},\delta} \cap \mathcal{E}_{\mathrm{conc}}$. The norm equivalence established on the event $\mathcal{E}_{\mathrm{KT},\delta} \cap \mathcal{E}_{\mathrm{conc}}$ in Prop. 1 is an important ingredient throughout.

Let $g \in \mathcal{H}$ be the function that satisfies three conditions: $\|g\|_2 \ge \delta_n$, $\|g\|_{n_{\mathrm{out}}} \ge u$, and

$$
\left|\frac{1}{n_{\mathrm{out}}}\sum_{i=1}^{n_{\mathrm{out}}} v_i' g(x_i')\right| \ge 3\|g\|_{n_{\mathrm{out}}} u.
$$

Define the normalized function

$$
\widetilde{g} = u \cdot g/\|g\|_{n_{\mathrm{out}}}
$$

so that it satisfies $\|\widetilde{g}\|_{n_{\mathrm{out}}} = u$ and also

$$
\left|\frac{1}{n_{\mathrm{out}}}\sum_{i=1}^{n_{\mathrm{out}}} v_i' \widetilde{g}(x_i')\right| \ge 3u^2. \tag{137}
$$

By triangle inequality, the LHS of (137) can be further upper bounded by

$$
\left|\frac{1}{n_{\mathrm{out}}}\sum_{i=1}^{n_{\mathrm{out}}} v_i' \widetilde{g}(x_i')\right| \le \left|\frac{1}{n}\sum_{i=1}^n v_i \widetilde{g}(x_i)\right| + \frac{u}{\|g\|_{n_{\mathrm{out}}}}\left|\frac{1}{n}\sum_{i=1}^n v_i g(x_i) - \frac{1}{n_{\mathrm{out}}}\sum_{i=1}^{n_{\mathrm{out}}} v_i' g(x_i')\right|. \tag{138}
$$

Recall the chosen $g$ satisies $\|g\|_{n_{\mathrm{out}}} \geq u$. Observe that

$$v_i g(x_i) \overset{(1)}{=} (y_i - f^\star(x_i))g(x_i) = -f^\star(x_i)g(x_i) + y_i g(x_i),$$

so we may apply Lem. 14 with $f_1 = f^\star$, $f_2 = g$ and $a = -1, b = 1$. Thus, on the event $\mathcal{E}_{\mathrm{KT},\delta} \cap \mathcal{E}_{\mathrm{conc}}$, we have

$$\left| \tfrac{1}{n} \sum_{i=1}^n v_i g(x_i) - \tfrac{1}{n_{\mathrm{out}}} \sum_{i=1}^{n_{\mathrm{out}}} v_i' g(x_i') \right| \leq \|g\|_{\mathbf{k}} (\|f^\star\|_{\mathbf{k}} + 1) \cdot \eta_{n,\mathbf{k}}. \tag{139}$$

Thus, we may rearrange (138) and combine with (137) and (139) to obtain

$$\left| \tfrac{1}{n} \sum_{i=1}^n v_i \widetilde{g}(x_i) \right| \geq 3u^2 - \tfrac{u}{\|g\|_{n_{\mathrm{out}}}} \|g\|_{\mathbf{k}} (\|f^\star\|_{\mathbf{k}} + 1) \cdot \eta_{n,\mathbf{k}}$$

Note that

$$\frac{\|g\|_{\mathbf{k}}}{\|g\|_{n_{\mathrm{out}}}} = \frac{\|g\|_{\mathbf{k}}}{\|g\|_2} \cdot \frac{\|g\|_2}{\|g\|_n} \cdot \frac{\|g\|_n}{\|g\|_{n_{\mathrm{out}}}}.$$

We tackle each term in turn. First, $\frac{\|g\|_{\mathbf{k}}}{\|g\|_2} \overset{(130)}{\leq} \sqrt{C}_m$. Since we assume $\|g\|_2 \geq \delta_n$, we have $\frac{\|g\|_2}{\|g\|_n} \leq \frac{\delta_n/2 + \|g\|_n}{\|g\|_n} \overset{(131)}{\leq} 2$ on event $\mathcal{E}_{\mathrm{conc}}$; and $\frac{\|g\|_n}{\|g\|_{n_{\mathrm{out}}}} \overset{(144)}{\leq} 2$ on event $\mathcal{E}_{\mathrm{conc}} \cap \mathcal{E}_{\mathrm{KT},\delta}$. Taken together,

$$\frac{\|g\|_{\mathbf{k}}}{\|g\|_{n_{\mathrm{out}}}} \leq 4\sqrt{C}_m \quad \text{on event} \quad \mathcal{E}_{\mathrm{conc}} \cap \mathcal{E}_{\mathrm{KT},\delta}. \tag{140}$$

As $u \geq \xi_n \geq 4\sqrt{C}_m(\|f^\star\|_{\mathbf{k}} + 1)\eta_{n,\mathbf{k}}$ by assumption, we have therefore found $\widetilde{g}$ with norm $\|\widetilde{g}\|_{n_{\mathrm{out}}} = u$ satisfying

$$\left| \tfrac{1}{n} \sum_{i=1}^n v_i \widetilde{g}(x_i) \right| \geq 3u^2 - u^2 = 2u^2.$$

We may further show that

$$\|\widetilde{g}\|_n = \tfrac{u}{\|g\|_{n_{\mathrm{out}}}} \|g\|_n \leq u \tfrac{\|g\|_n}{\|g\|_{n_{\mathrm{out}}}} \leq u \cdot 2 \quad \text{on event} \quad \mathcal{E}_{\mathrm{conc}} \cap \mathcal{E}_{\mathrm{KT},\delta},$$

where the last inequality follows from the fact that $\|g\|_2 \geq \delta_n$ and by applying (144). So we observe

$$2u^2 \leq \left| \tfrac{1}{n} \sum_{i=1}^n v_i \widetilde{g}(x_i) \right| \leq \sup_{\|\widetilde{g}\|_n \leq 2u} \left| \tfrac{1}{n} \sum_{i=1}^n v_i \widetilde{g}(x_i) \right| = Z_n(2u)$$

### D.5 Proof of Lem. 8: Controlling bad event when $\|\widehat{f}_{\mathrm{KT},\lambda'}\|_{\mathbf{k}} > 2$

Our aim is to show for any $g \in \partial\mathcal{H}$ with $\|g\|_{\mathbf{k}} \geq 1$,

$$\left| \tfrac{1}{n_{\mathrm{out}}} \sum_{i=1}^{n_{\mathrm{out}}} v_i' g(x_i') \right| \leq 2\xi_n \|g\|_{n_{\mathrm{out}}} + 2\xi_n^2 \|g\|_{\mathbf{k}} + \tfrac{1}{16} \|g\|_{n_{\mathrm{out}}}^2 \quad \text{with high probability.}$$

Note that it is sufficient to prove our aim for $g \in \partial\mathcal{H}$ with $\|g\|_{\mathbf{k}} = 1$—by proving only for $g$ with $\|g\|_{\mathbf{k}} = 1$, then for any $h \in \partial\mathcal{H}$ with $\|h\|_{\mathbf{k}} \geq 1$, we may plug $g = h/\|h\|_{\mathbf{k}}$ into

$$\left| \tfrac{1}{n_{\mathrm{out}}} \sum_{i=1}^{n_{\mathrm{out}}} v_i' g(x_i') \right| \leq 2\xi_n \|g\|_{n_{\mathrm{out}}} + 2\xi_n^2 + \tfrac{1}{16} \|g\|_{n_{\mathrm{out}}}^2 \tag{141}$$

to recover the aim of interest. So without loss of generality, we show (141) for all $g$ such that $g \in \partial\mathcal{H}$ and $\|g\|_{\mathbf{k}} = 1$.

Let $\mathcal{B}_{\mathrm{KT}}$ denote the event where (141) is violated, i.e. there exists $g \in \partial\mathcal{H}$ with $\|g\|_{\mathbf{k}} = 1$ so that

$$\left| \tfrac{1}{n_{\mathrm{out}}} \sum_{i=1}^{n_{\mathrm{out}}} v_i' g(x_i') \right| > 3\xi_n \|g\|_{n_{\mathrm{out}}} + 2\xi_n^2 + \tfrac{1}{4} \|g\|_{n_{\mathrm{out}}}^2. \tag{142}$$

We prove the following set inclusion,

$$\mathcal{B}_{\mathrm{KT}} \cap \mathcal{E}_{\mathrm{KT},\delta} \cap \mathcal{E}_{\mathrm{conc}}$$

$$\subseteq \left\{ \exists\, g \in \partial\mathcal{H} \text{ s.t. } \|g\|_{\mathbf{k}} = 1 \text{ and } \left| \tfrac{1}{n} \sum_{i=1}^{n} v_i g(x_i) \right| > 2\varepsilon_n \|g\|_n + 2\varepsilon_n^2 + \tfrac{1}{16}\|g\|_n^2 \right\}, \tag{143}$$

where we know the RHS event of (143) has probability bounded by $ce^{-n\xi_n^2/(c'\tilde{\sigma}^2)}$ which is proven in [29, Lem. 13.23]. So the set inclusion (143) implies a bound over the event $\mathcal{B}_{\mathrm{KT}}$,

$$\mathbb{P}(\mathcal{B}_{\mathrm{KT}}) \leq \mathbb{P}(\mathcal{E}_{\mathrm{KT},\delta}^c) + \mathbb{P}(\mathcal{E}_{\mathrm{conc}}^c) + ce^{-n\xi_n^2/(c'\tilde{\sigma}^2)},$$

where $\mathbb{P}(\mathcal{E}_{\mathrm{KT},\delta}^c) \leq \delta$ by (24) and $\mathbb{P}(\mathcal{E}_{\mathrm{conc}}^c)$ by (125).

Choose $g$ so that $\|g\|_{\mathbf{k}} = 1$ and (142) holds. Condition $\|g\|_{\mathbf{k}} = 1$ as well as the condition (130) resulting from a finite rank kernel $\mathbf{k}$ implies $\delta_n \leq 1 \leq \|g\|_{\mathbf{k}} \leq \sqrt{C_m}\|g\|_2$. Invoke Prop. 1 for the choice of $g$ that satisfies $\|g\|_2 \geq \delta_n/\sqrt{C_m} \geq \delta_n$, so that on the event $\mathcal{E}_{\mathrm{KT},\delta} \cap \mathcal{E}_{\mathrm{conc}}$, we have the following norm equivalence,

$$\tfrac{1}{2}\|g\|_n \leq \|g\|_{n_{\mathrm{out}}} \leq \tfrac{3}{2}\|g\|_n \quad \text{for any } n \text{ such that } C_m \cdot \eta_{n,\mathbf{k}} \leq 1/18. \tag{144}$$

Then we have the following chain of inequalities, which holds on event $\mathcal{E}_{\mathrm{conc}} \cap \mathcal{E}_{\mathrm{KT},\delta}$

$$\left| \tfrac{1}{n}\sum_{i=1}^{n} v_i g(x_i) \right| \overset{(i)}{\geq} \left| \tfrac{1}{n_{\mathrm{out}}}\sum_{i=1}^{n_{\mathrm{out}}} v_i' g(x_i') \right| - \left| \tfrac{1}{n}\sum_{i=1}^{n} v_i g(x_i) - \tfrac{1}{n_{\mathrm{out}}}\sum_{i=1}^{n_{\mathrm{out}}} v_i' g(x_i') \right|$$

$$\overset{(ii)}{\geq} 3\xi_n\|g\|_{n_{\mathrm{out}}} + 2\xi_n^2 + \tfrac{1}{4}\|g\|_{n_{\mathrm{out}}}^2 - \|g\|_{\mathbf{k}}(\|f^\star\|_{\mathbf{k}} + 1) \cdot \eta_{n,\mathbf{k}}$$

$$\overset{(140)}{\geq} 3\xi_n\|g\|_{n_{\mathrm{out}}} + 2\xi_n^2 + \tfrac{1}{4}\|g\|_{n_{\mathrm{out}}}^2 - 4\sqrt{C_m}\|g\|_{n_{\mathrm{out}}}(\|f^\star\|_{\mathbf{k}} + 1) \cdot \eta_{n,\mathbf{k}}$$

$$\overset{(144)}{\geq} (\tfrac{3}{2}\xi_n - 2\sqrt{C_m}(\|f^\star\|_{\mathbf{k}} + 1) \cdot \eta_{n,\mathbf{k}})\|g\|_n + 2\xi_n^2 + \tfrac{1}{16}\|g\|_n^2, \tag{145}$$

where step (i) follows from triangle inequality and step (ii) follows from our assumption (142) to bound the first term and our approximation guarantee (139) to bound the second term. By definition of $\xi_n$, we have

$$\tfrac{3}{2}\xi_n - 2\sqrt{C_m}(\|f^\star\|_{\mathbf{k}} + 1) \cdot \eta_{n,\mathbf{k}} \geq 2\xi_n.$$

Using this to refine (145), we have

$$\left| \tfrac{1}{n}\sum_{i=1}^{n} v_i g(x_i) \right| \geq 2\xi_n\|g\|_n + 2\xi_n^2 + \tfrac{1}{16}\|g\|_n^2,$$

which directly implies the inclusion (143) as desired.

# E   Proof of Thm. 3: KT-KRR guarantee for infinite-dimensional RKHS

We state a more detailed version of the theorem:

**Theorem 5** (KT-KRR guarantee for infinite-dimensional RKHS, detailed). *Assume $f^\star \in \mathcal{H}(\mathbf{k})$ and Assum. 1 is satisfied. If $\mathbf{k}$ is* LOGGROWTH$(\alpha, \beta)$, *then for some constant $c$ (depending on $d, \alpha, \beta$), $\widehat{f}_{\mathrm{KT},\lambda'}$ with $\lambda' = \mathcal{O}(1/n_{\mathrm{out}})$ satisfies*

$$\|\widehat{f}_{\mathrm{KT},\lambda'} - f^\star\|_2^2 \leq c\left( \tfrac{\log^\alpha n}{n} + \tfrac{\sqrt{\log^\alpha n_{\mathrm{out}}}}{n_{\mathrm{out}}} \right) \cdot [\|f^\star\|_{\mathbf{k}} + 1]^2 \tag{146}$$

*with probability at least $1 - 2\delta - 2e^{-\frac{n\delta_n^2}{c_1(\|f^\star\|_{\mathbf{k}}^2 + \sigma^2)}}$.*

*If $\mathbf{k}$ is* POLYGROWTH$(\alpha, \beta)$ *with $\alpha \in (0, 2)$, then for some constant $c$ (depending on $d, \alpha, \beta$), $\widehat{f}_{\mathrm{KT},\lambda'}$ with $\lambda = \mathcal{O}(n_{\mathrm{out}}^{-\frac{2-\alpha}{2}})$ satisfies*

$$\|\widehat{f}_{\mathrm{KT},\lambda'} - f^\star\|_2^2 \leq c\|f^\star\|_{\mathbf{k}}^{\frac{2}{2+\alpha}} n^{-\frac{2}{2+\alpha}} + [\|f^\star\|_{\mathbf{k}} + 1]^2 n_{\mathrm{out}}^{-\frac{2-\alpha}{2}} \log n_{\mathrm{out}} + c' b^{\frac{4}{2+\alpha}} n^{-\frac{2}{2+\alpha}} \tag{147}$$

*with probability at least $1 - 2\delta - 2e^{-\frac{n\delta_n^2}{c_1(\|f^\star\|_{\mathbf{k}}^2 + \sigma^2)}}$.*

## E.1 Generic KT-KRR guarantee

We state a generic result for infinite-dimensional RKHS that only depends on the Rademacher and Gaussian critical radii as well as the KT approximation term, all introduced in App. C.

**Theorem 6** (KT-KRR). *Let $f^\star \in \mathcal{H}(\mathbf{k})$ and Assum. 1 is satisfied. Let $\delta_n$, $\varepsilon_n$ denote the solutions to* (100), (101), *respectively. Denote $\widehat{f}_{\mathrm{KT},\lambda'}$ with regularization parameter $\lambda' \geq 2\eta_{n,\mathbf{k}}$, where $\eta_{n,\mathbf{k}}$ is defined by* (102). *Then with probability at least $1 - 2\delta - 2e^{-\frac{n\delta_n^2}{c(\|f^\star\|_{\mathbf{k}}^2 + \sigma^2)}} - c_1 e^{-c_2 \frac{n\|f^\star\|_{\mathbf{k}}^2 \varepsilon_n^2}{\sigma^2}}$, we have*

$$\|\widehat{f}_{\mathrm{KT},\lambda'} - f^\star\|_2^2 \leq \mathbb{U}^{\mathrm{full}} + \mathbb{U}^{\mathrm{KT}}, \quad where$$
$$\mathbb{U}^{\mathrm{full}} \triangleq c(\varepsilon_n^2 + \lambda')[\|f^\star\|_{\mathbf{k}} + 1]^2 + c\delta_n^2 \quad and$$
$$\mathbb{U}^{\mathrm{KT}} \triangleq c \cdot \eta_{n,\mathbf{k}} [\|f^\star\|_{\mathbf{k}} + 1]^2.$$

See App. F for the proof. The term $\mathbb{U}^{\mathrm{full}}$ follows from the excess risk bound of FULL-KRR $\widehat{f}_{\mathrm{full},\lambda}$. The term $\mathbb{U}^{\mathrm{KT}}$ follows from our KT approximation. Clearly, the best rates are achieved when we choose $\lambda = 2\eta_{n,\mathbf{k}}$.

## E.2 Proof of explicit rates

The strategy for each setting is as follows:

1. Bound the Gaussian critical radius (101) using [29, Cor. 13.18], which reduces to finding $\varepsilon > 0$ satisfying the inequality

$$\sqrt{\tfrac{2}{n}} \sqrt{\sum_{j=1}^n \min\{\varepsilon^2, \hat{\mu}_j\}} \leq \beta\varepsilon^2, \quad where \quad \beta \triangleq \tfrac{\|f^\star\|_{\mathbf{k}}}{4\sigma} \tag{148}$$

   and $\hat{\mu}_1 \geq \hat{\mu}_2 \geq \ldots \geq \hat{\mu}_n \geq 0$ are the eigenvalues of the normalized kernel matrix $\mathbf{K}/n$, where $\mathbf{K}$ is defined by (5).

2. Bound the Rademacher critical radius (100) using [29, Cor. 14.5], which reduces to solving the inequality

$$\sqrt{\tfrac{2}{n}} \sqrt{\sum_{j=1}^\infty \min\{\delta^2, \mu_j\}} \leq \tfrac{\delta^2}{b}, \tag{149}$$

   where $(\mu_j)_{j=1}^\infty$ are the eigenvalues of the $\mathbf{k}$ according to Mercer's theorem [29, Thm. 12.20] and $b$ is the uniform bound on the function class.

3. Bound $\eta_{n,\mathbf{k}}$ (102) using the covering number bound $\mathcal{N}(\mathbb{B}_2^d(\mathfrak{R}_{\mathrm{in}}), \epsilon)$ from Assum. 3.

In the sequel, we make use of the following notation. Let

$$R_n \triangleq 1 + \sup_{x \in \mathcal{S}_{\mathrm{in}}} \|x_i\|_2 \stackrel{(103)}{=} 1 + \mathfrak{R}_{\mathrm{in}} \quad and \quad L_{\mathbf{k}}(r) \triangleq \tfrac{\mathfrak{C}_d}{\log 2} r^\beta$$

according to [16, Eq. 6], where $\mathfrak{C}_d$ is the constant that appears in Assum. 3.

### E.2.1 Proof of (147)

We begin by solving (148).

**Lemma 9** (Critical Gaussian radius for POLYGROWTH kernels). *Suppose Assum. 1 is satisfied and $\mathbf{k}$ is POLYGROWTH with $\alpha < 2$ as defined by Assum. 3. Then the Gaussian critical radius satisfies*

$$\varepsilon_n^2 \simeq \left(\tfrac{2c}{\|f^\star\|_{\mathbf{k}}/4\sigma}\right)^{\frac{4}{2+\alpha}} \left(2^{-\alpha} L_{\mathbf{k}}(R_n)(1 + \tfrac{32\alpha}{2-\alpha})\right)^{\frac{2}{2+\alpha}} \cdot n^{-\frac{2}{2+\alpha}}. \tag{150}$$

*Proof.* [16, Cor. B.1] implies that

$$\hat{\mu}_j \leq 4\left(\tfrac{L_{\mathbf{k}}(R_n)}{j-1}\right)^{\frac{2}{\alpha}} \quad for \ all \quad j > L_{\mathbf{k}}(R_n) + 1$$

Let $k$ be the smallest integer such that

$$k > L_{\mathbf{k}}(R_n) + 1 \quad \text{and} \quad 4\left(\frac{L_{\mathbf{k}}(R_n)}{k-1}\right)^{\frac{2}{\alpha}} \le \varepsilon^2.$$

By Assum. 1, $R_n$ is a constant, so the first inequality is easily satisfied for large enough $n$

$$k \ge 2^{-\alpha} L_{\mathbf{k}}(R_n)\varepsilon^{-\alpha} + 1. \tag{151}$$

Then

$$\frac{2}{\sqrt{n}}\sqrt{\sum_{j=1}^n \min\{\varepsilon^2, \hat{\mu}_j\}} \le \frac{2}{\sqrt{n}}\sqrt{k\varepsilon^2 + \sum_{j=k+1}^n 4\left(\frac{L_{\mathbf{k}}(R_n)}{j-1}\right)^{\frac{2}{\alpha}}}$$

$$\overset{(i)}{\le} \frac{2}{\sqrt{n}}\sqrt{k\varepsilon^2 + \frac{4L_{\mathbf{k}}(R_n)^{2/\alpha}}{2/\alpha-1}k^{1-2/\alpha}}$$

$$\overset{(151)}{\le} \frac{2}{\sqrt{n}}\sqrt{2^{-\alpha}L_{\mathbf{k}}(R_n)\varepsilon^{2-\alpha} + \frac{4\cdot 2^{2-\alpha}L_{\mathbf{k}}(R_n)}{2/\alpha-1}\varepsilon^{2-\alpha}},$$

where step (i) follows from the approximation

$$\sum_{j=k}^{n-1} 4\left(\frac{L_{\mathbf{k}}(R_n)}{j}\right)^{\frac{2}{\alpha}} \le 4L_{\mathbf{k}}(R_n)^{2/\alpha}\int_k^\infty t^{-2/\alpha}dt = 4L_{\mathbf{k}}(R_n)^{2/\alpha}\frac{1}{2/\alpha-1}k^{1-\frac{2}{\alpha}}.$$

To solve (148), it suffices to solve

$$\frac{2c}{\sqrt{n}}\sqrt{2^{-\alpha}L_{\mathbf{k}}(R_n)(1 + \frac{16}{2/\alpha-1})\varepsilon^{2-\alpha}} \le \beta\varepsilon^2$$

$$\implies \quad \varepsilon \ge \left(\frac{2c}{\beta}\right)^{\frac{2}{2+\alpha}}\left(2^{-\alpha}L_{\mathbf{k}}(R_n)(1 + \frac{32\alpha}{2-\alpha})\right)^{\frac{1}{2+\alpha}} \cdot n^{-\frac{1}{2+\alpha}}.$$

Since $\varepsilon_n$ is the smallest such solution to (148) by definition, we have (150) as desired. $\square$

We proceed to solve (149).

**Lemma 10.** *Suppose Assum. 1 is satisfied and $\mathbf{k}$ is* POLYGROWTH *with $\alpha < 2$ as defined by Assum. 3. Then the Rademacher critical radius satisfies*

$$\varepsilon_n^2 \simeq b^{\frac{4}{2+\alpha}}\left(2^{-\alpha}L_{\mathbf{k}}(R_n)(1 + \frac{32\alpha}{2-\alpha})\right)^{\frac{2}{2+\alpha}}n^{-\frac{2}{2+\alpha}}.$$

*Proof.* Thus, we can solve the following inequality

$$\sqrt{\frac{2}{n}}\sqrt{\sum_{j=1}^n \min\{\delta^2, \hat{\mu}_j\}} \le \frac{1}{b}\delta^2,$$

Following the same logic as in the proof of Lem. 9 but with $\beta = 1/b$ yields the desired bound. $\square$

Finally, it remains to bound (102). We have

$$\eta_{n,\mathbf{k}} = \frac{\mathfrak{a}^2}{n_{\text{out}}}(2 + \mathfrak{W}_{\mathbf{k}}(n, n_{\text{out}}, \delta, \mathfrak{R}_{\text{in}}, \frac{\mathfrak{a}}{n_{\text{out}}}))$$

$$\le \frac{\mathfrak{a}^2}{n_{\text{out}}}(2 + \sqrt{\log\left(\frac{n_{\text{out}}\log(n/n_{\text{out}})}{\delta}\right) \cdot \left[\log\left(\frac{1}{\delta}\right) + \log\mathcal{N}_{\mathbf{k}}(\mathbb{B}_2^d(\mathfrak{R}_{\text{in}}), \frac{\mathfrak{a}}{n_{\text{out}}})\right]})$$

$$\le \frac{\mathfrak{a}^2}{n_{\text{out}}}(2 + \sqrt{\log\left(\frac{n_{\text{out}}\log(n/n_{\text{out}})}{\delta}\right) \cdot \left[\log\left(\frac{1}{\delta}\right) + \mathfrak{C}_d\left(\frac{n_{\text{out}}}{\mathfrak{a}}\right)^\alpha(\mathfrak{R}_{\text{in}}+1)^\beta)\right]})$$

$$\le \frac{\mathfrak{a}^2}{n_{\text{out}}}\left(2 + \sqrt{\log\left(\frac{n_{\text{out}}\log(n/n_{\text{out}})}{\delta}\right)} \cdot \left[\sqrt{\log\left(\frac{1}{\delta}\right)} + \sqrt{\mathfrak{C}_d\frac{(\mathfrak{R}_{\text{in}}+1)^\beta}{\mathfrak{a}^\alpha}}n_{\text{out}}^{\frac{\alpha}{2}}\right]\right)$$

$$\le n_{\text{out}}^{\frac{\alpha}{2}-1} \cdot \mathfrak{a}^2\left(2 + \sqrt{\log\left(\frac{n_{\text{out}}\log(n/n_{\text{out}})}{\delta}\right)} \cdot \sqrt{\mathfrak{C}_d\frac{(\mathfrak{R}_{\text{in}}+1)^\beta}{\mathfrak{a}^\alpha}}\right)$$

for some universal positive constant $c$.

In summary, there exists positive constants $c_0, c_1, c_2$ such that

$$\varepsilon_n^2 \le c_0\left(\frac{\sigma}{\|f^\star\|_{\mathbf{k}}}\right)^{\frac{4}{2+\alpha}}n^{-\frac{2}{2+\alpha}} \quad \delta_n^2 \le c_1 b^{\frac{4}{2+\alpha}}n^{-\frac{2}{2+\alpha}} \quad \eta_{n,\mathbf{k}} \le c_2\mathfrak{a}^2 n_{\text{out}}^{-\frac{2-\alpha}{2}}\log n_{\text{out}}$$

Setting $\lambda' = c_2 \mathfrak{a}^2 n_{\mathrm{out}}^{-\frac{2-\alpha}{2}} \log n_{\mathrm{out}}$, we have

$$\|\widehat{f}_{\mathrm{KT},\lambda'} - f^\star\|_2^2 \le c\big(\varepsilon_n^2 + \lambda' + \eta_{n,\mathbf{k}}\big) \cdot \big[\|f^\star\|_{\mathbf{k}} + 1\big]^2 + c'\delta_n^2$$

$$\le c\|f^\star\|_{\mathbf{k}}^{\frac{2}{2+\alpha}} n^{-\frac{2}{2+\alpha}} + \big[\|f^\star\|_{\mathbf{k}} + 1\big]^2 n_{\mathrm{out}}^{-\frac{2-\alpha}{2}} \log n_{\mathrm{out}} + c'b^{\frac{4}{2+\alpha}} n^{-\frac{2}{2+\alpha}}.$$

### E.2.2  Proof of (146)

We begin by solving (148).

**Lemma 11** (Critical Gaussian radius for LOGGROWTH kernels). *Under Assum. 1 and LOGGROWTH version of Assum. 3, Gaussian critical radius satisfies*

$$\varepsilon_n^2 \simeq \frac{\sigma^2}{\|f^\star\|_{\mathbf{k}}^2} \frac{\log(2e \cdot \frac{\|f^\star\|_{\mathbf{k}}}{4\sigma}\sqrt{n})^\alpha}{n} \cdot L_{\mathbf{k}}(R_n) C''_\alpha$$

*for some constant $C''_\alpha$ that only depends on $\alpha$. where we ignore log-log factors.*

*Proof.* [16, Cor. B.1] implies that

$$\hat{\mu}_j \le 4\exp\left(2 - 2\Big(\tfrac{j-1}{L_{\mathbf{k}}(R_n)}\Big)^{\frac{1}{\alpha}}\right) \quad \text{for all} \quad j > L_{\mathbf{k}}(R_n) + 1$$

Let $k$ be the smallest integer such that

$$k > L_{\mathbf{k}}(R_n) + 1 \quad \text{and} \quad 4\exp\left(2 - 2\Big(\tfrac{j-1}{L_{\mathbf{k}}(R_n)}\Big)^{\frac{1}{\alpha}}\right) \le \varepsilon^2.$$

By Assum. 1, $R_n$ is a constant, so the first inequality is easily satisfied for large enough $n$

$$k \ge L_{\mathbf{k}}(R_n)\log\big(\tfrac{2e}{\varepsilon}\big)^\alpha + 1. \tag{152}$$

Thus, $k = \lceil L_{\mathbf{k}}(R_n)\log\big(\tfrac{2e}{\varepsilon}\big)^\alpha + 1\rceil$. Then

$$\tfrac{2}{\sqrt{n}}\sqrt{\textstyle\sum_{j=1}^n \min\{\varepsilon^2, \hat{\mu}_j\}} \le \tfrac{2}{\sqrt{n}}\sqrt{k\varepsilon^2 + \textstyle\sum_{j=k+1}^n 4\exp\left(2 - 2\Big(\tfrac{j-1}{L_{\mathbf{k}}(R_n)}\Big)^{\frac{1}{\alpha}}\right)} \tag{153}$$

Consider the following approximation:

$$\textstyle\sum_{\ell=k}^{n-1} 4\exp\left(2 - 2\Big(\tfrac{\ell}{L_{\mathbf{k}}(R_n)}\Big)^{\frac{1}{\alpha}}\right) \le 4e^2 \int_k^\infty e^{-\frac{2t}{L_{\mathbf{k}}(R_n)}^{1/\alpha}} dt = \int_{k-1}^\infty c^{t^{1/\alpha}} dt,$$

where $c \triangleq \exp(-(L_{\mathbf{k}}(R_n)/2)^{-1/\alpha}) \in (0,1)$. Defining $m \triangleq -\log c > 0$ and $k' \triangleq k - 1$, we have

$$\int_{k'}^\infty c^{t^{1/\alpha}} dt \le C_\alpha(k'b^{-1} + b^{\alpha-1}m^{-\alpha})e^{-mk'^{1/\alpha}}, \tag{154}$$

by Li et al. [16, Eq. 50], where $C_\alpha > 0$ is a constant satisfying $(x+y)^\alpha \le C_\alpha(x^\alpha + y^\alpha)$ for any $x, y > 0$ and $b$ is a known constant depending only on $\alpha$. Plugging in $k' = \lceil L_{\mathbf{k}}(R_n)\log\big(\tfrac{2e}{\varepsilon}\big)^\alpha\rceil$, we can bound the exponential by

$$e^{-mk'^{1/\alpha}} \le e^{-mL_k(R_n)^{1/\alpha}\log\big(\tfrac{2e}{\varepsilon}\big)} = \big(\tfrac{2e}{\varepsilon}\big)^{-mL_{\mathbf{k}}(R_n)^{1/\alpha}}.$$

Note that we can simplify the exponent by $-mL_{\mathbf{k}}(R_n)^{1/\alpha} = -(L_{\mathbf{k}}(R_n)/2)^{-1/\alpha}L_{\mathbf{k}}(R_n)^{1/\alpha} = -2^{1/\alpha}$. Note that $k' = k - 1 \ge L(R_n) = 2m^{-\alpha}$. Thus, we can absorb the $b^{\alpha-1}m^{-\alpha}$ term in (154) into $k$ and obtain the following bound

$$\textstyle\sum_{\ell=k}^{n-1} 4\exp\left(2 - 2\Big(\tfrac{\ell}{L_{\mathbf{k}}(R_n)}\Big)^{\frac{1}{\alpha}}\right) \le C'_\alpha k'\big(\tfrac{2e}{\varepsilon}\big)^{-2^{1/\alpha}},$$

where $C'_\alpha$ depends only on $\alpha$. Plugging this bound into (153), we have

$$\tfrac{2}{\sqrt{n}}\sqrt{\textstyle\sum_{j=1}^n \min\{\varepsilon^2, \hat{\mu}_j\}} \le \tfrac{2}{\sqrt{n}}\sqrt{k\varepsilon^2 + C'_\alpha k\big(\tfrac{\varepsilon}{2e}\big)^{2^{1/\alpha}}}$$

$$\overset{(152)}{\le} \tfrac{2c}{\sqrt{n}}\sqrt{L_{\mathbf{k}}(R_n)\log\big(\tfrac{2e}{\varepsilon}\big)^\alpha(\varepsilon^2 + C'_\alpha\big(\tfrac{\varepsilon}{2e}\big)^{2^{1/\alpha}})}$$

$$\le \tfrac{2c}{\sqrt{n}} \sqrt{L_{\mathbf{k}}(R_n) \log\big(\tfrac{2e}{\varepsilon}\big)^{\alpha} C''_{\alpha} \varepsilon^{2^{1/(1\vee\alpha)}}}$$

for some constant $C''_{\alpha}$ that only depends on $\alpha$ and universal positive constant $c$. To solve (148), it suffices to solve

$$\tfrac{2c}{\sqrt{n}} \sqrt{L_{\mathbf{k}}(R_n) \log\big(\tfrac{2e}{\varepsilon}\big)^{\alpha} C''_{\alpha} \varepsilon^{2^{1/(1\vee\alpha)}}} \le \beta\varepsilon^2,$$

which is implied by the looser bound

$$\tfrac{1}{\beta^2} \cdot \tfrac{4c^2}{n} \cdot L_{\mathbf{k}}(R_n) C''_{\alpha} \le \varepsilon^2 \log\big(\tfrac{2e}{\varepsilon}\big)^{-\alpha}.$$

The solution to (148) (up to log-log factors) is

$$\varepsilon \simeq \tfrac{\log(2e\cdot\beta\sqrt{n})^{\alpha/2}}{\sqrt{n}} \sqrt{\tfrac{4c^2}{\beta^2} \cdot L_{\mathbf{k}}(R_n) C''_{\alpha}}.$$

$\square$

We proceed to solve (149).

**Lemma 12** (Critical Gaussian radius for LOGGROWTH kernels)**.** *Under Assum. 1 and* LOGGROWTH *version of Assum. 3, the Rademacher critical radius satisfies*

$$\delta_n^2 \simeq b^2 \tfrac{\log(\tfrac{2e}{b}\cdot\sqrt{n})^{\alpha}}{n} \cdot L_{\mathbf{k}}(R_n) C''_{\alpha}.$$

*Proof.* Following the same logic as in the proof of Lem. 11 but with $\beta = 1/b$ yields the desired bound. $\square$

Finally, it remains to bound (102). We have

$$\eta_{n,\mathbf{k}} = \tfrac{\mathfrak{a}^2}{n_{\text{out}}}(2 + \mathfrak{W}_{\mathbf{k}}(n, n_{\text{out}}, \delta, \mathfrak{R}_{\text{in}}, \tfrac{\mathfrak{a}}{n_{\text{out}}}))$$

$$\le \tfrac{\mathfrak{a}^2}{n_{\text{out}}}\Big(2 + \sqrt{\log\Big(\tfrac{n_{\text{out}}\log(n/n_{\text{out}})}{\delta}\Big) \cdot \Big[\log\big(\tfrac{1}{\delta}\big) + \log\mathcal{N}_{\mathbf{k}}(\mathbb{B}_2^d(\mathfrak{R}_{\text{in}}), \tfrac{\mathfrak{a}}{n_{\text{out}}})\Big]}\Big)$$

$$\le \tfrac{\mathfrak{a}^2}{n_{\text{out}}}\Big(2 + \sqrt{\log\Big(\tfrac{n_{\text{out}}\log(n/n_{\text{out}})}{\delta}\Big) \cdot \Big[\log\big(\tfrac{1}{\delta}\big) + \mathfrak{C}_d \log\big(\tfrac{en_{\text{out}}}{\mathfrak{a}}\big)^{\alpha}(\mathfrak{R}_{\text{in}} + 1)^{\beta}\Big]}\Big)$$

for some universal positive constant $c$.

In summary, there exists universal positive constants $c_0, c_1, c_2$ such that

$$\varepsilon_n^2 \le c_0 \tfrac{\sigma^2}{\|f^\star\|_{\mathbf{k}}^2} \tfrac{\log(2e\cdot\tfrac{\|f^\star\|_{\mathbf{k}}}{4\sigma}\sqrt{n})^{\alpha}}{n} \quad \delta_n^2 \le c_1 b^2 \tfrac{\log(\tfrac{2e}{b}\cdot\sqrt{n})^{\alpha/2}}{n} \quad \eta_{n,\mathbf{k}} \le c_2 \tfrac{\mathfrak{a}}{n_{\text{out}}} \log\big(\tfrac{en_{\text{out}}}{\mathfrak{a}}\big)^{\alpha/2} \mathfrak{R}_{\text{in}}^{\beta/2}.$$

Setting $\lambda' = 2c_2 \tfrac{\mathfrak{a}}{n_{\text{out}}} \log\big(\tfrac{en_{\text{out}}}{\mathfrak{a}}\big)^{\alpha/2}$, we have

$$\|\widehat{f}_{\text{KT},\lambda'} - f^\star\|_2^2 \le c\big(\varepsilon_n^2 + \lambda' + \eta_{n,\mathbf{k}}\big) \cdot \big[\|f^\star\|_{\mathbf{k}} + 1\big]^2 + c'\delta_n^2$$

$$\le c \tfrac{\log(2e\cdot\tfrac{\|f^\star\|_{\mathbf{k}}}{4\sigma}\sqrt{n})^{\alpha}}{n} + \big[\|f^\star\|_{\mathbf{k}} + 1\big]^2 \tfrac{c}{n_{\text{out}}} \log\big(\tfrac{en_{\text{out}}}{\mathfrak{a}}\big)^{\alpha/2} + c'b^2 \tfrac{\log(\tfrac{2e}{b}\cdot\sqrt{n})^{\alpha/2}}{n}.$$

# F Proof of Thm. 6: KT-KRR

Our first goal is to bound the in-sample prediction error. We relate $\|\widehat{f}_{\text{KT},\lambda'} - f^\star\|_n^2$ to $\|\widehat{f}_{\text{full},\lambda} - f^\star\|_n^2$, where the latter quantity has well known properties from standard analyses of the KRR estimator (refer to [29]). Note that regularization parameter $\lambda'$ of KT based estimator $\widehat{f}_{\text{KT},\lambda'}$ is independently chosen from the regularization parameter $\lambda$ of the estimator based on original samples $\widehat{f}_{\text{full},\lambda}$. For $\widehat{f}_{\text{full},\lambda}$, we choose the regularization parameter

$$\lambda = 2\varepsilon_n^2, \tag{155}$$

which is known to yield optimal $L^2$ error rates.

Define the main event of interest,

$$\mathcal{E} \triangleq \{\|\widehat{f}_{\mathrm{KT},\lambda'} - f^\star\|_2^2 \leq c(\varepsilon_n^2 + \delta_n^2 + \lambda' + \eta_{n,\mathbf{k}})[\|f^\star\|_\mathbf{k} + 1]^2\}.$$

Our goal is to show $\mathcal{E}$ occurs with high probability. For that end, we introduce several additional events that are used throughout this proof.

For some constant $c >$, define the event of an appealing in-sample prediction error of $\widehat{f}_{\mathrm{KT},\lambda'}$,

$$\mathcal{E}_{\mathrm{KT},n}(t) \triangleq \left\{\|\widehat{f}_{\mathrm{KT},\lambda'} - f^\star\|_n^2 \leq c\big[t^2 + \lambda' + \eta_{n,\mathbf{k}}\big] \cdot (\|f^\star\|_\mathbf{k} + 1)^2\right\} \quad \text{for} \quad t \geq \varepsilon_n.$$

where $\eta_{n,\mathbf{k}}$ is defined in (102). Recall $\mathcal{E}_{\mathrm{KT},\delta}$ is the event where KT-COMPRESS++ succeeds as defined by (24).

Further as $f^\star$ and $\widehat{f}_{\mathrm{KT},\lambda'}$ are both in $\{f \in \mathcal{H} : \|f\|_\mathbf{k} \leq R\}$, we may deduce that all the functions under consideration satisfies $\|f\|_\infty \leq \|\mathbf{k}\|_\infty \|f\|_\mathbf{k} \leq \|\mathbf{k}\|_\infty R$ where $\|\mathbf{k}\|_\infty < \infty$. Accordingly, we define a uniform concentration event,

$$\mathcal{E}'_{\mathrm{conc}} \triangleq \{\sup_{f \in \mathcal{F}} \big| \|f\|_2^2 - \|f\|_n^2 \big| \leq \|f\|_2^2/2 + \delta_n^2/2\} \quad \text{where} \quad \mathcal{F} = \{f \in \mathcal{H} : \|f\|_\infty \leq 2\|\mathbf{k}\|_\infty R\}. \tag{156}$$

Event (156) is analogous to the event $\mathcal{E}_{\mathrm{conc}}$ previously defined in (110) when dealing with finite rank kernels.

We first show that

$$\mathcal{E}_{\mathrm{KT},n}(\varepsilon_n \vee \delta_n) \cap \mathcal{E}'_{\mathrm{conc}} \subseteq \mathcal{E}. \tag{157}$$

Notice that almost surely we have

$$\|\widehat{f}_{\mathrm{KT},\lambda'} - f^\star\|_\infty \leq 2\|\mathbf{k}\|_\infty R,$$

thereby implying

$$\|\widehat{f}_{\mathrm{KT},\lambda'} - f^\star\|_2^2 \leq 2 \|\widehat{f}_{\mathrm{KT},\lambda'} - f^\star\|_n^2 + \delta_n^2 \quad \text{on the event } \mathcal{E}'_{\mathrm{conc}}. \tag{158}$$

Next invoking the event $\mathcal{E}_{\mathrm{KT},n}(\varepsilon_n \vee \delta_n)$ along with (158), we have

$$\begin{aligned}
\|\widehat{f}_{\mathrm{KT},\lambda'} - f^\star\|_2^2 &\leq 2c[(\varepsilon_n \vee \delta_n)^2 + \lambda' + \eta_{n,\mathbf{k}}] \cdot (\|f^\star\|_\mathbf{k} + 1)^2 + \delta_n^2 \\
&\leq c(\varepsilon_n^2 + \delta_n^2 + \lambda' + \eta_{n,\mathbf{k}})[\|f^\star\|_\mathbf{k} + 1]^2.
\end{aligned}$$

which recovers the event of $\mathcal{E}$.

The remaining task is to show $\mathcal{E}$ is of high-probability, which amounts to showing events $\mathcal{E}_{\mathrm{KT},n}(t)$ and $\mathcal{E}'_{\mathrm{conc}}$ are of high-probability by reflecting on (157). From [29, Thm. 14.1], we may immediately derive

$$\mathbb{P}(\mathcal{E}'_{\mathrm{conc}}) \geq 1 - c_1 e^{-c_2 \frac{n\delta_n^2}{\|\mathbf{k}\|_\infty^2 R^2}}$$

for some constants $c_1, c_2 > 0$.

We further claim that

$$\mathbb{P}(\mathcal{E}_{\mathrm{KT},n}(t) \mid \mathcal{S}_{\mathrm{in}}) \geq 1 - \delta - e^{-\frac{nt^2}{c_0 \sigma^2}} - c_1 e^{-c_2 \frac{n\|f^\star\|_\mathbf{k}^2 t^2}{\sigma^2}} \tag{159}$$

for some constants $c_0, c_1, c_2 > 0$. Proof of claim (159) is deferred to App. F.1. Plugging in $t = \varepsilon_n \vee \delta_n$ into (159), and invoking inequality $\varepsilon_n \vee \delta_n \geq \delta_n$ so as to decouple the dependence on $\mathcal{S}_{\mathrm{in}}$, we have

$$\mathbb{P}(\mathcal{E}_{\mathrm{KT},n}(\varepsilon_n \vee \delta_n) \mid \mathcal{S}_{\mathrm{in}}) \geq 1 - \delta - e^{-\frac{n\delta_n^2}{c_0 \sigma^2}} - c_1 e^{-c_2 \frac{\|f^\star\|_\mathbf{k}^2 n\delta_n^2}{\sigma^2}}$$

which further implies

$$\mathbb{P}(\mathcal{E}_{\mathrm{KT},n}(\varepsilon_n \vee \delta_n)) \geq 1 - \delta - e^{-\frac{n\delta_n^2}{c_0 \sigma^2}} - c_1 e^{-c_2 \frac{\|f^\star\|_\mathbf{k}^2 n\delta_n^2}{\sigma^2}}.$$

Putting the pieces together, for some constants $c_0, c_1 > 0$, we have

$$\mathbb{P}(\mathcal{E}) \geq 1 - \delta - c_0 e^{-c_1 \frac{n\delta_n^2}{\sigma^2 \wedge (\sigma^2/\|f^\star\|_\mathbf{k}^2) \wedge (\|\mathbf{k}\|_\infty^2 R^2)}}. \tag{160}$$

Overall, (157) and (160) collectively yields the desired result.

### F.1 Proof of claim (159)

To prove claim (159), we introduce two new intermediary and technical events. For some positive constant $c_0$, define the event [6] when in-sample prediction error of $\widehat{f}_{\text{full},\lambda}$ is appealing

$$\mathcal{E}_{\text{full},n}(t) \triangleq \left\{ \|\widehat{f}_{\text{full},\lambda} - f^\star\|_n^2 \leq 3c_0\|f^\star\|_{\mathbf{k}}^2 t^2 \right\} \quad \text{for} \quad t \geq \varepsilon_n. \tag{161}$$

The second intermediary event, denoted as $\mathcal{E}_{\widehat{\Delta}_{\text{KT}}}(t)$, is the intersection of (169) and (170), which we do not elaborate here due to its technical nature—event $\mathcal{E}_{\widehat{\Delta}_{\text{KT}}}(t)$ plays an analogous role to $\mathcal{A}_{\text{KT}}^c \cap \mathcal{B}_{\text{KT}}^c$ defined in (123) and (136) respectively.

Our goal here is two-folds: first is to show

$$\{\mathcal{E}_{\text{full},n}(t) \cap \mathcal{E}_{\text{KT},\delta} \cap \mathcal{E}_{\widehat{\Delta}_{\text{KT}}}(t)\} \quad \Longrightarrow \quad \mathcal{E}_{\text{KT},n}(t)$$

and second is to prove the following bound

$$\mathbb{P}\left(\mathcal{E}_{\text{full},n}(t) \cap \mathcal{E}_{\text{KT},\delta} \cap \mathcal{E}_{\widehat{\Delta}_{\text{KT}}}(t) \mid \mathcal{S}_{\text{in}}\right) \geq 1 - \delta - e^{-\frac{nt^2}{c_0 \sigma^2}} - c_1 e^{-c_2 \frac{n\|f^\star\|_{\mathbf{k}}^2 t^2}{\sigma^2}},$$

from which (159) follows. Note that Wainwright [29, Thm. 13.17] show

$$\mathbb{P}(\mathcal{E}_{\text{full},n}(t)) \geq 1 - c_1 e^{-c_2 \frac{n\|f^\star\|_{\mathbf{k}}^2 t^2}{\sigma^2}}$$

for some constants $c_1, c_2 > 0$ and that $\mathbb{P}(\mathcal{E}_{\text{KT},\delta} \mid \mathcal{S}_{\text{in}}) \geq 1 - \delta$. So it remains to bound the probability of event $\mathcal{E}_{\widehat{\Delta}_{\text{KT}}}(t)$, which we show below.

Given $f$, define the following quantities

$$L_n(f) \triangleq \frac{1}{n}\sum_{i=1}^n (f^2(x_i) - 2f(x_i)y_i) + \frac{1}{n}\sum_{i=1}^n y_i^2 \quad \text{and}$$
$$L_{n_{\text{out}}}(f) \triangleq \frac{1}{n_{\text{out}}}\sum_{i=1}^{n_{\text{out}}} (f^2(x_i') - 2f(x_i')y_i') + \frac{1}{n}\sum_{i=1}^n y_i^2.$$

In the sequel, we repeatedly make use of the following fact: on event $\mathcal{E}_{\text{KT},\delta}$ defined in (24), we have

$$|L_n(f) - L_{n_{\text{out}}}(f)| \leq \left(\|f\|_{\mathbf{k}}^2 + 2\right) \cdot \eta_{n,\mathbf{k}} \quad \text{for all non-zero} \quad f \in \mathcal{H}. \tag{162}$$

The claim of (162) is deferred to the end of this section. Given $f$, we can show with some algebra that

$$L_n(f) = \frac{1}{n}\sum_{i=1}^n (f(x_i) - y_i)^2 = \|f - f^\star\|_n^2 - \frac{2}{n}\langle Z, \boldsymbol{\xi}\rangle + \frac{1}{n}\sum_{i=1}^n \xi_i^2, \tag{163}$$

where $Z \triangleq (f(x_1) - f^\star(x_1), \ldots, f(x_n) - f^\star(x_n))$ and $\boldsymbol{\xi} \triangleq (\xi_1, \ldots, \xi_n)$ are vectors in $\mathbb{R}^n$. Define the shorthands

$$\widehat{\Delta}_{\text{KT}} \triangleq \widehat{f}_{\text{KT},\lambda'} - f^\star \quad \text{and} \quad \widehat{\Delta}_{\text{full}} \triangleq \widehat{f}_{\text{full},\lambda} - f^\star.$$

In the sequel, we use the following shorthands:

$$Z_{full} \triangleq (\widehat{\Delta}_{\text{full}}(x_1), \ldots, \widehat{\Delta}_{\text{full}}(x_n)) \quad \text{and} \quad Z_{KT} \triangleq (\widehat{\Delta}_{\text{KT}}(x_1), \ldots, \widehat{\Delta}_{\text{KT}}(x_n)).$$

Now for the main argument to bound $\|\widehat{f}_{\text{KT},\lambda'} - f^\star\|_n^2$. When $\|\widehat{\Delta}_{\text{KT}}\|_n < t$, we immediately have $\|\widehat{\Delta}_{\text{KT}}\|_n^2 < t^2$, which implies (159). Thus, we may assume that $\|\widehat{\Delta}_{\text{KT}}\|_n \geq t$. Note that

$$\|\widehat{\Delta}_{\text{KT}}\|_n^2 \overset{(163)}{=} L_n(\widehat{f}_{\text{KT},\lambda'}) + \frac{2}{n}\langle Z_{KT}, \xi\rangle - \frac{1}{n}\sum_{i=1}^n \xi_i^2$$
$$= L_n(\widehat{f}_{\text{full},\lambda}) + \left[L_n(\widehat{f}_{\text{KT},\lambda'}) - L_n(\widehat{f}_{\text{full},\lambda})\right] + \frac{2}{n}\langle Z_{KT}, \xi\rangle - \frac{1}{n}\sum_{i=1}^n \xi_i^2.$$

Given the optimality of $\widehat{f}_{\text{full},\lambda}$ on the objective (3), we have

$$L_n(\widehat{f}_{\text{full},\lambda}) \leq \frac{1}{n}\sum_{i=1}^n \xi_i^2 + \lambda\left\{\|f^\star\|_{\mathbf{k}}^2 - \|\widehat{f}_{\text{full},\lambda}\|_{\mathbf{k}}^2\right\} \leq \frac{1}{n}\sum_{i=1}^n \xi_i^2 + \lambda\|f^\star\|_{\mathbf{k}}^2,$$

---

[6]Since the input points in $\mathcal{S}_{\text{in}}$ are fixed, the randomness in $\widehat{f}_{\text{full},\lambda}$ originates entirely from the randomness of the noise variables $\boldsymbol{\xi}$.

where the last inequality follows trivially from dropping the $-\|\widehat{f}_{\text{full},\lambda}\|_{\mathbf{k}}^2$ term. Thus,

$$\|\widehat{\Delta}_{\text{KT}}\|_n^2 = \tfrac{1}{n}\sum_{i=1}^n \xi_i^2 + \lambda\|f^\star\|_{\mathbf{k}}^2 + \left[L_n(\widehat{f}_{\text{KT},\lambda'}) - L_n(\widehat{f}_{\text{full},\lambda})\right] + \tfrac{2}{n}\langle Z_{KT}, \xi\rangle - \tfrac{1}{n}\sum_{i=1}^n \xi_i^2$$

$$\leq \tfrac{2}{n}\langle Z_{KT}, \xi\rangle + \lambda\|f^\star\|_{\mathbf{k}}^2 + \left[L_n(\widehat{f}_{\text{KT},\lambda'}) - L_n(\widehat{f}_{\text{full},\lambda})\right]. \tag{164}$$

Using standard arguments to bound the term $\tfrac{2}{n}\langle Z_{KT}, \xi\rangle$, we claim that on the event $\mathcal{E}_{\widehat{\Delta}_{\text{KT}}}$, we have

$$\|\widehat{\Delta}_{\text{KT}}\|_n^2 \leq ct^2(\|f^\star\|_{\mathbf{k}} + 1)^2 + c'\left[L_n(\widehat{f}_{\text{KT},\lambda'}) - L_n(\widehat{f}_{\text{full},\lambda})\right] \tag{165}$$

for some positive constants $c, c'$, and that $\mathbb{P}(\mathcal{E}_{\widehat{\Delta}_{\text{KT}}} \mid \mathcal{S}_{\text{in}}) \geq 1 - e^{-\frac{nt^2}{2\sigma^2}}$. We defer the proof of claim (165) to the end of this section.

Now we bound the stochastic term $\left[L_n(\widehat{f}_{\text{KT},\lambda'}) - L_n(\widehat{f}_{\text{full},\lambda})\right]$ in (165)—first observe the following decomposition:

$$L_n(\widehat{f}_{\text{KT},\lambda'}) - L_n(\widehat{f}_{\text{full},\lambda}) = \left(L_n(\widehat{f}_{\text{KT},\lambda'}) - L_{n_{\text{out}}}(\widehat{f}_{\text{KT},\lambda'})\right) + \left(L_{n_{\text{out}}}(\widehat{f}_{\text{KT},\lambda'}) - L_n(\widehat{f}_{\text{full},\lambda})\right).$$

On the event $\mathcal{E}_{\text{KT},\delta}$ (24), the first term in the display can be bounded by

$$L_n(\widehat{f}_{\text{KT},\lambda'}) - L_{n_{\text{out}}}(\widehat{f}_{\text{KT},\lambda'}) \overset{(162)}{\leq} (\|\widehat{f}_{\text{KT},\lambda'}\|_{\mathbf{k}}^2 + 2)\,\eta_{n,\mathbf{k}}.$$

Note that $\widehat{f}_{\text{KT},\lambda'}$ is the solution to the following optimization problem,

$$\min_{f\in\mathcal{H}(\mathbf{k})} L_{n_{\text{out}}}(f) + \lambda'\|f\|_{\mathbf{k}}^2,$$

so the second term in the display can be bounded by the following basic inequality

$$L_{n_{\text{out}}}(\widehat{f}_{\text{KT},\lambda'}) + \lambda'\|\widehat{f}_{\text{KT},\lambda'}\|_{\mathbf{k}}^2 \leq L_{n_{\text{out}}}(\widehat{f}_{\text{full},\lambda}) + \lambda'\|\widehat{f}_{\text{full},\lambda}\|_{\mathbf{k}}^2$$

$$\text{so that} \quad L_{n_{\text{out}}}(\widehat{f}_{\text{KT},\lambda'}) - L_{n_{\text{out}}}(\widehat{f}_{\text{full},\lambda}) \leq \lambda'\left\{\|\widehat{f}_{\text{full},\lambda}\|_{\mathbf{k}}^2 - \|\widehat{f}_{\text{KT},\lambda'}\|_{\mathbf{k}}^2\right\}.$$

Thus, on event $\mathcal{E}_{\text{KT},\delta}$, we have

$$L_n(\widehat{f}_{\text{KT},\lambda'}) - L_n(\widehat{f}_{\text{full},\lambda}) \leq (\|\widehat{f}_{\text{KT},\lambda'}\|_{\mathbf{k}}^2 + 2)\,\eta_{n,\mathbf{k}} + \lambda'\left\{\|\widehat{f}_{\text{full},\lambda}\|_{\mathbf{k}}^2 - \|\widehat{f}_{\text{KT},\lambda'}\|_{\mathbf{k}}^2\right\}$$

$$= 2\eta_{n,\mathbf{k}} + \lambda'\|\widehat{f}_{\text{full},\lambda}\|_{\mathbf{k}}^2 + \{\eta_{n,\mathbf{k}} - \lambda'\}\cdot\|\widehat{f}_{\text{KT},\lambda'}\|_{\mathbf{k}}^2$$

$$\overset{(i)}{\leq} 2\eta_{n,\mathbf{k}} + \lambda'\|\widehat{f}_{\text{full},\lambda}\|_{\mathbf{k}}^2 \overset{(ii)}{\leq} \lambda'(\|\widehat{f}_{\text{full},\lambda}\|_{\mathbf{k}}^2 + 1)$$

where steps (i) and (ii) both follow from the fact that $\lambda' \geq 2\eta_{n,\mathbf{k}}$ (see assumptions in Thm. 6). To bound $\|\widehat{f}_{\text{full},\lambda}\|_{\mathbf{k}}^2$, we use the following lemma:

**Lemma 13** (RKHS norm of $\widehat{f}_{\text{full},\lambda}$). *On event $\mathcal{E}_{\text{full},n}$ (161), we have the following bound*

$$\|\widehat{f}_{\text{full},\lambda}\|_{\mathbf{k}}^2 \leq c_0(\|f^\star\|_{\mathbf{k}} + 1)^2 \tag{166}$$

*for some constant $c_0 > 0$.*

See App. F.2 for the proof. Putting things together, we have

$$L_n(\widehat{f}_{\text{KT},\lambda'}) - L_n(\widehat{f}_{\text{full},\lambda}) \leq c\lambda'(\|f^\star\|_{\mathbf{k}} + 1)^2$$

for some constant $c$—substituting this bound into (165) yields

$$\|\widehat{f}_{\text{KT},\lambda'} - f^\star\|_n^2 \leq ct^2(\|f^\star\|_{\mathbf{k}} + 1)^2 + c'\lambda'(\|f^\star\|_{\mathbf{k}} + 1)^2,$$

for some constants $c, c'$, which directly implies (159), i.e. implying

$$\{\mathcal{E}_{\text{full},n}(t) \cap \mathcal{E}_{\text{KT},\delta} \cap \mathcal{E}_{\widehat{\Delta}_{\text{KT}}}(t)\} \implies \mathcal{E}_{\text{KT},n}(t).$$

**Proof of claim (162).** Given $f$, define the function

$$\ell'_f : \mathcal{X} \times \mathcal{Y} \to \mathbb{R}, \quad \text{where} \quad \ell'_f(x, y) \triangleq f^2(x) - 2y \cdot f(x) \tag{167}$$

and note that

$$L_n(f) - L_{n_{\text{out}}}(f) = \tfrac{1}{n} \sum_{i=1}^{n} \ell'_f(x_i, y_i) - \tfrac{1}{n_{\text{out}}} \sum_{i=1}^{n_{\text{out}}} \ell'_f(x'_i, y'_i).$$

We first prove a generic technical lemma:

**Lemma 14** (KT-COMPRESS++ approximation bound using $\mathbf{k}_{\text{RR}}$). *Suppose $f_1, f_2 \in \mathcal{H}(\mathbf{k})$ and $a, b \in \mathbb{R}$. Then the function*

$$\ell_{f_1, f_2} : \mathcal{X} \times \mathcal{Y} \to \mathbb{R}, \quad \text{where} \quad \ell_{f_1, f_2}(x, y) \triangleq a \cdot f_1(x) f_2(x) + b \cdot y f_1(x) \tag{168}$$

*lies in the RKHS $\mathcal{H}(\mathbf{k}_{\text{RR}})$. Moreover, on event $\mathcal{E}_{KT,\delta}$, we have*

$$\mathbb{P}_{\text{in}} \ell_{f_1, f_2} - \mathbb{Q}_{\text{out}} \ell_{f_1, f_2} \leq (|a| \cdot \|f_1\|_{\mathbf{k}} \|f_2\|_{\mathbf{k}} + |b| \cdot \|f_2\|_{\mathbf{k}}) \cdot \eta_{n, \mathbf{k}}.$$

*uniformly for all non-zero $f_1, f_2 \in \mathcal{H}(\mathbf{k})$.*

See App. F.3 for the proof. Applying the lemma with $f_1 \triangleq f$, $f_2 \triangleq g$ and $a = 1, b = -2$, we have

$$\mathbb{P}_{\text{in}} \ell'_f - \mathbb{Q}_{\text{out}} \ell'_f \leq (\|f\|_{\mathbf{k}}^2 + 2) \cdot \eta_{n, \mathbf{k}},$$

which combined with the observation (168) yields the desired claim.

**Proof of claim (165).** Case I: First suppose that $\|\widehat{\Delta}_{\text{KT}}\|_{\mathbf{k}} \leq 1$. Recall that $\|\widehat{\Delta}_{\text{KT}}\|_n \geq t \geq \varepsilon_n$ by assumption. Thus, we may apply [29, Lem. 13.12] to obtain

$$\tfrac{1}{n} \langle Z_{KT}, \boldsymbol{\xi} \rangle \leq 2 \|\widehat{\Delta}_{\text{KT}}\|_n t \quad \text{w.p. at least} \quad 1 - e^{-\frac{nt^2}{2\sigma^2}} \tag{169}$$

Plugging the above bound into (164), we have with probability at least $1 - e^{-\frac{nt^2}{2\sigma^2}}$:

$$\|\widehat{\Delta}_{\text{KT}}\|_n^2 \leq 4 \|\widehat{\Delta}_{\text{KT}}\|_n t + \lambda \|f^\star\|_{\mathbf{k}}^2 + \left[ L_n(\widehat{f}_{\text{KT}, \lambda'}) - L_n(\widehat{f}_{\text{full}, \lambda}) \right].$$

We can solve for $\|\widehat{\Delta}_{\text{KT}}\|_n$ using the quadratic formula. Specifically, if $a, b \geq 0$ and $x^2 - ax - b \leq 0$, then $x \leq a + \sqrt{b}$. Thus, we have with probability at least $1 - e^{-\frac{n\varepsilon_n^2}{2\sigma^2}}$:

$$\|\widehat{\Delta}_{\text{KT}}\|_n \leq a + \sqrt{b}, \quad \text{where}$$
$$a \triangleq 4t \quad \text{and}$$
$$b \triangleq \lambda \|f^\star\|_{\mathbf{k}}^2 + \left[ L_n(\widehat{f}_{\text{KT}, \lambda'}) - L_n(\widehat{f}_{\text{full}, \lambda}) \right].$$

Using the fact that $(a + \sqrt{b})^2 \leq 2a^2 + 2b$, we have with probability at least $1 - e^{-\frac{nt^2}{2\sigma^2}}$:

$$\|\widehat{f}_{\text{KT}, \lambda'} - f^\star\|_n^2 \leq 32t^2 + 2\lambda \|f^\star\|_{\mathbf{k}}^2 + 2 \left[ L_n(\widehat{f}_{\text{KT}, \lambda'}) - L_n(\widehat{f}_{\text{full}, \lambda}) \right]$$
$$\overset{(155)}{\leq} ct^2(\|f^\star\|_{\mathbf{k}} + 1)^2 + 2 \left[ L_n(\widehat{f}_{\text{KT}, \lambda'}) - L_n(\widehat{f}_{\text{full}, \lambda}) \right]$$

Case II: Otherwise, we may assume that $\|\widehat{\Delta}_{\text{KT}}\|_{\mathbf{k}} > 1$. Now we apply [29, Thm. 13.23] to obtain

$$\tfrac{1}{n} \langle Z_{KT}, \boldsymbol{\xi} \rangle \leq 2t \|\widehat{\Delta}_{\text{KT}}\|_n + 2t^2 \|\widehat{\Delta}_{\text{KT}}\|_{\mathbf{k}} + \tfrac{1}{16} \|\widehat{\Delta}_{\text{KT}}\|_n^2 \quad \text{w.p. at least} \quad 1 - c_1 e^{-\frac{nt^2}{c_2\sigma^2}}, \tag{170}$$

for some universal positive constants $c_1, c_2$. Plugging the above bound into (164) and collecting terms, we have with probability at least $1 - c_1 e^{-\frac{nt^2}{c_2\sigma^2}}$:

$$\tfrac{7}{8} \|\widehat{\Delta}_{\text{KT}}\|_n^2 \leq 4t \|\widehat{\Delta}_{\text{KT}}\|_n + 4t^2 \|\widehat{\Delta}_{\text{KT}}\|_{\mathbf{k}} + \lambda \|f^\star\|_{\mathbf{k}}^2 + \left[ L_n(\widehat{f}_{\text{KT}, \lambda'}) - L_n(\widehat{f}_{\text{full}, \lambda}) \right].$$

Solving for $\|\widehat{\Delta}_{\mathrm{KT}}\|_n$ using the quadratic formula, we have with probability at least $1 - c_1 e^{-\frac{nt^2}{c_2\sigma^2}}$:

$$\|\widehat{\Delta}_{\mathrm{KT}}\|_n \le a + \sqrt{b}, \quad \text{where}$$
$$a \triangleq \tfrac{32}{7}t \quad \text{and}$$
$$b \triangleq \tfrac{32}{7}t^2\|\widehat{\Delta}_{\mathrm{KT}}\|_{\mathbf{k}}^2 + \tfrac{8}{7}\lambda\|f^\star\|_{\mathbf{k}}^2 + \tfrac{8}{7}\Big[L_n(\widehat{f}_{\mathrm{KT},\lambda'}) - L_n(\widehat{f}_{\mathrm{full},\lambda})\Big].$$

Using the fact that $(a + \sqrt{b})^2 \le 2a^2 + 2b$, we have with probability at least $1 - c_1 e^{-\frac{nt^2}{c_2\sigma^2}}$:

$$\|\widehat{f}_{\mathrm{KT},\lambda'} - f^\star\|_n^2 \le 42t^2 + 10t^2\|\widehat{\Delta}_{\mathrm{KT}}\|_{\mathbf{k}}^2 + 2.3\lambda\|f^\star\|_{\mathbf{k}}^2 + 2.3\Big[L_n(\widehat{f}_{\mathrm{KT},\lambda'}) - L_n(\widehat{f}_{\mathrm{full},\lambda})\Big]$$
$$\overset{(i)}{\le} 42t^2 + 10t^2\|\widehat{\Delta}_{\mathrm{KT}}\|_{\mathbf{k}}^2 + 4.6t^2\|f^\star\|_{\mathbf{k}}^2 + 2.3\Big[L_n(\widehat{f}_{\mathrm{KT},\lambda'}) - L_n(\widehat{f}_{\mathrm{full},\lambda})\Big]$$
$$\overset{(155),(166)}{\le} c_3 t^2(\|f^\star\|_{\mathbf{k}} + 1)^2 + c_4\Big[L_n(\widehat{f}_{\mathrm{KT},\lambda'}) - L_n(\widehat{f}_{\mathrm{full},\lambda})\Big]$$

for some positive constants $c_3, c_4$, where step (i) follows from that fact that $\lambda = 2\varepsilon_n^2$ by (155).

## F.2 Proof of Lem. 13: RKHS norm of $\widehat{f}_{\mathrm{full},\lambda}$

Given the optimality of $\widehat{f}_{\mathrm{full},\lambda}$ on the objective (3), we have the following basic inequality

$$L_n(\widehat{f}_{\mathrm{full},\lambda}) + \lambda\|\widehat{f}_{\mathrm{full},\lambda}\|_{\mathbf{k}}^2 \le \tfrac{1}{n}\sum_{i=1}^n \xi_i^2 + \lambda\|f^\star\|_{\mathbf{k}}^2$$
$$\implies \|\widehat{f}_{\mathrm{full},\lambda}\|_{\mathbf{k}}^2 \le \|f^\star\|_{\mathbf{k}}^2 + \tfrac{1}{\lambda}\Big(\tfrac{1}{n}\sum_{i=1}^n \xi_i^2 - L_n(\widehat{f}_{\mathrm{full},\lambda})\Big).$$

Since $\|\widehat{f}_{\mathrm{full},\lambda} - f^\star\|_n^2 \ge 0$, we also have the trivial lower bound

$$L_n(\widehat{f}_{\mathrm{full},\lambda}) \overset{(163)}{=} \|\widehat{f}_{\mathrm{full},\lambda} - f^\star\|_n^2 - \tfrac{2}{n}\langle Z_{full}, \boldsymbol{\xi}\rangle + \tfrac{1}{n}\sum_{i=1}^n \xi_i^2$$
$$\ge -\tfrac{2}{n}\langle Z_{full}, \boldsymbol{\xi}\rangle + \tfrac{1}{n}\sum_{i=1}^n \xi_i^2.$$

Thus,

$$\|\widehat{f}_{\mathrm{full},\lambda}\|_{\mathbf{k}}^2 \le \|f^\star\|_{\mathbf{k}}^2 + \tfrac{1}{\lambda}\Big(\tfrac{2}{n}\langle Z_{full}, \xi\rangle\Big) \tag{171}$$

and it remains to bound $\tfrac{2}{n}\langle Z_{full}, \xi\rangle$.

Case I:   First, suppose that $\|\widehat{\Delta}_{\mathrm{full}}\|_{\mathbf{k}} > 1$. Then we may apply [29, Lem. 13.23] to obtain

$$\tfrac{1}{n}\langle Z_{full}, \xi\rangle \le 2\varepsilon_n\|\widehat{\Delta}_{\mathrm{full}}\|_n + 2\varepsilon_n^2\|\widehat{\Delta}_{\mathrm{full}}\|_{\mathbf{k}} + \tfrac{1}{16}\|\widehat{\Delta}_{\mathrm{full}}\|_n^2 \quad \text{w.p. at least} \quad 1 - c_1 e^{-\frac{n\varepsilon_n^2}{c_2\sigma^2}}.$$

Combining this bound with (171), we have with probability at least $1 - c_1 e^{-\frac{n\varepsilon_n^2}{c_2\sigma^2}}$:

$$\|\widehat{f}_{\mathrm{full},\lambda}\|_{\mathbf{k}}^2 \le \|f^\star\|_{\mathbf{k}}^2 + \tfrac{2\varepsilon_n^2}{\lambda}\|\widehat{\Delta}_{\mathrm{full}}\|_{\mathbf{k}} + \tfrac{2}{\lambda}\Big(2\varepsilon_n\|\widehat{\Delta}_{\mathrm{full}}\|_n + \tfrac{1}{16}\|\widehat{\Delta}_{\mathrm{full}}\|_n^2\Big)$$
$$\overset{(i)}{\le} \|f^\star\|_{\mathbf{k}}^2 + \tfrac{2\varepsilon_n^2}{\lambda}(\|\widehat{f}_{\mathrm{full},\lambda}\|_{\mathbf{k}} + \|f^\star\|_{\mathbf{k}}) + \tfrac{2}{\lambda}\Big(2\varepsilon_n\|\widehat{\Delta}_{\mathrm{full}}\|_n + \tfrac{1}{16}\|\widehat{\Delta}_{\mathrm{full}}\|_n^2\Big)$$
$$\overset{(155)}{=} \|f^\star\|_{\mathbf{k}}^2 + \|\widehat{f}_{\mathrm{full},\lambda}\|_{\mathbf{k}} + \|f^\star\|_{\mathbf{k}} + \tfrac{2}{\lambda}\Big(2\varepsilon_n\|\widehat{\Delta}_{\mathrm{full}}\|_n + \tfrac{1}{16}\|\widehat{\Delta}_{\mathrm{full}}\|_n^2\Big),$$

where step (i) follows from triangle inequality. Solving for $\|\widehat{f}_{\mathrm{full},\lambda}\|_{\mathbf{k}}$ using the quadratic formula, we have

$$\|\widehat{f}_{\mathrm{full},\lambda}\|_{\mathbf{k}}^2 \le 2 + \|f^\star\|_{\mathbf{k}}^2 + \|f^\star\|_{\mathbf{k}} + \tfrac{2}{\lambda}\Big(2\varepsilon_n\|\widehat{\Delta}_{\mathrm{full}}\|_n + \tfrac{1}{16}\|\widehat{\Delta}_{\mathrm{full}}\|_n^2\Big).$$

On the event $\mathcal{E}_{\mathrm{full},n}$ (161), we have $\|\widehat{\Delta}_{\mathrm{full}}\|_n \le c\|f^\star\|_{\mathbf{k}}\varepsilon_n$ for some positive constant $c$, which implies the claimed bound (166) after some algebra.

Case II(a):   Otherwise, assume $\|\widehat{\Delta}_{\text{full}}\|_{\mathbf{k}} \le 1$ and $\|\widehat{\Delta}_{\text{full}}\|_n \le \varepsilon_n$. Applying [29, Thm. 2.26] to the function $\sup_{\substack{\|g\|_{\mathbf{k}} \le 1 \\ \|g\|_n \le \varepsilon_n}} \left| \frac{1}{n} \sum_{i=1}^n \xi_i g(x_i) \right|$, we have

$$\frac{1}{n}\langle Z_{full}, \boldsymbol{\xi}\rangle \le \frac{\varepsilon_n^2}{2} \quad \text{w.p. at least} \quad 1 - e^{-\frac{n\varepsilon_n^2}{8\sigma^2}}$$

Combining this bound with (171), we obtain

$$\|\widehat{f}_{\text{full},\lambda}\|_{\mathbf{k}}^2 \le \|f^\star\|_{\mathbf{k}}^2 + \frac{1}{\lambda}\varepsilon_n^2 \overset{(155)}{=} \|f^\star\|_{\mathbf{k}}^2 + \frac{1}{2},$$

which immediately implies the claimed bound (166).

Case II(b):   Finally, assume $\|\widehat{\Delta}_{\text{full}}\|_{\mathbf{k}} \le 1$ and $\|\widehat{\Delta}_{\text{full}}\|_n > \varepsilon_n$. Applying [29, Lem. 13.12] with $u = \varepsilon_n$, we have

$$\frac{1}{n}\langle Z_{full}, \boldsymbol{\xi}\rangle \le 2\varepsilon^2 \quad \text{w.p. at least} \quad 1 - e^{-\frac{n\varepsilon_n^2}{2\sigma^2}}.$$

Combining this bound with (171), we obtain

$$\|\widehat{f}_{\text{full},\lambda}\|_{\mathbf{k}}^2 \le \|f^\star\|_{\mathbf{k}}^2 + \frac{4}{\lambda}\varepsilon_n^2 \overset{(155)}{=} \|f^\star\|_{\mathbf{k}}^2 + 2,$$

which immediately implies the claimed bound (166).

### F.3   Proof of Lem. 14: KT-COMPRESS++ approximation bound using $k_{\text{RR}}$

By Grünewälder [12, Lem. 4], $\ell_{f_1,f_2}$ lies in the RKHS $\mathcal{H}(\mathbf{k}_{\text{RR}})$, which is a direct sum of two RKHS:

$$\mathcal{H}(\mathbf{k}_{\text{RR}}) = \mathcal{H}(\mathbf{k}_1) \oplus \mathcal{H}(\mathbf{k}_2),$$

where $\mathbf{k}_1, \mathbf{k}_2 : \mathcal{Z} \times \mathcal{Z} \to \mathbb{R}$ are the kernels defined by

$$\mathbf{k}_1((x_1,y_1),(x_2,y_2)) \triangleq \mathbf{k}^2(x_1,x_2) \quad \text{and} \quad \mathbf{k}_2((x_1,y_1),(x_2,y_2)) \triangleq \mathbf{k}(x_1,x_2) \cdot y_1 y_2.$$

Applying Lem. 2 with

$$\mathcal{Z} = \mathcal{X} \times \mathcal{Y}, \quad \mathbf{k}_{\text{ALG}} = \mathbf{k}_{\text{RR}}, \quad \text{and} \quad \epsilon^\star = \frac{(\|\mathbf{k}\|_{\infty,\text{in}}^{1/2} + Y_{\max})^2}{n_{\text{out}}},$$

yields the following bound on event $\mathcal{E}_{\text{KT},\delta}$ (24):

$$\sup_{\substack{h \in \mathcal{H}(\mathbf{k}_{\text{RR}}): \\ \|h\|_{\mathbf{k}_{\text{RR}}} \le 1}} |(\mathbb{P}_{\text{in}} - \mathbb{Q}_{\text{out}})h| \le 2\epsilon^\star + \frac{\|\mathbf{k}_{\text{RR}}\|_{\infty,\text{in}}^{1/2}}{n_{\text{out}}} \cdot \mathfrak{W}_{\mathbf{k}_{\text{RR}}}(n, n_{\text{out}}, \delta, \mathfrak{R}_{\text{in}}, \epsilon^\star).$$

We claim that

$$\|\mathbf{k}_{\text{RR}}\|_{\infty,\text{in}}^{1/2} \le \|\mathbf{k}\|_{\infty,\text{in}} + Y_{\max}^2 \quad \text{and} \tag{172}$$

$$\log \mathcal{N}_{\mathbf{k}_{\text{RR}}}^\dagger(\mathcal{S}_{\text{in}}, \epsilon^\star) \le c \cdot \log \mathcal{N}_{\mathbf{k}}(\mathcal{S}_{\text{in}}, \tfrac{\|\mathbf{k}\|_{\infty,\text{in}}^{1/2} + Y_{\max}}{n_{\text{out}}}), \tag{173}$$

for some positive constant $c$, where $\mathcal{N}_{\mathbf{k}_{\text{RR}}}^\dagger$ is the cardinality of the cover of $\mathcal{B}_{\mathbf{k}_{\text{RR}}}^\dagger \triangleq \left\{ \ell_f' / \|\ell_f'\|_{\mathbf{k}_{\text{RR}}} : f \in \mathcal{H}(\mathbf{k}) \right\}$ for $\ell_f'$ defined by (167). Proof of the claims (172) and (173) are deferred to the end of this section. By definition of $\mathfrak{W}_{\mathbf{k}_{\text{RR}}}$, we have

$$\mathfrak{W}_{\mathbf{k}_{\text{RR}}}(n, n_{\text{out}}, \delta, \mathfrak{R}_{\text{in}}, \epsilon^\star) \overset{(29)}{\le} \sqrt{c} \cdot \mathfrak{W}_{\mathbf{k}}(n, n_{\text{out}}, \delta, \mathfrak{R}_{\text{in}}, \tfrac{\|\mathbf{k}\|_{\infty,\text{in}}^{1/2} + Y_{\max}}{n_{\text{out}}}) \triangleq \mathfrak{W}_{\mathbf{k}}'.$$

On event $\mathcal{E}_{\text{KT},\delta}$, we have

$$\sup_{\substack{h \in \mathcal{H}(\mathbf{k}_{\text{RR}}): \\ \|h\|_{\mathbf{k}_{\text{RR}}} \le 1}} |(\mathbb{P}_{\text{in}} - \mathbb{Q}_{\text{out}})h| \le \frac{2(\|\mathbf{k}\|_{\infty,\text{in}}^{1/2} + Y_{\max})^2}{n_{\text{out}}} + \frac{\|\mathbf{k}\|_{\infty,\text{in}} + Y_{\max}^2}{n_{\text{out}}} \cdot \mathfrak{W}_{\mathbf{k}}'$$

$$\overset{(i)}{=} \frac{\|\mathbf{k}\|_{\infty,\text{in}} + Y_{\max}^2}{n_{\text{out}}} \cdot [2 + \mathfrak{W}_{\mathbf{k}}'], \tag{174}$$

where step (i) follows from the fact that $(\|\mathbf{k}\|_{\infty,\text{in}}^{1/2} + Y_{\max})^2 \le 2(\|\mathbf{k}\|_{\infty,\text{in}} + Y_{\max}^2)$.

Since $f_1, f_2$ are non-zero, we have $\|\ell_{f_1,f_2}\|_{\mathbf{k}} > 0$. Thus, the function $h \triangleq \ell_f / \|\ell_f\|_{\mathbf{k}_{RR}} \in \mathcal{H}(\mathbf{k}_{RR})$ is well-defined and satisfies $\|h\|_{\mathbf{k}_{RR}} = 1$. Applying (174), we obtain

$$|\mathbb{P}_{in} h - \mathbb{Q}_{out} h| \leq \frac{\|\mathbf{k}\|_{\infty,in} + Y_{max}^2}{n_{out}} \cdot (2 + \mathfrak{W}'_{\mathbf{k}}) \quad \text{on event} \quad \mathcal{E}_{KT,\delta}.$$

Multiplying both sides by $\|\ell_f\|_{\mathbf{k}_{RR}}$ and noting that

$$\begin{aligned}
\|\ell_{f,g}\|_{\mathbf{k}_{RR}}^2 &= \|a \cdot f_1 f_2\|_{\widehat{\mathcal{H} \odot \mathcal{H}}}^2 + \|b \cdot f_2 \otimes \langle \cdot, 1 \rangle_{\mathbb{R}}\|_{\mathcal{H} \otimes \mathcal{R}}^2 \\
&\leq a^2 \|f_1\|_{\mathbf{k}}^2 \|f_2\|_{\mathbf{k}}^2 + b^2 \|f_2\|_{\mathbf{k}}^2 \\
&\leq (|a| \cdot \|f_1\|_{\mathbf{k}} \|f_2\|_{\mathbf{k}} + |b| \cdot \|f_2\|_{\mathbf{k}})^2,
\end{aligned}$$

we have on event $\mathcal{E}_{KT,\delta}$,

$$\mathbb{P}_{in} \ell_{f_1,f_2} - \mathbb{Q}_{out} \ell_{f_1,f_2} \leq (|a| \cdot \|f_1\|_{\mathbf{k}} \|f_2\|_{\mathbf{k}} + |b| \cdot \|f_2\|_{\mathbf{k}}) \cdot \frac{\|\mathbf{k}\|_{\infty,in} + Y_{max}^2}{n_{out}} \cdot (2 + \mathfrak{W}'_{\mathbf{k}}),$$

which directly implies the bound (162) after applying the shorthand (102).

**Proof of (172)** Define $Y_{max} \triangleq \sup_{y \in (\mathcal{S}_{in})_y} y$. We have

$$\begin{aligned}
\|\mathbf{k}_{ALG}\|_{\infty,in} &= \sup_{(x_1,y_1),(x_2,y_2) \in \mathcal{S}_{in}} \{\mathbf{k}(x_1,x_2)^2 + \mathbf{k}(x_1,x_2) \cdot y_1 y_2 + (y_1 y_2)^2\} \\
&\leq \sup_{x_1,x_2 \in (\mathcal{S}_{in})_x} \mathbf{k}(x_1,x_2)^2 + \sup_{x_1,x_2 \in (\mathcal{S}_{in})_x} \mathbf{k}(x_1,x_2) \cdot \sup_{y_1,y_2 \in (\mathcal{S}_{in})_y} y_1 y_2 \\
&\quad + \sup_{y_1,y_2 \in (\mathcal{S}_{in})_y} (y_1 y_2)^2 \\
&= \|\mathbf{k}\|_{\infty,in}^2 + \|\mathbf{k}\|_{\infty,in} \cdot Y_{max}^2 + Y_{max}^4 \\
&\leq \left(\|\mathbf{k}\|_{\infty,in} + Y_{max}^2\right)^2.
\end{aligned}$$

**Proof of (173)** Since $\mathcal{H}(\mathbf{k}_{RR})$ is a direct sum, we have

$$\log \mathcal{N}_{\mathbf{k}_{RR}}^\dagger(\mathcal{S}_{in}, \epsilon^\star) \leq \log \mathcal{N}_{\mathbf{k}_1}^\dagger(\mathcal{S}_{in}, \epsilon^\star/2) + \log \mathcal{N}_{\mathbf{k}_2}^\dagger(\mathcal{S}_{in}, \epsilon^\star/2), \tag{175}$$

where $\mathcal{N}_{\mathbf{k}_1}^\dagger$ and $\log \mathcal{N}_{\mathbf{k}_2}^\dagger$ are the covering numbers of $\mathcal{B}_{\mathbf{k}_1}^\dagger \triangleq \{f^2/\|f^2\|_{\mathbf{k}_1} : f \in \mathcal{H}(\mathbf{k})\}$ and $\mathcal{B}_{\mathbf{k}_2}^\dagger \triangleq \{f \otimes \langle \cdot, y \rangle_{\mathbb{R}}/\|f \otimes \langle \cdot, y \rangle_{\mathbb{R}}\|_{\mathbf{k}_2} : f \otimes \langle \cdot, y \rangle_{\mathbb{R}} \in \mathcal{H}(\mathbf{k}_2)\}$, respectively.

Note that

$$\begin{aligned}
\log \mathcal{N}_{\mathbf{k}_1}^\dagger(\mathcal{S}_{in}, \epsilon^\star) &\leq 2 \log \mathcal{N}_{\mathbf{k}}(\mathcal{S}_{in}, \epsilon^\star/(2\|\mathbf{k}\|_\infty^{1/2})) \\
&\leq 2 \log \mathcal{N}_{\mathbf{k}}(\mathcal{S}_{in}, (1 + \frac{Y_{max}}{\|\mathbf{k}\|_{\infty,in}^{1/2}}) \frac{\|\mathbf{k}\|_{\infty,in}^{1/2} + Y_{max}}{2n_{out}}) \\
&\leq 2 \log \mathcal{N}_{\mathbf{k}}(\mathcal{S}_{in}, \frac{\|\mathbf{k}\|_{\infty,in}^{1/2} + Y_{max}}{2n_{out}}).
\end{aligned}$$

Define a kernel on $\mathbb{R}$ by $\mathbf{k}_{\mathbb{R}}(y_1,y_2) \triangleq y_1 y_2$. When $\sup_{y \in (\mathcal{S}_{in})_y} |y| \leq Y_{max}$, we have

$$\mathcal{N}_{\mathbf{k}_{\mathbb{R}}}([-Y_{max}, Y_{max}], \epsilon) = \mathcal{O}(Y_{max}^2/\epsilon) \quad \text{for} \quad \epsilon > 0.$$

Similarly, note that

$$\begin{aligned}
\log \mathcal{N}_{\mathbf{k}_2}^\dagger(\mathcal{S}_{in}, \epsilon^\star) &\leq \log \mathcal{N}_{\mathbf{k}}(\mathcal{S}_{in}, \epsilon^\star/(\|\mathbf{k}\|_\infty^{1/2} + \|\mathbf{k}_{\mathbb{R}}\|_\infty^{1/2})) + \log \mathcal{N}_{\mathbf{k}_{\mathbb{R}}}(\mathcal{S}_{in}, \epsilon^\star/(\|\mathbf{k}\|_{\infty,in}^{1/2} + \|\mathbf{k}_{\mathbb{R}}\|_\infty^{1/2})) \\
&\lesssim \log \mathcal{N}_{\mathbf{k}}(\mathcal{S}_{in}, \frac{\|\mathbf{k}\|_{\infty,in}^{1/2} + Y_{max}}{n_{out}}) + \log\left(\frac{Y_{max}^2(\|\mathbf{k}\|_\infty^{1/2} + Y_{max})}{n_{out}}\right)
\end{aligned}$$

Substituting the above two log-covering number expressions into (175) yields

$$\log \mathcal{N}_{\mathbf{k}_{ALG}}(\mathcal{S}_{in}, \epsilon^\star) \lesssim 3 \log \mathcal{N}_{\mathbf{k}}(\mathcal{S}_{in}, \frac{\|\mathbf{k}\|_{\infty,in}^{1/2} + Y_{max}}{n_{out}}) + \log\left(\frac{Y_{max}^2(\|\mathbf{k}\|_{\infty,in}^{1/2} + Y_{max})}{n_{out}}\right).$$

$$\leq c \cdot \log \mathcal{N}_{\mathbf{k}}(\mathcal{S}_{in}, \frac{\|\mathbf{k}\|_{\infty,in}^{1/2} + Y_{max}}{n_{out}})$$

for some universal positive constant $c$.

# G Non-compact kernels satisfying Assum. 2

The boundedness, Lipschitz assumption, square-integrability, and (15) follow from [10, App. O] and [10, Rmk. 8]. Thus, it only remains to verify (16) and (17) for each kernel.

## G.1 Gaussian

For the kernel by $\mathbf{k}(x_1, x_2) \triangleq \exp(-\frac{\|x_1 - x_2\|^2}{2h^2})$, we have $\kappa(u) \triangleq \exp(-u^2/2)$.

**Verifying (16).** For any $j \geq 1$ and $x \in \mathcal{X}$, we have

$$
\begin{aligned}
\int_{\|z\| \in [(2^{j-1} - \frac{1}{2})h, (2^j + \frac{1}{2})h]} \mathbf{k}(x, x-z)dz &\leq \int_{\|z\| \geq (2^{j-1} - \frac{1}{2})h} \mathbf{k}(x, x-z)dz \\
&= (2\pi h^2)^{d/2} \mathbb{P}_{X \sim \mathcal{N}(0, h^2 \mathbf{I}_d)}\big[\|X\|_2 \geq (2^{j-1} - \tfrac{1}{2})h\big] \\
&= (2\pi h^2)^{d/2} \mathbb{P}_{X \sim \mathcal{N}(0, \mathbf{I}_d)}\big[\|X\|_2^2 \geq (2^{j-1} - \tfrac{1}{2})^2\big] \\
&= (2\pi h^2)^{d/2} \mathbb{P}_{X \sim \mathcal{N}(0, \mathbf{I}_d)}\big[\|X\|_2^2 - d \geq (2^{j-1} - \tfrac{1}{2})^2 - d\big].
\end{aligned}
\tag{176}
$$

By [15, Lem. 1], we have

$$
\mathbb{P}_{X \sim \mathcal{N}(0, \mathbf{I}_d)}\Big[\|X\|_2^2 - d > 2\sqrt{dt} + 2t\Big] \leq e^{-t} \quad \text{for any} \quad t \geq 0.
\tag{177}
$$

Define $t \triangleq 2d$ and $R \triangleq 2\sqrt{t}$. We can directly verify that $R^2 - d \geq 2\sqrt{dt} + 2t$. Thus, we may further upper bound (176) by

$$
\begin{aligned}
\mathbb{P}_{X \sim \mathcal{N}(0, \mathbf{I}_d)}\big[\|X\|_2^2 - d \geq (2^{j-1} - \tfrac{1}{2})^2 - d\big] &\leq \mathbb{P}_{X \sim \mathcal{N}(0, \mathbf{I}_d)}\Big[\|X\|_2^2 - d \geq 2\sqrt{dt} + 2t\Big] \\
&\overset{(177)}{\leq} e^{-R^2/4} \leq e^{-(2^{j-2} - \frac{1}{4})^2},
\end{aligned}
$$

whenever $R \geq 2\sqrt{2d}$, or equivalently, $j \geq \log_2(1 + 4\sqrt{2d})$. Under this regime, we have

$$
2^{j\beta} \int_{\|z\| \in [(2^{j-1} - \frac{1}{2})h, (2^j + \frac{1}{2})h]} \mathbf{k}(x, x-z)dz \leq 2^{j\beta}(2\pi h^2)^{d/2} e^{-(2^{j-1} - \frac{1}{4})^2} < C
$$

for some positive constant $C$ that does not depend on $j$ or $n$ as desired.

**Verifying (17).** Fixing $j \geq 1$, we have

$$
\begin{aligned}
(2^j + \tfrac{1}{2})^d 2^{j\beta} \kappa(2^{j-1} - 1) &\leq 2^{j(d+\beta)+d} \exp(-\tfrac{1}{2}(2^{j-1} - 1)^2) \\
&\leq \exp\big((\log 2)(j(d+\beta) + d) - (2^{j-1} - 1)^2)\big) \\
&\leq \exp\big(\max_{j \in [T]}\big\{(\log 2)(j(d+\beta) + d) - (2^{j-1} - 1)^2\big\}\big)
\end{aligned}
$$

Note that $(\log 2)(j(d + \beta) + d) - (2^{j-1} - 1)^2$ is concave over $j > 1$ and maximized at $j = \lfloor \log(1 + \sqrt{1 + 2(\beta + d)})\rfloor$. Treating $\int_0^{\frac{1}{2}} \kappa(u) u^{d-1} du$ as a constant, there exists $c_2$ such that (17) is satisfied for all positive integers $j$.

## G.2 Matérn

We consider the kernel $\mathbf{k}(x_1, x_2) \triangleq c_{\nu - \frac{d}{2}} \big(\frac{\|x_1 - x_2\|_2}{h}\big)^{\nu - \frac{d}{2}} K_{\nu - \frac{d}{2}}\big(\frac{\|x_1 - x_2\|_2}{h}\big)$ with $\nu > d$, where $K_a$ denotes the modified Bessel function of the third kind [30, Def. 5.10], and $c_b \triangleq \frac{2^{1-b}}{\Gamma(b)}$. We have $\mathbf{k}(x_1, x_2) = \kappa\big(\frac{\|x_1 - x_2\|_2}{h}\big) \triangleq \widetilde{\kappa}_{\nu - \frac{d}{2}}\big(\frac{\|x_1 - x_2\|_2}{h}\big)$, where

$$
\widetilde{\kappa}_b(u) \triangleq c_b u^b K_b(u).
$$

**Verifying (16).** Fix $j \in \{0, 1, \ldots, T\}$. Applying Jensen's inequality, we have

$$\sup_{x \in \mathcal{X}} \int_{\|z\| \in [(2^{j-1} - \frac{1}{2})h, (2^j + \frac{1}{2})h]} \mathbf{k}(x, x - z) dz \leq \sup_{x \in \mathcal{X}} \left( \int_{\|z\| \in [(2^{j-1} - \frac{1}{2})h, (2^j + \frac{1}{2})h]} \mathbf{k}^2(x, x - z) dx \right)^{\frac{1}{2}}$$

$$\leq \sup_{x \in \mathcal{X}} \left( \int_{\|z\| \geq (2^{j-1} - \frac{1}{2})h} \mathbf{k}^2(x, x - z) dx \right)^{\frac{1}{2}}$$

$$= \tau_{\mathbf{k}}((2^{j-1} - \tfrac{1}{2})h),$$

where $\tau_{\mathbf{k}}(\cdot)$ is the kernel tail decay defined by [10, Assum. 3]. We may use the same logic as in [10, App. O.3.6]—but ignoring the $A_{\nu, \gamma, d}$ term and making the substitutions, $a \leftarrow \nu - \frac{d}{2}$, $\gamma \leftarrow \frac{1}{h}$, and $\Gamma(\nu - 1) \leftarrow \Gamma(2\nu - 1)$—to obtain

$$\tau_{\mathbf{k}}^2(R) \leq h^d \cdot 2\pi \frac{2^{2-2\nu-d}}{\Gamma^2(\nu - \frac{d}{2})} \cdot \frac{\pi^{\frac{d}{2}}}{\Gamma(\frac{d}{2} + 1)} \cdot \Gamma(2\nu - 1) \exp(-\tfrac{R}{2h}) \quad \text{for} \quad R \geq \tfrac{2\nu - d}{\sqrt{2}} h.$$

Hence, we have

$$2^{j\beta} \sup_{x \in \mathcal{X}} \int_{\|z\| \in [(2^{j-1} - \frac{1}{2})h, (2^j + \frac{1}{2})h]} \mathbf{k}(x, x - z) dz \leq 2^{j\beta} C_{h,d,\nu} \exp(-\tfrac{1}{4}(2^{j-1} - \tfrac{1}{2})) = \mathcal{O}(1),$$

whenever $(2^{j-1} - \frac{1}{2})h \geq \frac{2\nu - d}{\sqrt{2}} h \Leftrightarrow j \geq 1 + \log(\frac{2\nu - d}{\sqrt{2}} + \frac{1}{2})$ for some constant $C_{h,d,\nu}$ that does not depend on $j$.

**Verifying (17).** Following similar logic as in [10, App. O.3.1], we have

$$\widetilde{\kappa}_a(u) \leq \min\{1, \sqrt{2\pi} c_a u^{a-1} \exp(-\tfrac{u}{2})\} \quad \text{for} \quad u \geq 2(a - 1).$$

Fixing $j \geq 1$, we have

$$(2^j + 1)^d 2^{j\beta} \kappa(2^{j-1} - 1) \leq 2^{j(d+\beta)+d} \min\{1, \sqrt{2\pi} c_{\nu - \frac{d}{2}} (2^{j-1} - 1)^{\nu - \frac{d}{2} - 1} \exp(-\tfrac{1}{2}(2^{j-1} - 1))\}.$$

Note that when $\nu - \frac{d}{2} - 1 < 1$, the RHS can be rewritten as $C_{\beta,d}(2^{j-1})^{d+\beta} \exp(\frac{1}{2} 2^{j-1})$ for some constant $C_{\beta,d}$ that doesn't depend on $j$. When $\nu - \frac{d}{2} - 1 \geq 1$, the RHS can be upper-bounded by $C'_{\beta,d}(2^{j-1})^{j(\nu + \beta + \frac{d}{2} - 1)} \exp(-\frac{1}{2} 2^{j-1})$ for some constant $C'_{\beta,d}$ that doesn't depend on $j$. Observe that the function $t^b e^{-t/2}$ attains its maximum at $t = 2b$. Hence, the RHS can be bounded by a constant that does not depend on $n$ as desired.

