# OpenReview forum: "Supervised Kernel Thinning"
_NeurIPS.cc/2024/Conference — NeurIPS 2024 poster_

### Official Review · Reviewer_QHUc · 2024-07-12

**Soundness:** 3
**Presentation:** 3
**Contribution:** 3
**Rating:** 7
**Confidence:** 3

**Summary:**

This paper applies the kernel thinning approach to non-parametric regression, with two kinds of estimators (NW and KRR) considered. This approach speeds up the computational efficiency by using carefully chosen coresets for approximation. Theoretical results on the approximation error are established. Numerical experiments include synthetic data and the well-known California housing dataset.

**Strengths:**

1. The paper is well organized and written, with enough background and clear motivation
2. Solid theoretical results are developed for the proposed method
3. Certain experimental results are given to justify the proposed method

**Weaknesses:**

1. Despite the well developed theory, I am not fully convinced yet by the significance/contribution of this method - I am willing to stand corrected if more comparisons can be made with other existing methods for efficient non-parametric regression.
2. The application are only to toy examples or standard small datasets - e.g. for the Californian housing dataset with n = 20,640, doing non-parametric regression without thinning also seems fine.
3. In the experiments, the baseline is chosen to be NW / KRR with naive thinning - some more competitive, state-of-the-art baselines could have been chosen.

**Questions:**

1. In the beginning of Sec 3, the NW and KRR estimators are restricted to kernels taking the form of (4) and (6). What are the specific reasons to make these restrictions? And to see how restrictive/general this is, could you give some examples of commonly used kernels that fall under this category?
2. The theory makes the sub-Gaussian tail decay assumption on the kernel, which would exclude many commonly used kernels such as the Matern kernel. How would the theory work out, and how would the rates change if this assumption is generalized to sub-Weibull tail decay, or say a bound on the Orlicz-norm?
3. The method (or at least the theory) is developed for one-dimensional response data. How could this be generalized to multi-dimensional response data, where issues with computational efficiency becomes more significant?
4. In the discussion section, it is mentioned that the rates on the excess risk could be sub-optimal. Do you have a sense of what rate could be minimax optimal?

**Limitations:**

See weaknesses and questions.

---

> ### Author Rebuttal · Authors · 2024-08-06
>
> Thank you for the time you’ve spent reviewing our work and for your thoughtful feedback. We now address each of your questions and concerns in turn.
>
> $\blacktriangleright$ $\textbf{Comparisons to existing methods for efficient non-parametric regression}$
> - _"The application are only to toy examples or standard small datasets..."_
> - _"In the experiments, the baseline is chosen to be NW / KRR with naive thinning..."_
>
> Thank you for this suggestion.
> - We have added an experiment on the SUSY dataset ($d=18, N=5\times 10^6$) from Diaz et al. (2023). We sample $4\times 10^6$ points to use for training. (Full KRR is now infeasible as we can only store 3k-4k columns at a time, e.g., on a machine with 100 GB RAM).
> - We include the **RPCholesky preconditioning method** from Diaz et al. (2023) and the **FALKON method** from Rudi et al. (2017) as SOTA baselines. As a sanity check, we also include Conjugate Gradient method (without preconditioning).
> - In Fig. 2 of the attached PDF, we have added RPCholesky-NW (where $S_{NW}$ are the pivot points from RPCholesky) and Restricted KRR with RPCholesky for landmark selection (see box plots in red). **While it outperforms the KT method in the KRR setting, RPCholesky is less suitable for speeding up the NW estimator as shown in Fig. 2a of the attached PDF.**
>
> $\newline$
>
> $\blacktriangleright$ $\textbf{Kernel assumptions}$
> - _"In the beginning of Sec 3..."_
> - _"The theory makes the sub-Gaussian tail decay assumption on the kernel..."_
>
> Thank you for raising this question.
> - Our theory in fact permits a large class of base kernels. This strength is borrowed partially from the Kernel Thinning results of Dwivedi & Mackey (2021).
> - In our revision, we will use $\mathbf{k}$ exclusively to denote the base kernel and use $\mathbf{k} _{ALG}$ to denote the kernel for thinning. Given a valid base kernel $\mathbf{k}$, we can construct $\mathbf{k} _{NW}$ and $\mathbf{k} _{RR}$ following (4) and (6), respectively.
> - For NW, we only showed the results when the kernel is compactly supported or admits sub-Gaussian tails (for ease of presentation); **remarkably, the same results hold when the kernel has sub-exponential tails or poly-tails as well.** In fact, $\mathbf{k}$ can be any radial kernel $\mathbf{k}(x_1,x_2) := \kappa(\lVert x_1-x_2\rVert_2/h)$ such that $\kappa(u)$ is a bounded, L-Lipschitz function and decays (at any rate). This includes kernels with sub-Weibull tail decay. The kernel density estimation guarantees from Dwivedi & Mackey (2021), which we use in our analysis, indeed use an Orlicz bound.
> - For KRR, $\mathbf{k}$ can be any kernel such that the log-covering number satisfies the PolyGrowth (e.g., finitely-times differentiable kernels) or LogGrowth (e.g., analytic kernels, finite-rank kernels) property (see Assump. 4). For KRR, our results do degrade with increasingly less smooth kernels.
>
> $\newline$
>
> $\blacktriangleright$ $\textbf{Multi-dimensional response data}$
> - _"The method (or at least the theory) is developed for one-dimensional response data..."_
>
> The reviewer’s comment on multi-dimensional response data has direct and interesting applications for speeding up multivariate response regression problems.
>
> - Roughly speaking, KT with input kernel $\mathbf{k}_{ALG}$ provides a coreset that well approximates the averages of functions in its RKHS $\mathcal H$.
> - Dwivedi & Mackey (2022), Thm. 4 showed that thinning with $\mathbf{k} _{ALG} := \mathbf{k} _1 + \mathbf{k} _2$ produces a coreset that simultaneously well approximates averages for any $ f\in \mathcal{H}(\mathbf{k} _1)$ or $f\in \mathcal{H}(\mathbf{k} _2)$ (aka the "aggregate kernel trick").
>     - We use the aggregate kernel trick to tackle multi-variate regression via one-dimensional kernel thinning on an aggregated kernel.
> - To be precise, let $f  = (f _1, …, f _m)$ be multi-dimensional response function, and for all $j = 1, …, m$, $f _j : R^d \to R$ are coordinate functions in some RKHS $\mathcal{H}( \mathbf{k} _j)$ generated by some base kernel $\mathbf{k}(j)$.
> - Now define the aggregated kernel $\mathbf{k} _{ALG}((x_1,y_1),(x_2,y_2)) := \sum _{i=1}^{m} w _j \mathbf{k}(j) ((x _1,y _{1,j}),(x _2,y _{2,j}))$ (for NW and use $\mathbf{k} _{RR}(j)$ instead of $\mathbf{k} _{NW}(j)$ for KRR), where weights $w _j$ reflect prior beliefs.
> - Then $\mathbf{k} _{ALG}$-kernel thinned core-sets are shared across all coordinate functions, that also balances information across all coordinates.
> - Notably, aggregate kernel tricks have been used in Generalized Kernel Thinning by Dwivedi & Mackey (2022) for improving integration error guarantees and Compress-then-Test by Domingo-Enrich et al. (2023) for increasing power of kernel-based hypothesis tests. Here we use it for speeding multi-variate non-parameteric regression.
>
> $\newline$
>
> $\blacktriangleright$ $\textbf{Minimax optimal rates}$
> - _"In the discussion section, it is mentioned that the rates on the excess risk could be sub-optimal..."_
>
> We will include the expanded discussion in the revision:
> - The following is classically known (see Tsybakov (2009)):
>   - When $f^\star$ belongs to a Hölder class with smoothness $\beta > 0$, the minimax optimal rate is $\mathcal{O}(n^{-\frac{2\beta }{2\beta + d}} )$.
>   - When $f^\star$ belongs to a Sobolev space with smoothness $\nu=1/\alpha$ (for $\alpha\in (0,2)$), the minimax optimal rate is $\mathcal{O}(n^{-\frac{2}{2+\alpha}})$.
> - As shown by Dwivedi & Mackey (2021), KT produces a coreset that achieve the minimax rates (as if all $n$ points were used) for a large class of kernel density estimation tasks—e.g., employing any radial kernel $\mathbf{k}(x_1,x_2) = \kappa( \lVert x_1-x_2\rVert _2 / h)$ with bounded, L-Lipschitz $\kappa: \mathbb{R} \to \mathbb{R}$. However, our current proof techniques do not yield a minimax rate for supervised learning task. We believe a modified proof technique should yield the same rate as obtained by $n$ points for some class of kernels.

---

> > ### Author Response · Authors · 2024-08-12
> > **Further response**
> >
> > We thank Reviewer QHUc for your comments, which we believe have been addressed in our response. Please let us know if you have any other questions that we can address.​⬤

---

> > > ### Comment · Reviewer_QHUc · 2024-08-12
> > >
> > > Thank you to the authors for the detailed reply. Most of my concerns and questions have been addressed, and I will increase my rating accordingly.

---

### Official Review · Reviewer_PSUT · 2024-07-15

**Soundness:** 3
**Presentation:** 3
**Contribution:** 2
**Rating:** 7
**Confidence:** 2

**Summary:**

The authors speed up two non-parametric regression estimators, Nadaraya-Watson (NW) and Kernel Ridge Regression (KRR), by using a kernel thinning (KT) technique to compress the input data in a way that preserves important statistical properties. They include a theoretical analysis proving that that KT-based regression estimators are computational more efficient and also exhibit improved statistical efficiency over i.i.d. subsampling of the training data. They also include empirical validation of their method on simulated and real-world data.

**Strengths:**

Good presentation with a solid analytical analysis providing theoretical guarantees. Overall well-organized paper.
The proposed method of combining KT with the estimators is novel (to my knowledge) and shows potential.

**Weaknesses:**

The empirical analysis is limited. The shortcomings on the real-world data are not adequately addressed (see also limitations).

**Questions:**

I have no particular questions.

**Limitations:**

The comparison with other large-scale kernel methods is limited to RPCholesky; a more comprehensive comparison and overview would be nice.

Performance/utility of the proposed method on real-world data remains unclear. The results obtained on real-world dataset in section 4.2 are not convincing, and the authors acknowledge the problem of kernel mis-specification but do not elaborate or provide any resolution. Btw, there is an incomplete sentence in lines 283-284.

---

> ### Author Rebuttal · Authors · 2024-08-06
>
> Thank you for the time you’ve spent reviewing our work and for your thoughtful feedback. We now address each of your questions and concerns in turn.
>
> $\blacktriangleright$ $\textbf{Comparison with large-scale kernel methods}$
> - _"The comparison with other large-scale kernel methods is limited to RPCholesky"_
>
> Thank you for this suggestion. We have added new results in the global response pdf.
> - It includes an experiment on the SUSY dataset ($d=18, N=5\times 10^6$) from Diaz et al. (2023). We sample $4\times 10^6$ points to use for training. (Full KRR is now infeasible as we can only store 3k-4k columns at a time, e.g., on a machine with 100 GB RAM).
> - We include the _RPCholesky preconditioning method_ from Diaz et al. (2023) and the _FALKON method_ from Rudi et al. (2017) as SOTA baselines. As a sanity check, we also include the _Conjugate Gradient method_ (without preconditioning).
> - In Fig. 2 of the attached PDF, we have added RPCholesky-NW (where $S_{NW}$ are the pivot points from RPCholesky) and Restricted KRR with RPCholesky for landmark selection (see box plots in red). While it outperforms the KT method in the KRR setting, **RPCholesky is less suitable for speeding up the NW estimator as shown in Fig. 2a of the attached PDF and is outperformed by KT-NW.**
>
> $\newline$
>
> $\blacktriangleright$ $\textbf{Kernel mis-specificiation}$
> - _"the authors acknowledge the problem of kernel mis-specification but do not elaborate or provide any resolution"_
>
> We will elaborate our discussion in the revision.
> - The supervised KT guarantees apply when $f^\star$ lies in the RKHS $\mathcal{H}(\mathbf{k})$, where $\mathbf{k}$ is the base kernel.
> - In practice, choosing a good kernel $\mathbf{k}$ is indeed a challenge _common to all prior work_.
> - Our framework is friendly to recent developments in kernel selection to handle this problem:
>    - Dwivedi & Mackey (2022), Cor. 1 provide integration-error guarantees for KT when $f^\star \notin \mathcal{H}(\mathbf{k}).
>    - Moreover,  there are recent results on finding the best kernel (in testing; Dominigo-Enrich et al. (2023), Sec. 4.2).
>    - Radhakrishnan et al. (2024) introduce Recursive Feature Machines, which use a parameterized kernel $\mathbf{k} _{M}(x_1,x_2) := \exp(-\frac{(x_1-x_2)^\top M (x_1-x_2)}{2\sigma^2})$, and propose an efficient method to learn the matrix parameter $M$ via the average gradient outer product estimator. An exciting future direction would be to combine these parameterized (or "learned") kernels with our proposed KT methods for non-parametric regression.
> - Formally combining these results, potentially with the recursive feature machines of Radhakrishnan et al. (2024) to build a practical strategy with theoretical guarantees is an exciting future direction.

---

> > ### Comment · Reviewer_PSUT · 2024-08-12
> >
> > Thank you for your detailed response! Best of luck to the authors.

---

### Official Review · Reviewer_Vy1C · 2024-07-29

**Soundness:** 2
**Presentation:** 2
**Contribution:** 2
**Rating:** 5
**Confidence:** 2

**Summary:**

After the rebuttal, I have updated my score from 3 to 5.

----
This paper provides a meta-algorithm based on Kernel Thinning for non-parametric regression, in particular, the Nadaraya-Watson (NW) regression, and the Kernel Ridge Regression (KRR).

The idea is to run NW or KRR on a thinned coreset by KT. The core of the method is to choose a suitable kernel for which the KT assumptions are satisfied. The paper introduces to meta-kernels that takes a base kernel of features x and produces the kernel over (x,y) suitable for regression tasks.

The paper provides the theoretical guarantees for kernel thinned NW and kernel thinned KRR.

In empirical study, the paper includes simulation results and real data results which demonstrate the proposed method's strength over full computation and standard thnnning in accuracy and time; however, the SOTA RPCholesky method seems to have a better empirical performance than kernel thinned KRR.

**Strengths:**

- The paper in general is easy to read

- The idea to introduce kernel thinning to non-parametric regression methods is novel.

- Provides theoretical guarantees of the method under suitable assumptions.

**Weaknesses:**

- The empirical advantage of the proposed kernel thinned KRR over RPCholesky is not clear. In fact, from figure 5, RPCholesky has slighly higher training time compared to kernel thinned KRR, however, a much lower test MSE.

- Some technical background and discussion can be made more clear. See questions.

- Writing can be improved. See questions.

**Questions:**

- There is not an algorithmic description for Kernel Thinning algorithm.

- Eq 4 and 6: why do you choose the kernel in this way? In particular, can you comment on the part $k(x_1,x_2) y_1, y_2$. Does the performance of the kernel rely on the relationship of $y$ and $x$, and if so, how? Also, in eq 6, is this a typo $k^2(x_1,x_2)^2$?

- Notation suggestion: instead of $S_{KT}$ for KT-NW, could use $S_{NW}$ to match the use of $K_RR$ in KT-KRR part.

- Sec 4.1: are the kernels listed in 1.2.3 used as the base kernel to construct the kernel for thinning, or they are the kernel for thinning?

- Fig 2: Right panel: why not include RPCholesky?

- Fig 2: why si the full not performing the best?

- Fig 3 and Fig 4: why not use the function in (11)?

- Fig 3 and Fig 4: what is the unit of y axis? Are they all in the same unit?

- Sec 4.2: what is the conclusion from Fig 5?

- Sec 4.3: line 283-284: "In this example (and ....)" incomplete sentence.

- It'd be good to include Fig 5 in the main text instead of appendix.

**Limitations:**

- KT-KRR does not demonstrate an empirical advantage over RPCholesky. Missing a discussion on this point.

- Missing a discussion on potential failure cases of the method, though there is  a discussion on kernel mis-specification and potential way to deal with it.

---

> ### Author Rebuttal · Authors · 2024-08-06
>
> Thank you for the time you’ve spent reviewing our work and for your thoughtful feedback. We address each of your concerns below.
>
> $\blacktriangleright$ $\textbf{Comparison with RPCholesky}$
> - _"The empirical advantage..."_
> - _"KT-KRR does not demonstrate..."_
>
> Thank you for raising these points. The main focus of this work was to build a stepping stone towards bridging two literatures: distribution compression and non-parametric regression, by designing algorithms and developing theory. Tuning to optimize empirical performance, e.g., by combining the pre-conditioning step of RPCholesky with KT, is a great future direction.
>
> As another sanity check, we have added an experiment on the SUSY dataset ($d=18, N=5\times 10^6$) from Diaz et al. (2023). KT-KRR's (22%) performance lies between RPCholesky (20%) and ST-KRR (22.7%). Thinning 4 million points with our method took only 1.7 seconds on a single CPU core, with further speed-ups to be gained from parallelizing on a GPU.
>
> - _"Fig 2: Right panel: why not include RPCholesky?"_
>
> In Fig. 2b of the attached PDF, we have added Restricted KRR with RPCholesky for landmark selection. In Fig. 2a of the attached PDF, we have added RPCholesky-NW, where $S_{NW}$ are the pivot points from RPCholesky.
>
> $\newline$
>
> $\blacktriangleright$ $\textbf{Choice of kernel}$
> - _"Eq 4 and 6: why do you choose the kernel in this way?..."_
> - _"Sec 4.1: ... the kernels listed in 1.2.3 used... are the kernel for thinning?"_
>
> This is a great question central to the novelty of our work while bridging the two literatures. Directly applying generic kernels would not work in theory and our experiments confirm that. Overall the choice we make are informed by the following points:
> - Roughly speaking, KT with input kernel $\mathbf{k}_{ALG}$ provides a coreset that well approximates the averages of functions in its RKHS $\mathcal H$.
> - Dwivedi & Mackey (2022), Thm. 4 showed that thinning with $\mathbf{k} _{ALG} := \mathbf{k} _1 + \mathbf{k} _2$ produces a coreset that simultaneously well approximates averages for any $ f\in \mathcal{H}(\mathbf{k} _1)$ or $f\in \mathcal{H}(\mathbf{k} _2)$ (aka the "aggregate kernel trick").
> - The NW estimator is a ratio of averages: an average of $f(x)=k(x, x_0)$ in the denominator and an average of $f _{num}(x,y)=\langle y,1 \rangle k(x,x_0)$ for the numerator.
>     - $f _{num}$ lies in the RKHS corresponding to the kernel $\mathbf{k} _{num}((x_1,y_1),(x_2,y_2)) := \mathbf{k}(x_1,x_2) y_1 y_2$, so thinning with this kernel provides good approximation of the numerator.
>    - Applying aggregate kernel trick to $\mathbf{k} _1 = \mathbf{k}$ and  $\mathbf{k} _2= \mathbf{k} _{num}$ yields the $\mathbf{k} _{NW}$ from (4).
>
> - For KRR, we approximate the training loss, which is an average of $\ell_f(x,y) = (f(x)-y)^2 = f^2(x) -2 f(x)y + y^2$.
>      - Assuming $f\in \mathcal{H}(\mathbf{k})$, $\ell_f$  lies in the RKHS of $\mathbf{k}^2(x_1,x_2) + y_1 y_2 \mathbf{k}(x_1,x_2) + y_1^2 y_2^2$, where the last term can be dropped (see Sec. 3 of _Compressed Empirical Measures_ by Grünewälder (2022)), yielding (11) (thanks for pointing out the typo).
>
> -  **Remarkably, our ablation studies in Fig. 2 of the original manuscript and Fig. 2 of the attached PDF confirm this theory.**
>    - Thinning with $\mathbf{k} _{ALG}((x_1,y_1),(x_2,y_2)) = \mathbf{k}(x_1,x_2)$ (orange; Ablation #2 in Sec. 4.1) performs poorly because it only exploits the $x$ information.
>     - Thinning with $\mathbf{k} _{ALG}((x_1,y_1),(x_2,y_2)) = \mathbf{k}(x_1\oplus y_1,x_2\oplus y_2)$ (i.e., ) (brown; Ablation #1 in Sec. 4.1)—despite incorporating information from both $x$ and $y$ by concatenating the two—also yields poor empirical performance.
>     - Thinning with $\mathbf{k} _{ALG} = \mathbf{k} _{NW}$ and $\mathbf{k} _{ALG} = \mathbf{k} _{RR}$ perform the best for the NW and KRR settings, respectively.
>
> $\newline$
>
> $\blacktriangleright$ $\textbf{Improvements to notation and writing}$
> - _"Notation ... instead of for KT-NW, could use...KT-KRR part."_
>
> We will use $S_{NW}$ to denote the coreset for the thinned NW estimator and $S_{RR}$ to denote the coreset for the thinned KRR estimator. We will use $\mathbf{k}$ exclusively to denote the base kernel and use $\mathbf{k} _{ALG}$ to denote the kernel for thinning.
>
> - _"There is not an algorithmic description for Kernel Thinning algorithm."_
>
> We apologize for this oversight. We will add a paragraph in the revised paper and an algorithm with details in the appendix.
>
> - _"Fig 3 and Fig 4: why not use the function in (11)?"_
> - _"Fig 3 and Fig 4: what is the unit of y axis?..."_
>
> In the attached PDF, we have reproduced Fig. 3 and 4 with (11) and improved the formatting. The y-axis is in seconds and we use log-scale to make the difference between ST, KT, and RPCholesky clearer.
>
>  - _"Sec 4.2: what is the conclusion from Fig 5?"_
>
> We will revise the discussion for Fig. 5 as follows: On the California Housing dataset, KT-KRR lies between ST-KRR and RPCholesky in terms of test MSE. In terms of train time, KT-KRR is quite a bit faster than RPCholesky (0.0153 s vs 0.3237 s).
>
> $\newline$
>
> $\blacktriangleright$ $\textbf{Discussion on potential failure cases}$
> - _"Missing a discussion on potential failure cases of the method..."_
>
> We will expand our discussion:
> - The supervised KT guarantees apply when t $f^\star$ lies in the RKHS $\mathcal{H}(\mathbf{k})$, where $\mathbf{k}$ is the base kernel. In practice, choosing $\mathbf{k}$ is indeed a challenge _common to all prior work_.
> - Dwivedi & Mackey (2022), Cor. 1 provide integration-error guarantees for KT when $f^\star \notin \mathcal{H}(\mathbf{k}). Moreover,  there are recent results on finding the best kernel (in testing; Dominigo-Enrich et al. (2023), Sec. 4.2).
> - Combining these results, potentially with the recursive feature machines of Radhakrishnan et al. (2024) (also see our response to Rev PSUT) to build a practical strategy is an exciting future direction.

---

> > ### Comment · Reviewer_Vy1C · 2024-08-07
> > **followup questions**
> >
> > I would like to thank the authors for their detailed reply. Most of my concerns are addressed and I have some followup questions:
> >
> > - In Figure 2b in the attached pdf, why is RPCholesky not included at n=2^14? It is because it is not feasible to run RPCholesky at that point?
> >
> > - Can you also address this question: Fig 2: why is the full not performing the best?

---

> ### Author Response · Authors · 2024-08-08
>
> 1. Thank you for the catch. RPCholesky-KRR achieves an estimated test loss of 0.9980 when $n=2^{14}$  (see Table B below)—close to the Bayes optimal loss of 1. We have estimated the population test loss by sampling 10,000 points from our data generating distribution (as a proxy for integrating), so the estimated test loss can actually be less than 1 with some probability. When we use log-scaling on the y-axis, negative values of excess risk become undefined, so they don't appear on the plot. Note that Full-NW and Full-KRR for $n=2^{14}$ also face this issue. We will make sure to add a note about this in the revision.
> 2. In Fig. 2, Full-NW and Full-KRR are the black box plots (always the first in each group when reading from left to right). Full is deterministic, so the box plots appear as lines. _Note that Full-NW and Full-KRR indeed perform the best across all $n$._ For additional clarity, we have attached tables of the test loss (mean ± standard deviation across 100 trials). (Subtract each value by 1 to get the excess risk values used in Fig. 2.)
>
> **Table A: Test loss for NW estimators in Figure 2a of the attached PDF**
> |                                                       | $n=2^{8}$       | $n=2^{10}$      | $n=2^{12}$      | $n=2^{14}$      |
> |:------------------------------------------------------|:----------------|:----------------|:----------------|:----------------|
> | Full                                                  | **1.1423 ± 0.0000** | **1.0494 ± 0.0000** | **1.0189 ± 0.0000** | **0.9997 ± 0.0000** |
> | RPCholesky                                            | 3.0737 ± 0.2548 | 2.8583 ± 0.3150 | 2.0772 ± 0.2920 | 1.4720 ± 0.1026 |
> | ST                                                    | 3.1181 ± 0.1813 | 2.8844 ± 0.3063 | 2.4720 ± 0.3058 | 1.7632 ± 0.2364 |
> | KT w/ $\mathbf{k}_{\mathrm{RR}}((x_1,y_1),(x_2,y_2))$ | 2.9203 ± 0.4216 | 2.2230 ± 0.4160 | 1.3952 ± 0.0955 | 1.1403 ± 0.0274 |
> | KT w/ $\mathbf{k}(x_1,x_2)$                           | 3.1040 ± 0.1928 | 2.4149 ± 0.2585 | 1.6458 ± 0.1386 | 1.3679 ± 0.0705 |
> | KT w/ $\mathbf{k}((x_1\oplus x_2), (y_1\oplus y_2))$  | 3.0814 ± 0.1648 | 2.6633 ± 0.3603 | 1.7260 ± 0.1791 | 1.4282 ± 0.0825 |
> | KT w/ $\mathbf{k}_{\mathrm{NW}}((x_1,y_1),(x_2,y_2))$ | 2.9838 ± 0.2056 | 2.2423 ± 0.3736 | 1.3799 ± 0.0808 | 1.1363 ± 0.0267
>
> $\newline$
>
> **Table B: Test loss for KRR estimators in Figure 2b of the attached PDF**
> |                                                       | $n=2^{8}$       | $n=2^{10}$      | $n=2^{12}$      | $n=2^{14}$      |
> |:------------------------------------------------------|:----------------|:----------------|:----------------|:----------------|
> | Full                                                  | **1.3750 ± 0.0000** | **1.0584 ± 0.0000** | **1.0121 ± 0.0000** | **0.9981 ± 0.0000** |
> | RPCholesky                                            | 2.7112 ± 0.3636 | 1.7153 ± 0.3839 | 1.0112 ± 0.0029 | 0.9980 ± 0.0000 |
> | ST                                                    | 2.9433 ± 0.3129 | 2.5855 ± 0.3570 | 2.1328 ± 0.3374 | 1.5634 ± 0.1437 |
> | KT w/ $\mathbf{k}_{\mathrm{NW}}((x_1,y_1),(x_2,y_2))$ | 2.3068 ± 0.4381 | 1.9520 ± 0.2783 | 1.4859 ± 0.1146 | 1.2394 ± 0.0538 |
> | KT w/ $\mathbf{k}(x_1,x_2)$                           | 2.8103 ± 0.1677 | 2.4297 ± 0.2559 | 1.7077 ± 0.1433 | 1.4974 ± 0.0885 |
> | KT w/ $\mathbf{k}((x_1\oplus x_2), (y_1\oplus y_2))$  | 3.0312 ± 0.2431 | 2.6189 ± 0.3281 | 2.1086 ± 0.3178 | 1.4590 ± 0.1206 |
> | KT w/ $\mathbf{k}_{\mathrm{RR}}((x_1,y_1),(x_2,y_2))$ | 2.0422 ± 0.3406 | 1.8856 ± 0.2575 | 1.4336 ± 0.1142 | 1.2316 ± 0.0499 |

---

> > ### Comment · Reviewer_Vy1C · 2024-08-12
> >
> > Thank you for your clarification.
> >
> > - I don't get why the loss of 0.9980 becomes undefined when y is in log2 scale?
> >
> > - Additionally, can you provide some reasoning between the performance comparison of Cholesky and the proposed method? In table A, cholesky performs better than the proposed for most of cases, but the trend is reversed in table B.

---

> ### Author Response · Authors · 2024-08-12
> **Response to further comments**
>
> Thank you for your follow-up questions.
>
> 1. The values in Table A and B are test loss while the values in Figure 2 are excess risk. Hence a test loss of 0.9980 becomes an excess risk = test loss - 1 = -0.0020, which is not a valid value on log scale; hence the missing marker for RPCholesky in Figure 2.
>
> 2. Since the Tables are test loss, a lower value is better. So the opposite trend to what you wrote is true. In particular, in Table A, RPCholesky performs **worse** than our proposed method. On the other hand, RPCholesky is tends to do better than our proposed method In Table B. Overall the main takeaways from Table A/B (and Figure 2) is that **our Kernel Thinning based estimator**
>    - _improves upon standard thinning for both NW and KRR estimators across all coreset sizes_
>    - _improves upon RPCholesky for NW estimator across all coreset sizes_
>    - _improves upon RPCholesy for small coreset sizes for KRR, and remains competitive (but a bit worse) in large coreset sizes for KRR._
>
> RPCholesky’s superior performance over KT for KRR is not too surprising since RPCholesky a spectral method tuned to perform better in such settings, while NW estimator does not benefit immediately from an improved spectral approximation.
>
> We remark that these results are consistent with our original motivation of designing a unified framework for speeding up general non-parametric regression estimators using a single recipe.

---

### Author Rebuttal · Authors · 2024-08-07

We thank all the reviewers for their helpful and detailed feedback on our work. We summarize our additions as follows:

$\blacktriangleright$ $\textbf{New experiment on SUSY dataset}$
- In Table 1 of the attached PDF, we have added results on the SUSY dataset ($d=18,N=5\times 10^6$). We use $4\times 10^6$ points for training and the remaining $10^6$ points for testing.
- We compare our method KT-KRR ($\sigma=10,\lambda=10^{-1}$) to several large-scale kernel methods, namely RPCholesky preconditioning, FALKON, and Conjugate Gradient (all with $\sigma=10,\lambda=10^{-3}$). The parameters are chosen with cross-validation.
- We show that KT-KRR achieves test MSE between ST-KRR and RPCholesky with training time almost half that of RPCholesky. These findings are consistent with our results on the California Housing dataset (Fig. 5 in the original manuscript).

$\newline$

$\blacktriangleright$ $\textbf{Addtitional comparisons with RPCholesky}$
- In Fig. 2a of the attached PDF, we have added a new baseline: RPCholesky-NW, where $S_{NW}$ are the pivot points from RPCholesky (see box plots in red).
  - **KT-NW outperforms RPCholesky-NW in terms of both test MSE and train times.** While it provides a good low-rank approximation of the kernel matrix, RPCholesky is not designed to preserve averages. Moreover, RPCholesky requires $\mathcal{O}(n^2)$ time to produce a coreset of size $\sqrt{n}$, whereas KT-NW requires $\mathcal{O}(n \log^3 n)$ time. This difference is reflected in our wall-clock timings (see Fig. 3a of the attached PDF).
- In Fig. 2b of the attached PDF, we have added a new baseline: Restricted KRR using RPCholesky for landmark selection (see box plots in red).
  - Consistent with our real-world experiments, RPCholesky-KRR outperforms KT-KRR in terms of test MSE (Fig. 2b of the attached PDF), but with slower train times (Fig. 3b of the attached PDF).

$\newline$

$\blacktriangleright$ $\textbf{Insight into the choice of kernels}$

We expand the discussion of the following points:
- **Choice of base kernels $\mathbf{k}$**:
  - Our theory for KT-NW permits any bounded, Lipschitz radial kernel with tails that decay at any rate—e.g., compact, sub-Gaussian, sub-exponential, and poly tails.
  - Our theory for KT-KRR permits any kernel satisfying PolyGrowth or LogGrowth log-covering number—e.g., finitely-times differentiable, analytic, finite-rank kernels.
- **Construction of supervised kernels**: We explain the form of $\mathbf{k} _{NW}$ and  $\mathbf{k} _{RR}$ and validate these design choices with our ablation studies (see Fig. 2 of attached PDF and Fig. 2 of original manuscript).
- **Kernel mis-specification**: We describe several potential approaches for finding the right kernel and for deriving guarantees when the regression function $f^\star$ lies outside the RKHS $\mathcal{H}(\mathbf{k})$.
- **Multi-dimensional response data**: Our theory in fact applies effortlessly to the setting when $f^\star$ is multi-dimensional (e.g., for multi-class classification tasks). The resulting supervised kernel looks like the sum of one-dimensional supervised kernels (or equivalently, replacing $y_1 y_2$ with $\langle \mathbf{y} _1,\mathbf{y} _2\rangle$ in $\mathbf{k} _{NW}$ and $\mathbf{k} _{RR}$ ).
- **Minimax rates**: We discuss the minimax rates for each regression problem and although our rates our sub-optimal, we believe we this is a limitation in the analysis and that minimax rates can be achieved for some class of kernels.

$\newline$

We now respond to the specific comments and questions from each review.

---

### Decision · Program_Chairs · 2024-09-25

**Decision:**

Accept (poster)

**Comment:**

This paper introduces a meta-algorithm based on Kernel Thinning for non-parametric regression. The authors apply their meta-algorithm to Nadaraya-Watson regression and Kernel Ridge Regression and prove computational speedups and statistical efficiency in both cases. Overall the paper could benefit from more experimental evidence. In the original set of results there was not much empirical advantage over RPCholesky. Moreover the paper was missing comparisons to other nonparametric regression methods, there were no experiments on large datasets, and the results were limited to kernels with fast decaying tails (excluding Matern kernels). The authors have clarified all of these points in the rebuttal, including new experimental results and a claim (without proof) that their results extend to heavy tailed kernels. I believe that this paper should be accepted for publication despite these omissions. Nevertheless, I urge the authors to include all new results in the camera-ready, as well as a detailed proof for non sub-Gaussian kernels.